# Accessible, Realistic, and Fair Evaluation of Positive-Unlabeled Learning Algorithms

**Wei Wang** [1,2*†]    **Dong-Dong Wu** [2,1*]   **Ming Li** [3,2]
**Jingxiong Zhang** [2]    **Gang Niu** [1]    **Masashi Sugiyama** [1,2]
[1] RIKEN, Tokyo 103-0027, Japan
[2] The University of Tokyo, Chiba 277-8561, Japan
[3] University of Maryland, College Park, MD 20742, USA
wei.wang@riken.jp
{dongdongwu1230,lm1640362161,gang.niu.ml}@gmail.com
7656472907@edu.k.u-tokyo.ac.jp sugi@k.u-tokyo.ac.jp

## Abstract

Positive-unlabeled (PU) learning is a weakly supervised binary classification problem, in which the goal is to learn a binary classifier from only positive and unlabeled data, without access to negative data. In recent years, many PU learning algorithms have been developed to improve model performance. However, experimental settings are highly inconsistent, making it difficult to identify which algorithm performs better. In this paper, we propose the first PU learning benchmark to systematically compare PU learning algorithms. During our implementation, we identify subtle yet critical factors that affect the realistic and fair evaluation of PU learning algorithms. On the one hand, many PU learning algorithms rely on a validation set that includes negative data for model selection. This is unrealistic in traditional PU learning settings, where no negative data are available. To handle this problem, we systematically investigate model selection criteria for PU learning. On the other hand, PU learning involves different problem settings and corresponding solution families, i.e., the one-sample and two-sample settings. However, existing evaluation protocols are heavily biased towards the one-sample setting and neglect the significant difference between them. We identify the internal label shift problem of unlabeled training data for the one-sample setting and propose a simple yet effective calibration approach to ensure fair comparisons within and across families. We hope our framework will provide an accessible, realistic, and fair environment for evaluating PU learning algorithms in the future.

## 1 Introduction

In binary classification, both positive and negative data are usually necessary to train an effective classifier. However, in many real-world applications, collecting negative data can be more challenging than collecting positive data (Hsieh et al., 2015; Zhou et al., 2021). In positive-unlabeled (PU) learning, only positive and unlabeled data are needed. The objective is to train a binary classifier that assigns positive or negative labels to unseen instances. Therefore, PU learning is a promising weakly supervised binary classification approach for many real-world problems where negative data are difficult to obtain, including recommender systems (Yi et al., 2017; Chen et al., 2023), anomaly detection (Ju et al., 2020; Tian et al., 2024; Takahashi et al., 2025), knowledge graphs (Yin et al., 2024), and link prediction (Wu et al., 2024; Mao et al., 2025).

In recent years, there has been significant progress in PU learning algorithms. PU learning can be divided into three groups: cost-sensitive PU learning algorithms (du Plessis et al., 2014; Zhao et al., 2022), sample-selection PU learning algorithms (Chen et al., 2020b; Wang et al., 2023a), and biased PU learning algorithms (Teisseyre et al., 2025). Cost-sensitive algorithms assign different weights to positive and unlabeled data to approximate the classification risk. Sample-selection algorithms select high-confidence negative data from unlabeled data, which are then given to supervised learning

---

*Equal contributions. [†]Corresponding author: Wei Wang <wei.wang@riken.jp>.

algorithms. Biased PU learning algorithms model the biased generation process of positive data and exploit various correction approaches.

Although many PU learning algorithms have been developed to improve generalization performance, there is a lack of a unified experimental setup in the literature for fairly comparing different PU learning algorithms. The experimental settings of different papers are not consistent with each other, making it difficult to tell which algorithm is better. It has been observed that subtle differences in experimental settings can greatly affect the model performance of PU learning algorithms. Additionally, subtle algorithm details, including data augmentation, algorithm tricks, and warm-up strategies, can also greatly affect model performance (Zhu et al., 2023b; Wang et al., 2023a). Therefore, a unified experimental protocol is necessary to further promote the development of PU learning algorithms. In this paper, we propose the first PU learning benchmark to systematically and fairly compare state-of-the-art PU learning algorithms with unified experimental settings. We propose careful and unified implementations of the data generation, algorithm training, and evaluation processes for PU learning algorithms. This makes it easier for users to validate the effectiveness of their newly developed algorithms.

In our implementations, we observe that many PU learning algorithms rely on a validation set containing both positive and negative data for meta-learning, model selection, or early stopping (Chen et al., 2020b; Zhu et al., 2023b; Long et al., 2024). However, accessing negative data is unrealistic and contradicts the original motivation of PU learning (Elkan & Noto, 2008), which goes against the advantages of PU learning in not depending on negative data. Actually, if we can obtain some negative data, we can directly apply supervised learning techniques, which can greatly boost model performance (Sakai et al., 2017). Therefore, standardizing the composition and use of the validation set is vital to fairly and practically evaluating PU learning algorithms. In this paper, we systematically revisit the model selection criteria for PU learning by using only positive and unlabeled validation data, and validate their effectiveness with both theoretical and empirical analyses.

In addition, there are different families and corresponding solutions of PU learning algorithms, but existing evaluations fail to consider the differences between these families. From the perspective of data generation processes, there are two types of PU learning problems: the one-sample (OS) and two-sample (TS) settings. In the OS setting, the positive and unlabeled training sets are generated sequentially. An unlabeled

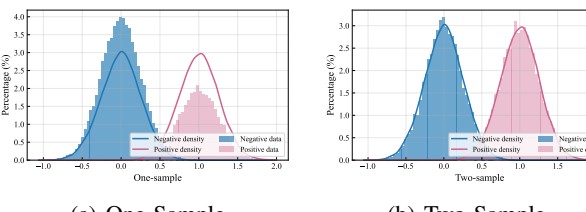

| (a) One-Sample | (b) Two-Sample |

Figure 1: An example of the comparison of the distribution of unlabeled training data in different PU learning settings.

dataset is first sampled from the marginal density. Then, if an instance in the unlabeled dataset is positive, its positive label is observed with a constant probability. If an instance in the unlabeled dataset is negative, its label is never observed, and the instance remains unlabeled. Finally, the observed positive data constitute the positive training set, while the remaining unlabeled data constitute the unlabeled training set. In the TS setting, the positive and unlabeled training sets are generated independently, meaning that the density of unlabeled training data is the same as the marginal density. This indicates that the density of unlabeled training data is different in these two settings. Figure 1 shows an example of the distribution of unlabeled data under the OS and TS settings. We can find that the class priors of the two settings are different. This causes an internal label shift (ILS) problem for the unlabeled training data when adopting the OS setting as the evaluation setting. Unfortunately, this problem has typically been overlooked. Existing evaluation protocols are heavily biased towards the OS setting and compare OS and TS algorithms together without specific manipulations. This can deteriorate the performance of TS PU learning algorithms and lead to unfair experimental comparisons. Therefore, we identify the ILS problem for the first time in the PU learning literature and propose a simple yet effective calibration approach to overcome it with theoretical guarantees.

We draw the following key takeaways from our benchmark results:

- No single algorithm outperforms all others on every dataset or evaluation metric; some early, simple methods already achieve strong classification performance. Therefore, we should choose which PU learning algorithm to use on a case-by-case basis.

- The model-selection problem in PU learning must be addressed when designing new algorithms or conducting empirical comparisons, and different selection criteria should be used for different test metrics.
- The performance of TS PU learning algorithms degrades significantly when evaluated under the OS setting without proper adaptation, so OS protocols in the existing PU learning literature do not reflect the true performance of TS methods. Hence, differences between OS and TS settings must be considered to ensure fair cross-family comparisons.

## 2 PRELIMINARIES

In this section, we present the background of PU learning and existing state-of-the-art algorithms.

### 2.1 POSITIVE-UNLABELED LEARNING

**Problem Setting.** Let $\mathcal{X} \subseteq \mathbb{R}^d$ denote the $d$-dimensional feature space and $\mathcal{Y} = \{+1, -1\}$ denote the binary label space. Let $p(\boldsymbol{x}, y)$ denote the joint probability density over the random variables $(\boldsymbol{x}, y) \in \mathcal{X} \times \mathcal{Y}$. In PU learning, we are given a positive training set $D_\mathrm{P} = \{(\boldsymbol{x}_i, +1)\}_{i=1}^{n_\mathrm{P}}$ and an unlabeled training set $D_\mathrm{U} = \{\boldsymbol{x}_i\}_{i=n_\mathrm{P}+1}^{n_\mathrm{P}+n_\mathrm{U}}$. Let $\pi = p(y = +1)$ denote the class prior probability of the positive class. Let $p(\boldsymbol{x}|y = +1)$ and $p(\boldsymbol{x}|y = -1)$ denote the positive and negative class-conditional densities, respectively. Let $p(\boldsymbol{x})$ denote the marginal density. The goal of PU learning is to learn a binary classifier $f : \mathcal{X} \to \mathbb{R}$ from $D_\mathrm{P} \bigcup D_\mathrm{U}$ that maximizes the *expected accuracy*

$$\mathrm{ACC}(f) = \mathbb{E}_{p(\boldsymbol{x}, y)} \mathbb{I}\left(y f(\boldsymbol{x}) \geqslant 0\right), \tag{1}$$

where $\mathbb{E}$ denotes the expectation and $\mathbb{I}$ denotes the indicator function. However, since the 0-1 loss function is difficult to optimize, we usually use a surrogate loss function $\ell$, such as the logistic loss. Then, the *classification risk* to be minimized can be expressed as

$$R(f) = \mathbb{E}_{p(\boldsymbol{x}, y)}\left[\ell\left(f\left(\boldsymbol{x}\right), y\right)\right]. \tag{2}$$

**Data Generation Assumption.** There are mainly two data generation assumptions for PU learning, i.e., the TS setting (du Plessis et al., 2014; Niu et al., 2016; Chen et al., 2020a) and the OS setting (Elkan & Noto, 2008; Coudray et al., 2023). In the TS setting, we assume that $\mathcal{D}_\mathrm{P}$ and $\mathcal{D}_\mathrm{U}$ are generated *independently*, where $\mathcal{D}_\mathrm{P}$ is sampled from the positive conditional density $p(\boldsymbol{x}|y = +1)$ and $\mathcal{D}_\mathrm{U}$ is sampled from the marginal density $p(\boldsymbol{x})$. In the OS setting, $\mathcal{D}_\mathrm{U}$ and $\mathcal{D}_\mathrm{P}$ are generated *sequentially*. First, $\mathcal{D}_\mathrm{U}$ is sampled from the marginal density $p(\boldsymbol{x})$. Second, for each example in $\mathcal{D}_\mathrm{U}$, if it is positive, its positive label is observed with a *constant probability* $c > 0$. If an example is negative, its negative label is never observed and the example remains unlabeled with probability 1. Finally, the observed positive data constitute $\mathcal{D}_\mathrm{P}$ and all the unlabeled data left constitute $\mathcal{D}_\mathrm{U}$.

### 2.2 POSITIVE-UNLABELED LEARNING ALGORITHMS

From a methodology taxonomy perspective, PU learning algorithms can be divided into three groups: cost-sensitive algorithms, sample-selection algorithms, and biased PU learning algorithms. Cost-sensitive algorithms assign different weights to positive and unlabeled data to approximate the classification risk (du Plessis et al., 2015; Kiryo et al., 2017; Hsieh et al., 2019). Some algorithms are equipped with other regularization techniques to further improve performance, such as entropy minimization (Zhao et al., 2022; Jiang et al., 2023) and mixup technique (Chen et al., 2020a; Li et al., 2022). Sample-selection algorithms select reliable negative examples from the unlabeled dataset for supervised learning (Chen et al., 2020b; Garg et al., 2021; Wang et al., 2023a; Li et al., 2024). Biased PU learning algorithms consider the density of positive data to be biased and adopt different strategies to model the bias (Bekker et al., 2019; Gong et al., 2022; Coudray et al., 2023; Wang et al., 2023b; Teisseyre et al., 2025).

## 3 MODEL SELECTION FOR POSITIVE-UNLABELED LEARNING

In this section, we first explain our motivation for studying the model selection problem in PU learning. Next, we review the criteria used for model selection in PU learning, including the proxy accuracy, proxy area under the curve score, and oracle accuracy.

## 3.1 MOTIVATION

Although model selection is well established for supervised learning, it is non-trivial for PU learning because negative data are inaccessible. This problem is particularly important for deep learning algorithms because they have many hyperparameters, including universal hyperparameters (e.g., learning rates and weight decay) and algorithm-specific hyperparameters. Previous work has usually conducted model selection by assuming a validation set with labels (i.e., both positive and negative labels) is available. However, this assumption is inconsistent with the definition of PU learning, in which negative data are unavailable. Therefore, it is important to study the model selection problem systematically for PU learning. According to the original definition of PU learning (Bekker & Davis, 2020), we assume that the validation set consists of a positive validation set $D'_{\mathrm{P}} = \{(\boldsymbol{x}'_i, +1)\}_{i=1}^{n'_{\mathrm{P}}}$ and an unlabeled validation set $D'_{\mathrm{U}} = \{\boldsymbol{x}'_i\}_{i=n'_{\mathrm{P}}+1}^{n'_{\mathrm{P}}+n'_{\mathrm{U}}}$.

## 3.2 PROXY ACCURACY

Although the validation accuracy cannot be directly calculated because of the absence of negative data, it has been shown that the expected accuracy can be expressed using only positive and unlabeled data (du Plessis et al., 2014). This motivates us to apply it for model selection.

**Definition 1** (Proxy accuracy (PA)). The proxy accuracy of a binary classifier $f$ on the PU validation dataset is defined as

$$\mathrm{PA}(f) = \begin{cases} \frac{2\pi}{n'_{\mathrm{P}}} \sum_{i=1}^{n'_{\mathrm{P}}} \mathbb{I}\left(f(\boldsymbol{x}'_i) \geqslant 0\right) + \frac{1}{n'_{\mathrm{U}}} \sum_{i=n'_{\mathrm{P}}+1}^{n'_{\mathrm{P}}+n'_{\mathrm{U}}} \mathbb{I}\left(f(\boldsymbol{x}'_i) < 0\right), & \text{if the setting is TS;} \\ \frac{2\pi}{n'_{\mathrm{P}}} \sum_{i=1}^{n'_{\mathrm{P}}} \mathbb{I}\left(f(\boldsymbol{x}'_i) \geqslant 0\right) + \frac{1}{n'_{\mathrm{P}}+n'_{\mathrm{U}}} \sum_{i=1}^{n'_{\mathrm{P}}+n'_{\mathrm{U}}} \mathbb{I}\left(f(\boldsymbol{x}'_i) < 0\right), & \text{if the setting is OS.} \end{cases} \quad (3)$$

PA can be calculated using only PU validation data when the class prior $\pi$ is known or estimated (Ramaswamy et al., 2016; Yao et al., 2022; Zhu et al., 2023a). The following proposition then holds.

**Proposition 1.** For two classifiers $f_1$ and $f_2$ that satisfy $\mathbb{E}\left[\mathrm{PA}(f_1)\right] < \mathbb{E}\left[\mathrm{PA}(f_2)\right]$, we have $\mathrm{ACC}(f_1) < \mathrm{ACC}(f_2)$.

The proof can be found in Appendix B.1. According to Proposition 1, a classifier with a higher expected value of the proxy accuracy can achieve a higher expected accuracy even when the true labels are inaccessible. This means that when the number of validation data is large, the best model chosen using the PA metric will achieve the highest accuracy in expectation. One limitation of PA is that knowledge of the class prior is necessary. However, knowledge of $\pi$ is an intrinsic and common issue in PU learning. Addressing this issue is beyond the scope of our paper. In practice, we can estimate it using off-the-shelf estimation methods (Ramaswamy et al., 2016; Garg et al., 2021; Yao et al., 2022), and we can even obtain this knowledge in some real-world applications (Sugiyama et al., 2022).

## 3.3 PROXY AUC SCORE

It has been shown that the area under the curve (AUC) score can be robust to corrupted labels for binary classification (Charoenphakdee et al., 2019; Wei et al., 2022). Therefore, it is promising to employ it for PU model selection. First, we introduce the expected AUC score as follows:

$$\mathrm{AUC}(f) = \mathbb{E}_{p(\boldsymbol{x}|y=+1)} \mathbb{E}_{p(\boldsymbol{x}'|y'=-1)} \left[ \mathbb{I}\left(f(\boldsymbol{x}) > f(\boldsymbol{x}')\right) + \frac{1}{2} \mathbb{I}\left(f(\boldsymbol{x}) = f(\boldsymbol{x}')\right) \right]. \quad (4)$$

We then consider the unlabeled validation data to be corrupted negative data and calculate the AUC score as follows, which is suitable for both OS and TS settings.

**Definition 2** (Proxy AUC score (PAUC)). The proxy AUC of a binary classifier $f$ on the PU validation dataset is defined as

$$\mathrm{PAUC}(f) = \frac{1}{n'_{\mathrm{P}} n'_{\mathrm{U}}} \sum_{i=1}^{n'_{\mathrm{P}}} \sum_{j=n'_{\mathrm{P}}+1}^{n'_{\mathrm{P}}+n'_{\mathrm{U}}} \left( \mathbb{I}\left(f(\boldsymbol{x}'_i) > f(\boldsymbol{x}'_j)\right) + \frac{1}{2} \mathbb{I}\left(f(\boldsymbol{x}'_i) = f(\boldsymbol{x}'_j)\right) \right). \quad (5)$$

The following proposition then holds.

**Proposition 2.** *Under both the OS and TS settings, for two classifiers $f_1$ and $f_2$ that satisfy* $\mathbb{E}\left[\text{PAUC}(f_1)\right] < \mathbb{E}\left[\text{PAUC}(f_2)\right]$*, we have* $\text{AUC}(f_1) < \text{AUC}(f_2)$.

The proof can be found in Appendix B.2. Proposition 2 shows that a classifier with a higher expected value of the proxy AUC score will achieve a higher expected AUC score, regardless of whether the setting is OS or TS. Therefore, when the number of validation data is large, the model selected with the highest PAUC can also achieve the highest expected value of the AUC score. An advantage is that the class prior $\pi$ is not necessary when calculating the PAUC.

### 3.4 Oracle Accuracy

Finally, we introduce the oracle accuracy metric if the true labels of unlabeled data are available.

**Definition 3** (Oracle accuracy (OA)). The oracle accuracy of a binary classifier $f$ on the PU validation dataset is defined as

$$\text{OA}(f) = \begin{cases} \frac{1}{n'_{\text{U}}} \sum_{i=n'_{\text{P}}+1}^{n'_{\text{P}}+n'_{\text{U}}} \mathbb{I}\left(y'_i f(\boldsymbol{x}'_i) \geqslant 0\right), & \text{if the setting is TS;} \\ \frac{1}{n'_{\text{P}}+n'_{\text{U}}} \sum_{i=1}^{n'_{\text{P}}+n'_{\text{U}}} \mathbb{I}\left(y'_i f(\boldsymbol{x}'_i) \geqslant 0\right), & \text{if the setting is OS.} \end{cases} \tag{6}$$

Here, $y'_i$ is the true label of $\boldsymbol{x}'_i$.

Notably, the implementations for the OS and TS settings differ slightly, as it is important to ensure that the validation data have the same distribution as the test data. OA is a natural metric for supervised learning. However, due to the absence of negative data, it cannot be calculated in the traditional PU learning setting. Unfortunately, this metric has actually been widely used in the PU learning literature because of a lack of standardized benchmarking. Therefore, this paper only includes the results of OA for comparison. We recommend using PA and PAUC in future PU learning experiments, especially in real-world applications where negative data cannot be obtained.

## 4 Internal Label Shift in Positive-Unlabeled Learning

In this section, we first introduce the ILS problem in PU learning. Then, we provide a calibration approach to solve it with both theoretical and empirical analysis.

### 4.1 Problem Statement

The difference between the OS and TS settings lies in the density of the unlabeled training data. Specifically, the density of the unlabeled training data equals the marginal density in the TS setting but differs from it in the OS setting. We formalize the ILS problem as follows.

**Definition 4** (Internal label shift in OS PU learning). In the OS setting, the density of $\mathcal{D}_{\text{U}}$ is $\bar{p}(\boldsymbol{x}) = \bar{\pi} p(\boldsymbol{x}|y=+1) + (1-\bar{\pi})p(\boldsymbol{x}|y=-1)$, where $\bar{\pi}$ is the class prior under the OS setting. Here, the positive and negative class-conditional densities are the same as those of the test data; however, the class prior is $\bar{\pi} = (1-c)\pi/(1-c\pi)$, which differs from $\pi$, the class prior of the test data. This mismatch causes an internal label shift between the unlabeled training data and the test data.

Many cost-sensitive PU learning algorithms have been developed for the TS setting. In these algorithms, positive and unlabeled data are assigned different weights to approximate the classification risk (du Plessis et al., 2014; Chen et al., 2020a; Zhao et al., 2022). Because the weights are theoretically derived, small discrepancies in data assumptions can degrade performance. Conversely, sample-selection PU learning algorithms select reliable negative data from $\mathcal{D}_{\text{U}}$ and need not rely strictly on the specific data generation process (Zhu et al., 2023b; Wang et al., 2023a; Li et al., 2024). However, many papers adopt only the OS setting and ignore the distribution mismatch, causing experimental datasets to violate the assumptions of TS approaches.

To demonstrate how ILS affects model performance, we use uPU (du Plessis et al., 2015) as an example in Section 4; it is a representative TS algorithm and underpins many subsequent cost-sensitive methods.[1] Under the TS assumption $\mathcal{D}_{\text{U}} \overset{\text{i.i.d.}}{\sim} p(\boldsymbol{x})$, du Plessis et al. (2015) proposed the

---

[1] Our analysis and calibration approach can be extended to other TS algorithms as well.

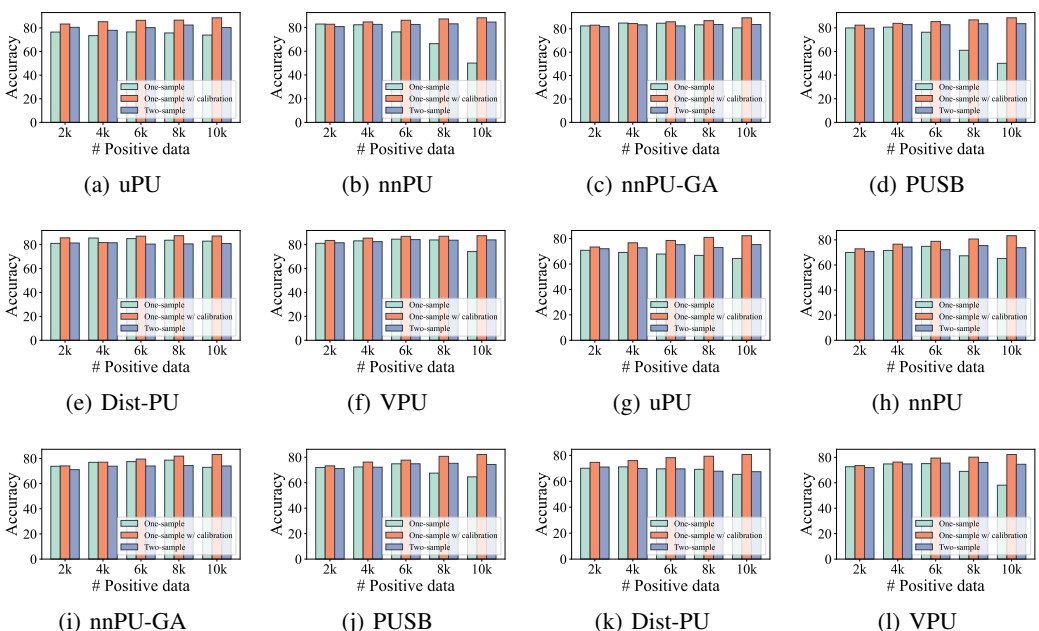

Figure 2: Classification accuracies of TS PU learning algorithms in OS and TS settings of a PU version of CIFAR-10 with varying amounts of positive data. Figures (a) to (f) are for Case 1, and Figures (g) to (l) are for Case 2.

unbiased risk estimator (URE)

$$\widehat{R}(f) = \frac{\pi}{n_{\mathrm{P}}} \sum_{i=1}^{n_{\mathrm{P}}} \left( \ell\left( f(\boldsymbol{x}_i), +1 \right) - \ell\left( f(\boldsymbol{x}_i), -1 \right) \right) + \frac{1}{n_{\mathrm{U}}} \sum_{i=n_{\mathrm{P}}+1}^{n_{\mathrm{P}}+n_{\mathrm{U}}} \ell\left( f(\boldsymbol{x}_i), -1 \right), \qquad (7)$$

which enjoys risk consistency because $\mathbb{E}[\widehat{R}(f)] = R(f)$. Let $\widehat{f} = \arg\min_{f \in \mathcal{F}} \widehat{R}(f)$ and $f^* = \arg\min_{f \in \mathcal{F}} R(f)$ denote the classifiers that minimize the empirical risk in Eq. (7) and the risk in Eq. (2), respectively, where $\mathcal{F}$ is the model class. It is known that $\widehat{f} \to f^*$ as $n_{\mathrm{P}} \to \infty$ and $n_{\mathrm{U}} \to \infty$ under the TS setting (Niu et al., 2016). Under the OS setting, however, $\mathbb{E}[\widehat{R}(f)] \neq R(f)$, so $\widehat{f} \to f^*$ no longer holds (see Appendix B.3). Consequently, minimizing losses designed for the TS setting may not yield high-performing classifiers when datasets are generated under the OS setting, leading to unfair comparisons when all methods are evaluated in the OS setting. The bias stems from the ILS problem: under the OS setting, the class prior of $\mathcal{D}_{\mathrm{U}}$ differs from $\pi$, breaking the consistency of many TS algorithms and degrading their performance.

## 4.2 THE PROPOSED CALIBRATION APPROACH

To address the bias, we incorporate the true densities of $\mathcal{D}_{\mathrm{U}}$ for TS algorithms. The following theorem shows that the risk rewrite for the uPU approach differs under the OS setting.

**Theorem 1.** *Under the OS setting, the classification risk in Eq. (2) can be equivalently expressed as*

$$R(f) = \pi \mathbb{E}_{p(\boldsymbol{x}|y=+1)} \left[ \ell(f(\boldsymbol{x}), +1) + (c-1)\ell(f(\boldsymbol{x}), -1) \right] + (1 - c\pi) \, \mathbb{E}_{\bar{p}(\boldsymbol{x})} \left[ \ell(f(\boldsymbol{x}), -1) \right].$$

The proof is given in Appendix B.4. Theorem 1 shows that the classification risk can be equivalently expressed as expectations w.r.t. the densities of positive and unlabeled data under the OS setting. We then obtain a calibrated risk estimator using the positive and unlabeled datasets:

$$\bar{R}(f) = \frac{\pi}{n_{\mathrm{P}}} \sum_{i=1}^{n_{\mathrm{P}}} \left( \ell\left( f(\boldsymbol{x}_i), +1 \right) + (c-1)\ell\left( f(\boldsymbol{x}_i), -1 \right) \right) + \frac{1 - c\pi}{n_{\mathrm{U}}} \sum_{i=n_{\mathrm{P}}+1}^{n_{\mathrm{P}}+n_{\mathrm{U}}} \ell\left( f(\boldsymbol{x}_i), -1 \right). \qquad (8)$$

When the class prior $\pi$ is known or estimated, we obtain an unbiased estimate of $c$ as $c = n_{\mathrm{P}}/\pi(n_{\mathrm{P}} + n_{\mathrm{U}})$. Let $\bar{f} = \arg\min_{f \in \mathcal{F}} \bar{R}(f)$ denote the optimal classifier that minimizes the calibrated risk

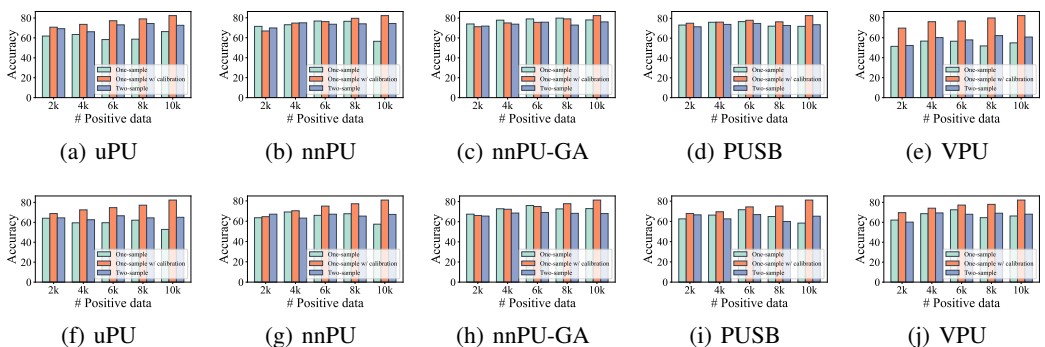

Figure 3: Classification accuracies of TS PU learning algorithms in OS and TS settings of a PU version of ImageNette with varying amounts of positive data. Figures (a) to (e) are for Case 1, and Figures (f) to (j) are for Case 2.

estimator in Eq. (8). Let $\mathfrak{R}_{n_P}(\mathcal{F})$ and $\mathfrak{R}'_{n_U}(\mathcal{F})$ denote the Rademacher complexities defined in Appendix B.5. Then, the following theorem holds.

**Theorem 2.** *Assume that there exists a constant $C_f$ such that $\sup_{f\in\mathcal{F}}\|f\|_\infty \leqslant C_f$ and a constant $C_\ell$ such that $\forall y, \sup_{|z|\leqslant C_f}\ell(z,y) \leqslant C_\ell$. We also assume that $\forall y$, the binary loss function $\ell(z,y)$ is Lipschitz continuous in $z$ with a Lipschitz constant $L_\ell$. For any $\delta > 0$, the following inequality holds with probability at least $1 - \delta$:*

$$R(\bar{f}) - R(f^*) \leqslant (8 - 4c)\pi L_\ell \mathfrak{R}_{n_P}(\mathcal{F}) + (4 - 4c\pi)L_\ell \mathfrak{R}'_{n_U}(\mathcal{F})$$
$$+ \left(\frac{(4-2c)\pi C_\ell}{\sqrt{n_P}} + \frac{(2-2c\pi)C_\ell}{\sqrt{n_U}}\right)\sqrt{\frac{\ln 2/\delta}{2}}. \tag{9}$$

The proof is given in Appendix B.5. Theorem 2 shows that $\bar{f} \to f^*$ as $n_P \to \infty$ and $n_U \to \infty$, because $\mathfrak{R}_{n_U,\bar{p}}(\mathcal{F}) \to 0$ and $\mathfrak{R}_{n_P,p_+}(\mathcal{F}) \to 0$ for all parametric models with a bounded norm, such as deep neural networks trained with weight decay (Golowich et al., 2018). Notably, Eq. (8) can be equivalently transformed into Eq. (7) if we incorporate $\mathcal{D}_P$ into $\mathcal{D}_U$ when computing the last loss term w.r.t. unlabeled data in Eq. (7) (see Appendix B.6). Thus, when $\mathcal{D}_P$ is used in both loss terms, the ILS bias is eliminated, because the union of positive and unlabeled data is unbiased w.r.t. the marginal density. This motivates a simple yet effective calibration approach that adapts TS algorithms to the OS setting, summarized in Algorithm 1. We augment $\mathcal{D}_U$ with $\mathcal{D}_P$ when computing the loss on unlabeled data, so the replenished set is marginally unbiased and suitable for TS PU learners.

### 4.3 EMPIRICAL ANALYSIS

We validated the existence of the ILS problem and the effectiveness of the proposed calibration approach. We used uPU (du Plessis et al., 2015), nnPU (Kiryo et al., 2017), nnPU-GA (Kiryo et al., 2017), PUSB (Kato et al., 2019), VPU (Chen et al., 2020a), and Dist-PU (Zhao et al., 2022), six representative TS PU learning algorithms. We used

---

**Algorithm 1** Calibrated Two-Sample PU Learning

**Require:** Two-sample PU learning algorithm $\mathcal{A}$, positive training set $\mathcal{D}_P$, unlabeled training set $\mathcal{D}_U$, maximum epochs $T_{\max}$, maximum iterations $I_{\max}$.
**Ensure:** Classifier $f$ produced by $\mathcal{A}$.
1: **for** $t = 1, 2, \ldots, T_{\max}$ **do**
2:     **Shuffle** $\mathcal{D}_P$ and $\mathcal{D}_U$;
3:     **for** $k = 1, \ldots, I_{\max}$ **do**
4:         **Fetch** mini-batch $\mathcal{D}_k^P$ from $\mathcal{D}_P$ and $\mathcal{D}_k^U$ from $\mathcal{D}_U$;
5:         **Call** $\mathcal{A}.\text{TRAIN\_ONE\_BATCH}(\mathcal{D}_k^P, \mathcal{D}_k^U \bigcup \mathcal{D}_k^P)$
6:     **end for**
7: **end for**

---

CIFAR-10 (Krizhevsky & Hinton, 2009) and ImageNette (Deng et al., 2009) as the datasets. We synthesized PU training datasets with different definitions of positive and negative labels, where the details are presented in Appendix C. We did not include the results of Dist-PU on ImageNette since Dist-PU did not work well on this dataset. We considered both the OS and TS cases using the same experimental settings, and the only difference lay in how positive data were generated. Figures 2 and 3 show the experimental results on CIFAR-10 and ImageNette with varying amounts of positive data, respectively. We can observe that using TS approaches directly in the OS setting yields inferior performance. Their performance consistently drops when the number of positive data increases,

Table 1: Test results (mean±std) of accuracy, AUC, and F1 score for each algorithm on CIFAR-10 (Case 1) under different model selection criteria. The best performance w.r.t. each validation metric is shown in bold. Here, "-c" indicates using the proposed calibration technique in Algorithm 1.

| Test metric | Accuracy | | | AUC | | | F1 | | |
|---|---|---|---|---|---|---|---|---|---|
| Val metric | PA | PAUC | OA | PA | PAUC | OA | PA | PAUC | OA |
| PUbN | 86.46±0.46 | 86.24±0.84 | 87.33±0.28 | 93.96±0.49 | 93.44±0.74 | 94.33±0.23 | 86.62±0.52 | 86.07±0.98 | 86.98±0.24 |
| PAN | 76.64±0.78 | 77.56±0.41 | 78.91±0.59 | 87.11±0.86 | 87.28±0.93 | 85.70±0.68 | 79.08±0.73 | 79.41±0.61 | 78.74±0.95 |
| CVIR | 85.45±1.03 | 83.32±0.44 | 86.47±0.48 | 93.74±0.73 | 93.67±0.62 | 93.73±0.31 | 86.19±0.88 | 84.71±0.33 | 86.51±0.40 |
| P3MIX-E | 72.68±6.26 | 50.00±0.00 | 73.96±5.63 | 88.80±2.65 | 92.62±0.67 | 89.56±2.18 | 77.65±3.65 | 66.67±0.00 | 67.45±12.03 |
| P3MIX-C | 86.36±0.58 | 85.75±0.76 | 86.65±0.57 | 92.70±0.71 | 93.09±0.65 | 93.16±0.43 | 86.44±0.51 | 85.93±0.70 | 86.72±0.58 |
| LBE | 82.71±0.73 | 73.60±1.29 | 85.03±0.38 | 92.09±0.15 | 93.21±0.04 | 92.26±0.31 | 83.79±0.49 | 78.72±0.76 | 84.31±0.31 |
| Count Loss | 80.89±0.32 | 79.86±0.88 | 82.39±0.37 | 90.63±0.69 | 90.40±0.45 | 89.20±1.27 | 82.60±0.28 | 81.83±0.39 | 83.11±0.39 |
| Robust-PU | 85.57±0.18 | 85.61±0.55 | 85.91±0.35 | 91.56±0.49 | 92.89±0.29 | 91.04±1.60 | 85.88±0.09 | 84.80±0.96 | 85.47±0.32 |
| Holistic-PU | 50.20±0.10 | 50.00±0.00 | 81.81±0.49 | 64.56±11.51 | 69.45±5.04 | 90.60±0.41 | 66.64±0.03 | 66.67±0.00 | 82.97±0.37 |
| PUe | 77.85±0.85 | 78.51±0.33 | 80.45±0.46 | 86.84±0.61 | 86.60±0.45 | 87.58±0.44 | 79.45±0.55 | 78.01±0.48 | 78.99±0.28 |
| GLWS | 84.46±0.45 | 79.83±2.30 | 85.66±0.44 | 93.55±0.07 | 93.54±0.14 | 93.48±0.16 | 85.65±0.36 | 82.69±1.46 | 86.26±0.32 |
| uPU | 80.24±1.25 | 76.07±2.83 | 82.04±0.49 | 88.72±0.40 | 89.05±0.17 | 87.36±0.73 | 81.05±0.90 | 77.01±1.41 | 80.34±0.56 |
| uPU-c | 85.89±0.44 | 84.20±0.49 | 86.48±0.21 | 92.65±0.38 | 93.03±0.22 | 93.22±0.15 | 85.96±0.43 | 83.04±0.92 | 86.12±0.10 |
| nnPU | 82.03±0.11 | 75.56±0.29 | 82.40±0.31 | 92.62±0.15 | 92.32±0.47 | 91.95±0.44 | 83.51±0.05 | 79.64±0.25 | 83.49±0.05 |
| nnPU-c | 85.52±0.20 | 86.03±0.68 | 86.35±0.26 | 92.19±0.33 | 93.07±0.55 | 92.95±0.38 | 85.90±0.28 | 85.71±0.70 | 86.29±0.30 |
| nnPU-GA | 84.26±0.80 | 84.18±0.40 | 84.93±0.70 | 92.79±0.47 | 92.26±0.36 | 92.25±0.46 | 84.87±0.62 | 84.63±0.42 | 84.58±0.53 |
| nnPU-GA-c | 85.80±0.29 | 86.28±0.31 | 86.13±0.25 | 92.81±0.42 | 92.96±0.47 | 93.00±0.42 | 85.90±0.31 | 85.66±0.27 | 85.57±0.19 |
| PUSB | 81.53±0.77 | 82.49±1.02 | 82.91±0.70 | 81.53±0.77 | 82.49±1.02 | 82.91±0.70 | 83.29±0.47 | 83.80±0.77 | 84.12±0.53 |
| PUSB-c | 86.15±0.37 | 84.76±0.17 | 86.49±0.17 | 86.15±0.37 | 84.76±0.17 | 86.49±0.17 | 86.09±0.44 | 83.89±0.19 | 86.23±0.18 |
| VPU | 84.93±0.52 | 65.71±7.32 | 85.80±0.40 | 91.89±0.08 | 92.89±0.54 | 92.86±0.20 | 84.15±0.59 | 42.73±17.09 | 84.91±0.49 |
| VPU-c | 86.41±0.75 | 82.85±1.68 | 87.65±0.25 | 92.30±0.31 | 93.51±0.53 | 91.79±1.62 | 86.73±0.55 | 84.56±1.15 | 87.41±0.29 |
| Dist-PU | 81.64±0.45 | 79.31±0.51 | 83.56±0.46 | 90.91±0.54 | 91.90±0.48 | 90.59±0.49 | 83.34±0.26 | 81.94±0.23 | 83.26±0.60 |
| Dist-PU-c | **87.06±0.45** | **87.38±0.23** | **88.47±0.25** | **94.93±0.31** | **94.55±0.21** | **94.90±0.32** | **87.63±0.33** | **87.28±0.29** | **88.18±0.25** |

even though we have more knowledge of the true labels of positive data in the unlabeled dataset. By using our proposed calibration approach, the performance can be improved greatly and can even sometimes surpass the performance in the TS setting. This shows the effectiveness of our calibration approach in improving TS approaches under the OS setting.

## 5 BENCHMARKING POSITIVE-UNLABELED LEARNING

In this section, we first introduce the benchmark settings, then we present the benchmark experimental results. The code package is available at https://github.com/wu-dd/PUBench.

### 5.1 BENCHMARK SETTINGS

We included seventeen representative PU learning algorithms: uPU (du Plessis et al., 2015), nnPU (Kiryo et al., 2017), nnPU-GA (Kiryo et al., 2017), PUSB (Kato et al., 2019), PUbN (Hsieh et al., 2019), VPU (Chen et al., 2020a), PAN (Hu et al., 2021), CVIR (Garg et al., 2021), Dist-PU (Zhao et al., 2022), P³MIX-E (Li et al., 2022), P³MIX-C (Li et al., 2022), LBE (Gong et al., 2022), Count Loss (Shukla et al., 2023), Robust-PU (Zhu et al., 2023b), Holistic-PU (Wang et al., 2023a), PUe (Wang et al., 2023b), and GLWS (Chen et al., 2024). We evaluated our methods on two image datasets (CIFAR-10 (Krizhevsky & Hinton, 2009) and ImageNette (Deng et al., 2009)) and two UCI datasets (USPS and Letter) from the UCI Machine Learning Repository (Kelly et al., 2023). ImageNette is a curated subset of the larger ImageNet corpus, containing ten easily distinguishable categories: *tench, English springer, cassette player, chain saw, church, French horn, garbage truck, gas pump, golf ball, and parachute*. We synthesized PU versions of these datasets; detailed information can be found in Appendix C. We used ResNet-34 (He et al., 2016) for image datasets and a multilayer perceptron (MLP) with a hidden layer width of 500 equipped with the ReLU (Nair & Hinton, 2010) activation function for tabular datasets.

Following the widely used validation protocol (Raschka, 2018; Gulrajani & Lopez-Paz, 2021; Wang et al., 2025), we divided some training data from the positive and unlabeled datasets into the positive validation set $D'_P$ and the unlabeled validation set $D'_U$, respectively. We used various test metrics, including accuracy, AUC score, F1 score, precision, and recall. We first trained a model with training sets $D_P$ and $D_U$. Then, we evaluated its validation performance based on the metrics in Section 3 as well as its test performance on a test set with true labels. We randomly selected a set of hyperparameter configurations from a given pool. For each validation metric, we selected the checkpoint with the best validation performance on $D'_P \bigcup D'_U$, and recorded the corresponding test metrics. We recorded the mean test metrics and standard deviations obtained with different data splits.

Table 2: Test results (mean±std) of accuracy, AUC, and F1 score for each algorithm on CIFAR-10 (Case 2) under different model selection criteria. The best performance w.r.t. each validation metric is shown in bold.

| Test metric | Test ACC | | | AUC | | | Test F1 | | |
|---|---|---|---|---|---|---|---|---|---|
| Val metric | PA | PAUC | OA | PA | PAUC | OA | PA | PAUC | OA |
| PUbN | 78.26±1.01 | **79.50±0.38** | 79.94±0.36 | 87.81±0.65 | 88.00±0.38 | 88.08±0.45 | 80.47±0.65 | 79.17±0.88 | 79.91±0.21 |
| PAN | 61.43±2.74 | 60.61±4.34 | 63.48±2.71 | 68.71±5.63 | 71.54±4.68 | 69.63±5.43 | 70.73±1.40 | 70.87±1.72 | 69.25±3.04 |
| CVIR | 78.49±1.49 | **79.50±1.46** | **80.44±0.68** | **88.10±0.87** | 87.98±1.33 | **88.68±0.81** | **80.86±0.97** | **80.69±1.33** | **81.44±0.58** |
| P3MIX-E | 59.04±4.54 | 50.00±0.00 | 59.13±4.62 | 74.26±4.26 | 84.52±0.84 | 74.11±4.16 | 70.45±2.00 | 44.44±18.14 | 70.45±2.00 |
| P3MIX-C | 78.05±0.95 | 77.42±1.40 | 78.70±0.50 | 85.87±1.02 | 84.92±1.40 | 86.13±0.79 | 79.82±0.56 | 79.06±0.92 | 79.90±0.49 |
| LBE | 72.47±1.50 | 63.54±2.86 | 75.96±0.88 | 84.02±0.40 | 84.26±0.78 | 83.47±0.97 | 77.13±0.72 | 72.96±1.42 | 76.04±0.83 |
| Count Loss | 74.44±0.68 | 74.75±0.45 | 76.87±0.75 | 82.88±1.02 | 82.99±1.03 | 84.44±0.75 | 77.41±0.54 | 76.70±0.55 | 78.27±0.99 |
| Robust-PU | 78.94±0.79 | 78.43±0.61 | 79.60±0.81 | 85.23±1.09 | 87.13±0.76 | 86.33±0.63 | 80.37±0.72 | 77.16±0.68 | 79.79±0.89 |
| Holistic-PU | 55.60±0.16 | 56.04±4.93 | 71.18±1.20 | 78.03±2.53 | 67.96±6.67 | 76.93±3.13 | 69.02±0.04 | 44.49±18.12 | 73.64±2.09 |
| PUe | 68.60±0.41 | 67.40±1.90 | 71.05±0.52 | 78.06±0.31 | 79.27±0.51 | 78.69±0.36 | 73.41±0.44 | 73.05±0.71 | 71.06±1.35 |
| GLWS | 77.71±0.71 | 76.22±1.33 | 79.58±0.61 | 87.86±0.33 | **88.08±0.43** | 87.44±0.51 | 80.40±0.37 | 79.75±0.81 | 80.47±0.47 |
| uPU | 66.21±1.40 | 69.03±1.04 | 70.46±0.70 | 76.46±1.65 | 78.80±0.74 | 77.97±0.90 | 71.52±0.73 | 72.78±0.47 | 70.89±1.53 |
| uPU-c | 77.22±0.26 | 79.29±0.37 | 79.02±0.99 | 85.19±0.46 | 87.76±0.38 | 87.11±0.83 | 79.48±0.22 | 78.19±0.45 | 78.60±1.22 |
| nnPU | 74.27±1.26 | 62.67±1.09 | 77.62±0.68 | 86.16±0.07 | 86.53±0.16 | 86.42±0.58 | 78.00±0.55 | 72.57±0.51 | 79.20±0.52 |
| nnPU-c | 77.74±0.53 | 78.49±0.35 | 79.37±0.30 | 84.84±0.44 | 86.63±0.31 | 86.16±0.22 | 79.79±0.18 | 77.25±0.61 | 79.07±0.39 |
| nnPU-GA | 76.59±1.15 | 76.73±0.88 | 78.38±0.74 | 86.41±1.24 | 86.09±1.23 | 86.58±0.84 | 79.14±0.95 | 78.76±1.11 | 78.22±0.53 |
| nnPU-GA-c | 78.00±0.52 | 78.32±0.71 | 79.12±0.90 | 85.32±0.71 | 85.82±1.04 | 85.63±1.27 | 79.26±0.81 | 77.78±0.48 | 79.03±0.92 |
| PUSB | 75.74±0.61 | 78.80±0.55 | 78.35±0.41 | 75.74±0.61 | 78.80±0.55 | 78.35±0.41 | 79.18±0.43 | 79.83±0.59 | 79.79±0.61 |
| PUSB-c | **79.06±0.45** | 77.98±0.54 | 79.19±0.32 | 79.06±0.45 | 77.98±0.54 | 79.19±0.32 | 80.06±0.36 | 77.43±0.40 | 79.29±0.40 |
| VPU | 76.99±1.00 | 63.22±5.30 | 77.31±0.86 | 85.47±0.98 | 87.08±0.43 | 86.07±0.67 | 75.15±1.31 | 39.92±15.74 | 75.43±1.33 |
| VPU-c | 77.70±0.41 | 78.20±0.90 | 79.81±0.66 | 86.90±0.39 | 87.50±0.46 | 86.32±0.28 | 80.12±0.27 | 80.52±0.53 | 80.56±0.71 |
| Dist-PU | 73.46±0.59 | 74.83±0.58 | 74.69±0.60 | 80.70±0.45 | 82.09±0.40 | 81.48±0.78 | 76.90±0.31 | 76.88±0.16 | 76.65±0.15 |
| Dist-PU-c | 72.57±3.47 | 74.41±2.67 | 74.30±2.73 | 80.34±3.48 | 82.49±2.68 | 81.94±2.90 | 75.50±2.34 | 75.27±2.67 | 73.68±3.25 |

Table 3: Test results (mean±std) of accuracy, AUC, and F1 score for each algorithm on ImageNette (Case 1) under different model selection criteria. The best performance w.r.t. each validation metric is shown in bold.

| Test metric | Test ACC | | | AUC | | | Test F1 | | |
|---|---|---|---|---|---|---|---|---|---|
| Val metric | PA | PAUC | OA | PA | PAUC | OA | PA | PAUC | OA |
| PUbN | 75.69±0.02 | 77.07±0.47 | 78.99±0.57 | 84.82±0.28 | 86.25±0.83 | 87.45±0.37 | 77.63±0.16 | 76.30±1.26 | 79.25±0.76 |
| PAN | 50.74±1.17 | 51.52±0.46 | 56.93±1.83 | 53.71±3.39 | 55.48±1.03 | 55.73±1.79 | 65.24±0.19 | 32.31±14.81 | 45.43±2.65 |
| CVIR | 78.78±0.86 | 78.26±1.62 | **81.01±0.67** | **87.98±0.35** | **88.29±0.65** | **89.28±0.38** | 80.12±0.36 | 79.52±0.78 | **81.51±0.45** |
| P3MIX-E | 74.81±2.36 | 49.71±0.48 | 75.19±2.39 | 82.71±2.74 | 85.84±0.68 | 82.91±2.92 | 76.23±1.97 | 43.92±17.93 | 76.41±2.04 |
| P3MIX-C | **78.81±1.61** | **78.91±1.84** | 80.25±0.82 | 85.85±1.50 | 86.41±1.31 | 87.33±1.27 | **80.26±1.24** | **80.35±1.27** | 80.48±0.37 |
| LBE | 78.52±0.41 | 78.73±0.65 | 79.20±0.36 | 86.84±0.37 | 86.31±0.61 | 86.16±0.78 | 78.90±0.32 | 77.09±1.48 | 78.14±0.75 |
| Count Loss | 74.98±0.85 | 75.95±1.56 | 78.07±0.73 | 85.50±0.23 | 85.44±0.52 | 85.75±0.74 | 77.84±0.40 | 77.95±0.87 | 78.92±0.91 |
| Robust-PU | 77.67±0.27 | 75.53±2.04 | 78.73±0.43 | 83.93±0.64 | 85.22±0.11 | 84.46±0.93 | 77.86±0.47 | 71.78±4.44 | 78.06±0.59 |
| Holistic-PU | 51.16±0.47 | 54.42±3.66 | 53.62±0.24 | 58.85±1.01 | 56.45±6.07 | 55.25±0.43 | 65.18±0.31 | 64.23±1.18 | 51.58±1.27 |
| PUe | 67.47±1.88 | 71.46±1.27 | 70.90±1.28 | 75.35±1.52 | 77.29±1.49 | 77.47±1.55 | 70.39±0.48 | 70.97±1.81 | 71.46±1.56 |
| GLWS | 76.14±0.86 | 74.96±1.62 | 78.68±0.70 | 87.00±0.40 | 86.89±0.71 | 86.96±0.74 | 78.93±0.45 | 78.52±1.01 | 79.56±0.67 |
| uPU | 71.07±0.95 | 64.14±6.15 | 73.69±0.74 | 82.24±0.61 | 81.60±1.06 | 81.94±0.41 | 74.88±0.50 | 71.58±2.45 | 74.95±0.78 |
| uPU-c | 75.00±0.97 | 72.54±4.40 | 77.76±0.66 | 84.13±0.33 | 85.82±0.55 | 84.65±0.65 | 77.16±0.27 | 63.33±9.88 | 77.19±0.67 |
| nnPU | 75.63±1.34 | 66.81±1.09 | 77.80±0.74 | 86.56±0.38 | 86.12±0.71 | 86.72±0.26 | 78.52±0.77 | 73.97±0.57 | 78.19±0.46 |
| nnPU-c | 76.51±0.61 | 76.95±0.75 | 77.66±0.63 | 83.87±0.71 | 85.08±0.67 | 83.93±1.24 | 77.89±0.33 | 74.59±1.65 | 77.33±0.99 |
| nnPU-GA | 75.70±0.36 | 78.72±0.64 | 79.40±0.47 | 83.74±0.65 | 86.06±0.88 | 84.45±1.58 | 78.33±0.16 | 78.98±1.22 | 79.13±0.24 |
| nnPU-GA-c | 77.65±0.58 | 72.91±2.33 | 78.56±0.06 | 81.45±1.32 | 84.75±0.53 | 82.69±1.21 | 77.88±0.51 | 65.42±5.22 | 78.34±0.42 |
| PUSB | 72.73±0.54 | 77.03±0.74 | 76.73±0.35 | 73.10±0.53 | 77.19±0.68 | 76.91±0.33 | 77.26±0.33 | 78.65±0.09 | 78.66±0.16 |
| PUSB-c | 76.37±0.16 | 77.36±0.36 | 77.81±0.60 | 76.48±0.15 | 77.31±0.33 | 77.86±0.60 | 77.37±0.17 | 76.42±0.04 | 78.15±0.80 |
| VPU | 56.36±2.98 | 50.91±0.03 | 61.72±0.41 | 61.21±2.22 | 82.35±0.27 | 73.84±4.69 | 53.86±6.31 | 0.14±0.11 | 45.88±4.26 |
| VPU-c | 77.48±0.83 | 78.00±0.50 | 78.06±0.91 | 83.35±0.33 | 84.63±0.49 | 84.28±0.96 | 78.09±0.69 | 78.60±0.40 | 77.64±0.69 |
| Dist-PU | 70.40±2.37 | 71.86±2.34 | 74.68±0.79 | 83.97±1.16 | 83.18±1.51 | 83.92±0.73 | 75.58±1.50 | 75.76±1.61 | 77.00±0.75 |
| Dist-PU-c | 72.03±0.99 | 65.88±3.33 | 73.84±1.06 | 79.51±0.44 | 77.83±0.61 | 80.91±1.18 | 74.78±0.58 | 52.05±10.16 | 74.10±0.81 |

## 5.2 BENCHMARK RESULTS

Tables 1 to 4, and 7 to 18 in Appendix D report detailed experimental results in terms of different metrics on CIFAR-10, ImageNette, Letter, and USPS, and the hyperparameters are determined with PA, PAUC, and OA, respectively. In addition, Figures 4 to 7 show the overall performance of different algorithms. For ease of presentation in figures, we did not include the algorithms where the performance is obviously inferior. We can draw the following conclusions based on the experimental results: 1) The TS algorithms without calibration perform worse due to the ILS problem, indicating the existence of an evaluation pitfall in the literature. The proposed calibration technique consistently improves the classification performance for TS approaches, demonstrating the effectiveness of the proposed calibration technique. 2) There is no algorithm that can win in every case of the dataset and evaluation metric. Besides, some early algorithms can already achieve satisfactory classification performance. 3) Our proposed validation metrics are effective in hyperparameter selection. However, the effectiveness may also depend on the test metric. For example, we can observe from Table 1 that the model selected using PAUC can achieve better performance than using OA when the test metric is the AUC score.

Table 4: Test results (mean±std) of accuracy, AUC, and F1 score for each algorithm on Ima-geNette (Case 2) under different model selection criteria. The best performance w.r.t. each validation metric is shown in bold.

| Test metric | Test ACC | | | AUC | | | Test F1 | | |
|---|---|---|---|---|---|---|---|---|---|
| Val metric | PA | PAUC | OA | PA | PAUC | OA | PA | PAUC | OA |
| PUbN | 75.30±0.58 | 75.97±0.61 | 77.39±0.45 | 83.97±0.64 | 84.03±0.50 | 85.20±0.66 | 76.89±0.51 | 75.98±1.24 | 76.44±0.60 |
| PAN | 53.31±0.36 | 64.73±1.69 | 64.37±2.18 | 65.28±1.32 | 70.38±1.92 | 69.19±2.61 | 66.49±0.27 | 58.68±5.06 | 63.03±2.55 |
| CVIR | **76.60±0.74** | **77.87±0.67** | **79.29±0.47** | **85.84±0.26** | 86.25±0.32 | **87.22±0.49** | 78.43±0.48 | **78.67±0.35** | **79.39±0.09** |
| P3MIX-E | 60.42±4.27 | 49.86±0.23 | 60.82±4.16 | 70.79±2.16 | 81.61±0.76 | 71.51±2.47 | 67.11±1.54 | 44.19±18.04 | 67.38±1.42 |
| P3MIX-C | 74.17±0.90 | 75.40±0.81 | 75.35±0.81 | 83.92±0.88 | 83.97±1.13 | 83.68±0.43 | 76.85±0.73 | 77.21±0.65 | 77.13±0.44 |
| LBE | 74.51±0.94 | 74.67±0.54 | 76.31±0.92 | 83.06±1.01 | 82.33±0.45 | 83.33±0.99 | 76.85±0.74 | 73.81±1.63 | 74.99±1.41 |
| Count Loss | 73.27±0.28 | 73.62±0.23 | 74.43±0.66 | 82.20±0.51 | 82.04±0.66 | 82.05±0.72 | 76.28±0.22 | 76.11±0.30 | 76.46±0.45 |
| Robust-PU | 72.58±1.19 | 72.78±0.43 | 75.52±0.68 | 80.19±0.81 | 80.57±0.71 | 81.69±0.38 | 73.75±0.62 | 69.94±1.57 | 74.06±1.05 |
| Holistic-PU | 56.12±0.93 | 54.70±2.16 | 59.19±0.53 | 61.46±0.21 | 59.83±1.01 | 62.22±0.69 | 64.75±1.38 | 60.81±1.16 | 58.83±0.57 |
| PUe | 64.65±0.59 | 65.89±1.36 | 67.63±0.53 | 72.74±1.48 | 72.62±1.26 | 74.27±0.79 | 69.33±1.07 | 68.66±0.71 | 69.42±0.90 |
| GLWS | 75.61±0.65 | 75.38±0.24 | 76.99±0.21 | 85.81±0.55 | **86.55±0.37** | 85.77±0.29 | **78.47±0.34** | 78.40±0.13 | 78.65±0.19 |
| uPU | 60.42±2.82 | 66.42±1.08 | 66.29±1.00 | 67.49±3.09 | 73.24±0.86 | 72.82±0.52 | 67.95±0.68 | 67.50±1.57 | 66.46±1.37 |
| uPU-c | 72.57±1.56 | 73.20±1.05 | 75.07±0.54 | 79.19±2.25 | 81.76±0.92 | 82.22±0.87 | 72.60±1.24 | 69.52±2.07 | 74.58±0.79 |
| nnPU | 69.83±0.52 | 55.99±3.35 | 72.76±0.55 | 82.83±1.26 | 80.53±0.83 | 82.65±0.76 | 74.92±0.35 | 69.09±1.48 | 75.65±0.45 |
| nnPU-c | 74.42±0.75 | 73.55±1.07 | 74.17±1.07 | 82.13±0.89 | 82.44±0.97 | 81.77±1.05 | 75.24±1.25 | 71.49±2.66 | 73.68±1.79 |
| nnPU-GA | 72.19±1.31 | 75.23±1.08 | 75.87±0.60 | 81.37±0.66 | 82.62±1.08 | 83.37±0.99 | 75.44±0.43 | 74.24±2.14 | 74.85±0.80 |
| nnPU-GA-c | 72.27±1.25 | 73.85±0.58 | 74.62±0.11 | 79.16±1.52 | 81.60±0.47 | 79.72±1.52 | 73.86±0.71 | 71.02±0.68 | 74.79±0.81 |
| PUSB | 71.29±1.80 | 72.57±0.76 | 75.30±0.56 | 71.44±1.78 | 72.65±0.76 | 75.35±0.54 | 75.76±0.85 | 74.73±0.74 | 76.77±0.29 |
| PUSB-c | 72.98±1.11 | 73.69±0.38 | 74.86±0.52 | 73.00±1.10 | 73.64±0.38 | 74.86±0.50 | 73.69±0.94 | 71.85±0.48 | 74.70±0.37 |
| VPU | 70.42±1.87 | 58.37±6.43 | 73.28±0.73 | 78.68±1.11 | 78.52±1.50 | 80.21±0.47 | 73.29±0.72 | 24.24±19.50 | 70.20±1.70 |
| VPU-c | 76.30±0.79 | 77.75±0.57 | 77.38±1.00 | 84.37±0.45 | 84.11±0.49 | 83.80±0.79 | 78.34±0.83 | 78.32±0.40 | 77.88±0.75 |
| Dist-PU | 63.97±1.03 | 67.74±0.50 | 68.58±1.04 | 72.64±0.26 | 75.26±0.57 | 74.62±0.96 | 69.88±0.13 | 71.92±0.37 | 70.92±0.75 |
| Dist-PU-c | 60.43±4.37 | 68.02±1.27 | 67.29±1.74 | 67.51±3.90 | 74.10±1.29 | 71.97±2.69 | 67.65±0.89 | 69.06±1.11 | 65.39±3.35 |

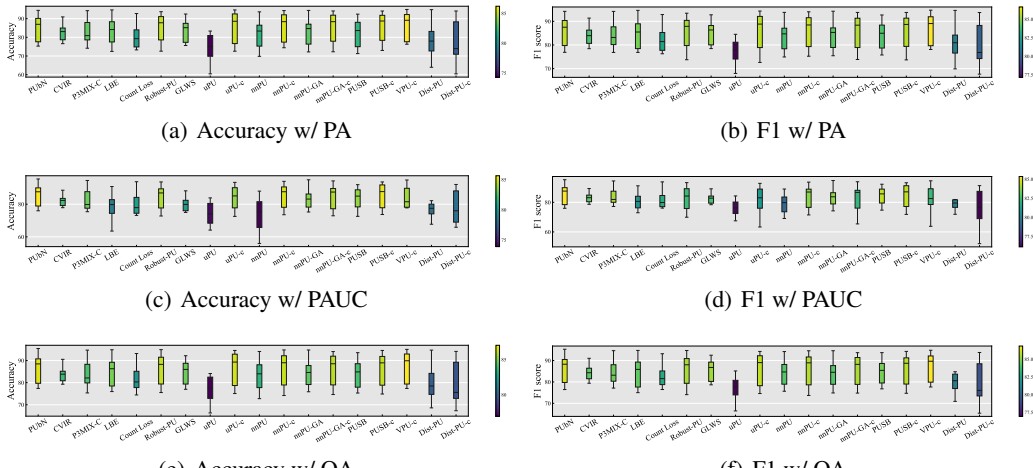

(a) Accuracy w/ PA          (b) F1 w/ PA

(c) Accuracy w/ PAUC          (d) F1 w/ PAUC

(e) Accuracy w/ OA          (f) F1 w/ OA

Figure 4: Overall performance w.r.t. accuracy and the F1 score across all datasets. Hyperparameters were tuned using PA, PAUC and OA, respectively; bar colors indicate means.

## 6 CONCLUSION

In this paper, we conducted a comprehensive empirical study of PU learning algorithms. To the best of our knowledge, we proposed the first PU learning benchmark to systematically compare different PU learning algorithms in a unified framework. We investigated model selection criteria to facilitate realistic evaluation of PU learning algorithms. We also identified the ILS problem for the one-sample setting of PU learning and proposed a calibration approach to ensure fair comparisons of different families of PU learning algorithms. We hope that our framework can facilitate accessible, realistic, and fair evaluation of PU learning algorithms in the future. A limitation of our work is that we use relatively small benchmark datasets following previous work. In the future, it is also promising to investigate the performance of different algorithms on collected large-scale PU benchmark datasets.

ACKNOWLEDGMENTS

WW and DDW were supported by the Junior Research Associate (JRA) program of RIKEN. MS was supported by JST ASPIRE Grant Number JPMJAP2405.

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

THE USE OF LARGE LANGUAGE MODELS (LLMS)

LLMs were used solely for correcting grammatical and spelling errors.

## A  COMPARISON WITH A CONCURRENT WORK

A concurrent work (Chen et al., 2026) also proposed a PU learning benchmark. The authors primarily focused on establishing a fair comparison and evaluation pipeline for PU learning algorithms. They noted the differences between one-sample and two-sample settings, yet they did not analyze the ILS problem or propose solutions. Furthermore, they assumed that the validation set had true labels, which is unrealistic, as discussed in the paper.

## B  PROOFS

### B.1  PROOF OF PROPOSITION 1

$$
\begin{aligned}
\text{ACC}(f) =& \pi\mathbb{E}_{p(\boldsymbol{x}|y=+1)}\left[\mathbb{I}\left(f(\boldsymbol{x})\geqslant 0\right)\right] + (1-\pi)\mathbb{E}_{p(\boldsymbol{x}|y=-1)}\left[\mathbb{I}\left(f(\boldsymbol{x})< 0\right)\right] \\
=& \pi\mathbb{E}_{p(\boldsymbol{x}|y=+1)}\left[\mathbb{I}\left(f(\boldsymbol{x})\geqslant 0\right)\right] + \mathbb{E}_{p(\boldsymbol{x})}\left[\mathbb{I}\left(f(\boldsymbol{x})< 0\right)\right] - \pi\mathbb{E}_{p(\boldsymbol{x}|y=+1)}\left[\mathbb{I}\left(f(\boldsymbol{x})< 0\right)\right] \\
=& \pi\mathbb{E}_{p(\boldsymbol{x}|y=+1)}\left[\mathbb{I}\left(f(\boldsymbol{x})\geqslant 0\right)\right] + \mathbb{E}_{p(\boldsymbol{x})}\left[\mathbb{I}\left(f(\boldsymbol{x})< 0\right)\right] - \pi\mathbb{E}_{p(\boldsymbol{x}|y=+1)}\left[1 - \mathbb{I}\left(f(\boldsymbol{x})\geqslant 0\right)\right] \\
=& 2\pi\mathbb{E}_{p(\boldsymbol{x}|y=+1)}\left[\mathbb{I}\left(f(\boldsymbol{x})\geqslant 0\right)\right] + \mathbb{E}_{p(\boldsymbol{x})}\left[\mathbb{I}\left(f(\boldsymbol{x})< 0\right)\right] - \pi \\
=& \mathbb{E}\left[\text{PA}(f)\right] - \pi.
\end{aligned}
$$

Here, the last equation is obtained since $\mathcal{D}_{\mathrm{U}} \overset{\text{i.i.d.}}{\sim} p(\boldsymbol{x})$ for the TS setting and $\mathcal{D}_{\mathrm{P}} \bigcup \mathcal{D}_{\mathrm{U}} \overset{\text{i.i.d.}}{\sim} p(\boldsymbol{x})$ for the OS setting. Therefore, for two classifiers $f_1$ and $f_2$ that satisfy $\mathbb{E}\left[\text{PA}(f_1)\right] < \mathbb{E}\left[\text{PA}(f_2)\right]$, we have $\text{ACC}(f_1) < \text{ACC}(f_2)$. The proof is complete. $\qquad\square$

### B.2  PROOF OF PROPOSITION 2

For the TS setting,

$$
\begin{aligned}
\text{AUC}(f) =& \mathbb{E}_{p(\boldsymbol{x}|y=+1)}\mathbb{E}_{p(\boldsymbol{x}'|y'=-1)}\left[\mathbb{I}\left(f(\boldsymbol{x})> f(\boldsymbol{x}')\right) + \frac{1}{2}\mathbb{I}\left(f(\boldsymbol{x})= f(\boldsymbol{x}')\right)\right] \\
=& \frac{1}{1-\pi}\mathbb{E}_{p(\boldsymbol{x}|y=+1)}\mathbb{E}_{p(\boldsymbol{x}')}\left[\mathbb{I}\left(f(\boldsymbol{x})> f(\boldsymbol{x}')\right) + \frac{1}{2}\mathbb{I}\left(f(\boldsymbol{x})= f(\boldsymbol{x}')\right)\right] \\
& - \frac{\pi}{1-\pi}\mathbb{E}_{p(\boldsymbol{x}|y=+1)}\mathbb{E}_{p(\boldsymbol{x}'|y'=+1)}\left[\mathbb{I}\left(f(\boldsymbol{x})> f(\boldsymbol{x}')\right) + \frac{1}{2}\mathbb{I}\left(f(\boldsymbol{x})= f(\boldsymbol{x}')\right)\right] \\
=& \frac{1}{1-\pi}\mathbb{E}_{p(\boldsymbol{x}|y=+1)}\mathbb{E}_{p(\boldsymbol{x}')}\left[\mathbb{I}\left(f(\boldsymbol{x})> f(\boldsymbol{x}')\right) + \frac{1}{2}\mathbb{I}\left(f(\boldsymbol{x})= f(\boldsymbol{x}')\right)\right] - \frac{\pi}{2-2\pi} \\
=& \frac{1}{1-\pi}\mathbb{E}\left[\text{PAUC}(f)\right] - \frac{\pi}{2-2\pi}.
\end{aligned}
$$

For the OS setting,

$$
\begin{aligned}
\text{AUC}(f) =& \mathbb{E}_{p(\boldsymbol{x}|y=+1)}\mathbb{E}_{p(\boldsymbol{x}'|y'=-1)}\left[\mathbb{I}\left(f(\boldsymbol{x})> f(\boldsymbol{x}')\right) + \frac{1}{2}\mathbb{I}\left(f(\boldsymbol{x})= f(\boldsymbol{x}')\right)\right] \\
=& \frac{1}{1-\overline{\pi}}\mathbb{E}_{p(\boldsymbol{x}|y=+1)}\mathbb{E}_{p(\boldsymbol{x}')}\left[\mathbb{I}\left(f(\boldsymbol{x})> f(\boldsymbol{x}')\right) + \frac{1}{2}\mathbb{I}\left(f(\boldsymbol{x})= f(\boldsymbol{x}')\right)\right] \\
& - \frac{\overline{\pi}}{1-\overline{\pi}}\mathbb{E}_{p(\boldsymbol{x}|y=+1)}\mathbb{E}_{p(\boldsymbol{x}'|y'=+1)}\left[\mathbb{I}\left(f(\boldsymbol{x})> f(\boldsymbol{x}')\right) + \frac{1}{2}\mathbb{I}\left(f(\boldsymbol{x})= f(\boldsymbol{x}')\right)\right] \\
=& \frac{1}{1-\overline{\pi}}\mathbb{E}_{p(\boldsymbol{x}|y=+1)}\mathbb{E}_{p(\boldsymbol{x}')}\left[\mathbb{I}\left(f(\boldsymbol{x})> f(\boldsymbol{x}')\right) + \frac{1}{2}\mathbb{I}\left(f(\boldsymbol{x})= f(\boldsymbol{x}')\right)\right] - \frac{\overline{\pi}}{2-2\overline{\pi}} \\
=& \frac{1}{1-\overline{\pi}}\mathbb{E}\left[\text{PAUC}(f)\right] - \frac{\overline{\pi}}{2-2\overline{\pi}}.
\end{aligned}
$$

Therefore, under both OS and TS settings, for two classifiers $f_1$ and $f_2$ that satisfy $\mathbb{E}\left[\text{PAUC}(f_1)\right] < \mathbb{E}\left[\text{PAUC}(f_2)\right]$, we have $\text{AUC}(f_1) < \text{AUC}(f_2)$. $\qquad\square$

### B.3 Bias of the Risk Estimator

Under the OS setting, we have

$$
\begin{aligned}
\mathbb{E}\left[\widehat{R}(f)\right] - R(f) =& \mathbb{E}_{\bar{p}(\boldsymbol{x})}\left[\ell(f(\boldsymbol{x}), -1)\right] - \mathbb{E}_{p(\boldsymbol{x})}\left[\ell(f(\boldsymbol{x}), -1)\right] \\
=& (\bar{\pi} - \pi)\left(\mathbb{E}_{p(\boldsymbol{x}|y=+1)}\left[\ell(f(\boldsymbol{x}), -1)\right] - \mathbb{E}_{p(\boldsymbol{x}|y=-1)}\left[\ell(f(\boldsymbol{x}), -1)\right]\right),
\end{aligned}
$$

which is not equal to 0. Therefore, it means that the bias of the risk estimator always exists. Then, the minimizers of $\mathbb{E}\left[\widehat{R}(f)\right]$ and $R(f)$ are not the same.

### B.4 Proof of Theorem 1

First, we have

$$
\begin{aligned}
\bar{p}(\boldsymbol{x}) =& \bar{\pi}p(\boldsymbol{x}|y = +1) + (1 - \bar{\pi})p(\boldsymbol{x}|y = -1) \\
=& \frac{(1-c)\pi}{1 - c\pi}p(\boldsymbol{x}|y = +1) + \frac{1-\pi}{1 - c\pi}p(\boldsymbol{x}|y = -1).
\end{aligned}
$$

Therefore, we have

$$
p(\boldsymbol{x}|y = -1) = \frac{1 - c\pi}{1 - \pi}\bar{p}(\boldsymbol{x}) - \frac{(1-c)\pi}{1 - \pi}p(\boldsymbol{x}|y = +1).
$$

Then,

$$
\begin{aligned}
R(f) =& \pi\mathbb{E}_{p(\boldsymbol{x}|y=+1)}\left[\ell(f(\boldsymbol{x}), +1)\right] + (1 - \pi)\mathbb{E}_{p(\boldsymbol{x}|y=-1)}\left[\ell(f(\boldsymbol{x}), -1)\right] \\
=& \pi\mathbb{E}_{p(\boldsymbol{x}|y=+1)}\left[\ell(f(\boldsymbol{x}), +1)\right] + (1 - c\pi)\mathbb{E}_{\bar{p}(\boldsymbol{x})}\left[\ell(f(\boldsymbol{x}), -1)\right] - (1 - c)\pi\mathbb{E}_{p(\boldsymbol{x}|y=+1)}\left[\ell(f(\boldsymbol{x}), -1)\right] \\
=& \pi\mathbb{E}_{p(\boldsymbol{x}|y=+1)}\left[\ell(f(\boldsymbol{x}), +1) + (c - 1)\ell(f(\boldsymbol{x}), -1)\right] + (1 - c\pi)\mathbb{E}_{\bar{p}(\boldsymbol{x})}\left[\ell(f(\boldsymbol{x}), -1)\right],
\end{aligned}
$$

which concludes the proof. $\qquad\square$

### B.5 Proof of Theorem 2

**Definition 5** (Rademacher complexity)**.** Let $\mathcal{X}_{n_{\text{P}}}^{\text{P}} = \{\boldsymbol{x}_1, \cdots \boldsymbol{x}_{n_{\text{P}}}\}$ denote $n_{\text{P}}$ i.i.d. random variables drawn from density $p(\boldsymbol{x}|y = +1)$. Let $\mathcal{X}_{n_{\text{U}}}^{\text{U}} = \{\boldsymbol{x}_{n_{\text{P}}+1}, \cdots \boldsymbol{x}_{n_{\text{P}}+n_{\text{U}}}\}$ denote $n_{\text{U}}$ i.i.d. random variables drawn from density $\bar{p}(\boldsymbol{x})$. Let $\mathcal{F} = \{f : \mathcal{X} \mapsto \mathbb{R}\}$ denote a class of measurable functions, $\boldsymbol{\sigma}_{\text{P}} = (\sigma_1, \sigma_2, \cdots, \sigma_{n_{\text{P}}})$, and $\boldsymbol{\sigma}_{\text{U}} = (\sigma_{n_{\text{P}}+1}, \sigma_{n_{\text{P}}+2}, \cdots, \sigma_{n_{\text{P}}+n_{\text{U}}})$ denote Rademacher variables taking values from $\{+1, -1\}$ uniformly. Then, the (expected) Rademacher complexities of $\mathcal{F}$ are defined as

$$
\mathfrak{R}_{n_{\text{P}}}(\mathcal{F}) = \mathbb{E}_{\mathcal{X}_{n_{\text{P}}}^{\text{P}}}\mathbb{E}_{\boldsymbol{\sigma}_{\text{P}}}\left[\sup_{f \in \mathcal{F}} \frac{1}{n_{\text{P}}}\sum_{i=1}^{n_{\text{P}}}\sigma_i f(\boldsymbol{x}_i)\right],
$$

$$
\mathfrak{R}'_{n_{\text{U}}}(\mathcal{F}) = \mathbb{E}_{\mathcal{X}_{n_{\text{U}}}^{\text{U}}}\mathbb{E}_{\boldsymbol{\sigma}_{\text{U}}}\left[\sup_{f \in \mathcal{F}} \frac{1}{n_{\text{U}}}\sum_{i=n_{\text{P}}+1}^{n_{\text{P}}+n_{\text{U}}}\sigma_i f(\boldsymbol{x}_i)\right].
$$

**Lemma 1.** *For any $\delta > 0$, we have the following inequality with probability at least $1 - \delta$:*

$$
\sup_{f \in \mathcal{F}}\left|\bar{R}(f) - R(f)\right| \leqslant 2(2 - c)\pi L_\ell \mathfrak{R}_{n_{\text{P}}}(\mathcal{F}) + 2(1 - c\pi)L_\ell \mathfrak{R}'_{n_{\text{U}}}(\mathcal{F})
$$

$$
+ \left(\frac{\pi(2 - c)C_\ell}{\sqrt{n_{\text{P}}}} + \frac{(1 - c\pi)C_\ell}{\sqrt{n_{\text{U}}}}\right)\sqrt{\frac{\ln 2/\delta}{2}}.
$$

*Proof.* First, we give the upper bound for the one-side uniform deviation $\sup_{f \in \mathcal{F}}\left(\bar{R}(f) - R(f)\right)$. When an instance in $\mathcal{X}_{n_{\text{P}}}^{\text{P}}$ is replaced by another instance, the value of $\sup_{f \in \mathcal{F}}\left(\bar{R}(f) - R(f)\right)$

changes at most $\pi(2-c)C_\ell/n_{\mathrm{P}}$; when an instance in $\mathcal{X}_{n_{\mathrm{U}}}^{\mathrm{U}}$ is replaced by another instance, the value of $\sup_{f\in\mathcal{F}}\left(\bar{R}(f) - R(f)\right)$ changes at most $(1-c\pi)C_\ell/n_{\mathrm{U}}$. Therefore, according to McDiarmid's inequality, we have the following inequality with probability at least $1 - \delta/2$:

$$
\begin{aligned}
\sup_{f\in\mathcal{F}}\left(\bar{R}(f) - R(f)\right) &\leqslant \mathbb{E}\left[\sup_{f\in\mathcal{F}}\left(\bar{R}(f) - R(f)\right)\right] + \sqrt{\frac{\pi^2(2-c)^2C_\ell^2}{n_{\mathrm{P}}} + \frac{(1-c\pi)^2C_\ell^2}{n_{\mathrm{U}}}}\sqrt{\frac{\ln 2/\delta}{2}} \\
&\leqslant \mathbb{E}\left[\sup_{f\in\mathcal{F}}\left(\bar{R}(f) - R(f)\right)\right] + \left(\frac{\pi(2-c)C_\ell}{\sqrt{n_{\mathrm{P}}}} + \frac{(1-c\pi)C_\ell}{\sqrt{n_{\mathrm{U}}}}\right)\sqrt{\frac{\ln 2/\delta}{2}}.
\end{aligned}
$$

Then, by symmetrization (Vapnik, 1998), it is a routine work to have

$$
\mathbb{E}\left[\sup_{f\in\mathcal{F}}\left(\bar{R}(f) - R(f)\right)\right] \leqslant 2(2-c)\pi\mathfrak{R}_{n_{\mathrm{P}}}(\ell\circ\mathcal{F}) + 2(1-c\pi)\mathfrak{R}'_{n_{\mathrm{U}}}(\ell\circ\mathcal{F}).
$$

According to Talagrand's contraction lemma (Shalev-Shwartz & Ben-David, 2014), we have

$$
\mathfrak{R}_{n_{\mathrm{P}}}(\ell\circ\mathcal{F}) \leqslant L_\ell\mathfrak{R}_{n_{\mathrm{P}}}(\mathcal{F}), \quad \mathfrak{R}'_{n_{\mathrm{U}}}(\ell\circ\mathcal{F}) \leqslant L_\ell\mathfrak{R}'_{n_{\mathrm{U}}}(\mathcal{F}).
$$

By combining the above inequalities, we have the following inequality with probability at least $1 - \delta/2$:

$$
\begin{aligned}
\sup_{f\in\mathcal{F}}\left(\bar{R}(f) - R(f)\right) \leqslant &\, 2(2-c)\pi L_\ell\mathfrak{R}_{n_{\mathrm{P}}}(\mathcal{F}) + 2(1-c\pi)L_\ell\mathfrak{R}'_{n_{\mathrm{U}}}(\mathcal{F}) \\
&+ \left(\frac{\pi(2-c)C_\ell}{\sqrt{n_{\mathrm{P}}}} + \frac{(1-c\pi)C_\ell}{\sqrt{n_{\mathrm{U}}}}\right)\sqrt{\frac{\ln 2/\delta}{2}}.
\end{aligned}
$$

In a similar way, we have the following inequality with probability at least $1 - \delta/2$:

$$
\begin{aligned}
\sup_{f\in\mathcal{F}}\left(R(f) - \bar{R}(f)\right) \leqslant &\, 2(2-c)\pi L_\ell\mathfrak{R}_{n_{\mathrm{P}}}(\mathcal{F}) + 2(1-c\pi)L_\ell\mathfrak{R}'_{n_{\mathrm{U}}}(\mathcal{F}) \\
&+ \left(\frac{\pi(2-c)C_\ell}{\sqrt{n_{\mathrm{P}}}} + \frac{(1-c\pi)C_\ell}{\sqrt{n_{\mathrm{U}}}}\right)\sqrt{\frac{\ln 2/\delta}{2}}.
\end{aligned}
$$

Therefore, we have the following inequality with probability at least $1 - \delta$:

$$
\begin{aligned}
\sup_{f\in\mathcal{F}}\left|\bar{R}(f) - R(f)\right| \leqslant &\, 2(2-c)\pi L_\ell\mathfrak{R}_{n_{\mathrm{P}}}(\mathcal{F}) + 2(1-c\pi)L_\ell\mathfrak{R}'_{n_{\mathrm{U}}}(\mathcal{F}) \\
&+ \left(\frac{\pi(2-c)C_\ell}{\sqrt{n_{\mathrm{P}}}} + \frac{(1-c\pi)C_\ell}{\sqrt{n_{\mathrm{U}}}}\right)\sqrt{\frac{\ln 2/\delta}{2}}.
\end{aligned}
$$

The proof is complete. $\qquad\square$

Then, we give the proof of Theorem 2.

*Proof of Theorem 2.*

$$
\begin{aligned}
R(\bar{f}) - R(f^*) &= R(\bar{f}) - \bar{R}((\bar{f}) + \bar{R}((\bar{f}) - \bar{R}(f^*) + \bar{R}(f^*) - R(f^*) \\
&\leqslant R(\bar{f}) - \bar{R}((\bar{f}) + \bar{R}((\bar{f}) - \bar{R}(f^*) + \bar{R}(f^*) - R(f^*) \\
&\leqslant 2\sup_{f\in\mathcal{F}}\left|\bar{R}(f) - R(f)\right|.
\end{aligned}
$$

By Lemma 1, the proof is complete. $\qquad\square$

## B.6 Derivation of Equivalence of Risk Estimators

$$\bar{R}(f)$$

$$= \frac{\pi}{n_{\mathrm{P}}} \sum_{i=1}^{n_{\mathrm{P}}} \left( \ell\left(f(\boldsymbol{x}_i), +1\right) + (c-1)\ell\left(f(\boldsymbol{x}_i), -1\right) \right) + \frac{1-c\pi}{n_{\mathrm{U}}} \sum_{i=n_{\mathrm{P}}+1}^{n_{\mathrm{P}}+n_{\mathrm{U}}} \ell\left(f(\boldsymbol{x}_i), -1\right)$$

$$= \sum_{i=1}^{n_{\mathrm{P}}} \left( \frac{\pi}{n_{\mathrm{P}}} \ell\left(f(\boldsymbol{x}_i), +1\right) + \left( \frac{1}{n_{\mathrm{P}}+n_{\mathrm{U}}} - \frac{\pi}{n_{\mathrm{P}}} \right) \ell\left(f(\boldsymbol{x}_i), -1\right) \right) + \frac{1}{n_{\mathrm{P}}+n_{\mathrm{U}}} \sum_{i=n_{\mathrm{P}}+1}^{n_{\mathrm{P}}+n_{\mathrm{U}}} \ell\left(f(\boldsymbol{x}_i), -1\right)$$

$$= \frac{\pi}{n_{\mathrm{P}}} \sum_{i=1}^{n_{\mathrm{P}}} \left( \ell\left(f(\boldsymbol{x}_i), +1\right) - \ell\left(f(\boldsymbol{x}_i), -1\right) \right) + \frac{1}{n_{\mathrm{U}}} \sum_{i=1}^{n_{\mathrm{P}}+n_{\mathrm{U}}} \ell\left(f(\boldsymbol{x}_i), -1\right), \tag{10}$$

where the second equation uses the estimation $c = n_{\mathrm{P}}/\pi(n_{\mathrm{P}} + n_{\mathrm{U}})$.

## C More Experimental Details

### C.1 More Details of Benchmark Datasets

Table 5 summarizes their key characteristics, including the number of examples, feature dimensionality, positive class configurations, and task domains. For all datasets, we vary the positive rate in $\{10\%, 20\%, 30\%, 40\%, 50\%\}$. For the benchmark experiments in Section 5, we used the positive rate 30%.

Table 5: Summary of datasets used in this PU learning benchmark.

| Dataset | # Examples | # Features | Positive Classes (Case 1) | Positive Classes (Case 2) | Task Domain |
|---------|-----------|-----------|---------------------------|---------------------------|-------------|
| CIFAR-10 | 20,000 | 3,072 | {0,1,2,8,9} | {2,3,5,7,9} | Image classification |
| ImageNette | 6,000 | 12,288 | {0,1,2,8,9} | {2,3,5,7,9} | Image classification |
| USPS | 4,000 | 256 | {4,7,9,5,8} | {1,6,4,9,8} | Digit recognition |
| Letter | 13,000 | 16 | {B,V,L,R,I,O,W,S,J,K,C,H,Z} | {D,T,A,Y,Q,G,B,L,I,W,J,C,Z} | Character recognition |

### C.2 Descriptions of Algorithms

- uPU (du Plessis et al., 2015): An unbiased risk estimator that is convex when the loss function satisfies certain linear-odd conditions.
- nnPU (Kiryo et al., 2017): A non-negative risk estimator that alleviates the overfitting issue in PU learning.
- nnPU-GA (Kiryo et al., 2017): A gradient ascent formulation of nnPU.
- PUSB (Kato et al., 2019): A method that accounts for selection bias in the labeling process.
- PUbN (Hsieh et al., 2019): A framework that incorporates biased negative data into empirical risk minimization.
- VPU (Chen et al., 2020a): A variational approach that directly evaluates the modeling error of a Bayesian classifier from data.
- PAN (Hu et al., 2021): A predictive adversarial network built upon the generative adversarial network framework.
- CVIR (Garg et al., 2021): A mixture-proportion estimation method combining best bin estimation and conditional Value Ignoring Risk.
- Dist-PU (Zhao et al., 2022): A method that enforces consistency between predicted and ground-truth label distributions.
- P³MIX-E (Li et al., 2022): A mixup-based method that pairs marginal pseudo-negative instances with boundary-near positive instances, with early-learning regularization.
- P³MIX-C (Li et al., 2022): A mixup-based method that pairs marginal pseudo-negative instances with boundary-near positive instances, with pseudo-negative correction.
- LBE (Gong et al., 2022): An instance-dependent PU algorithm that jointly estimates labeling bias and learns the classifier.
- Count Loss (Shukla et al., 2023): A unified approach introducing a count-based loss penalizing deviations from arithmetic label-count constraints.

- Robust-PU (Zhu et al., 2023b): A reweighted learning framework that dynamically adjusts sample weights based on training progress and sample hardness.
- Holistic-PU (Wang et al., 2023a): A holistic method interpreting prediction scores as a temporal point process.
- PUe (Wang et al., 2023b): A causality-based method that reconstructs the loss via normalized propensity scores and inverse probability weighting.
- GLWS (Chen et al., 2024): A general weak-supervision framework formulated as Expectation-Maximization, accommodating PU data as one supervision source.

## C.3 IMPLEMENTATION DETAILS

All algorithms were implemented in PyTorch (Paszke et al., 2019), and all experiments were conducted on a single NVIDIA Tesla V100 GPU. We used the SGD optimizer and trained for 20,000 iterations across all datasets. Model performance on the validation and test sets was recorded every 100 iterations. For each dataset, we generated three random data splits. For each split, 10 random hyperparameter configurations were sampled from a predefined pool. Table 6 provides the details of the hyperparameter configurations used for all algorithms.

Table 6: Hyperparameters, their default values, and distributions for random search.

| Condition | Parameter | Default Value | Random Distribution |
|---|---|---|---|
| ResNet | learning rate | 0.001 | $10^{\text{Uniform}(-4.5,-2.5)}$ |
| | batch size | 64 | $2^{\text{Uniform}(5,8)}$ |
| | momentum | 0.9 | 0.9 |
| MLP | learning rate | 0.001 | $10^{\text{Uniform}(-4.5,-2.5)}$ |
| | batch size | 128 | $2^{\text{Uniform}(4,7)}$ |
| | momentum | 0.9 | 0.9 |
| nnPU | tolerance threshold | 0.0 | 0.0 |
| PUbN | importance of unlabeled data | 0.5 | RandomChoice([0.5,0.7,0.9]) |
| PAN | balance factor of the KL-divergences | 0.0001 | 0.0001 |
| $P^3$MIX-E | predictive score threshold | 0.85 | 0.85 |
| | size of the candidate mixup pool | 96 | 96 |
| | weight of the positive loss | 1 | 1 |
| | weight of the unlabeled loss | 1 | 1 |
| | weight of the entropy loss | 0.5 | 0.5 |
| | weight of the early-learning regularization | 5 | 5 |
| $P^3$MIX-C | predictive score threshold | 0.8 | 0.8 |
| | size of the candidate mixup pool | 96 | 96 |
| | mixup coefficient | 1.0 | 1.0 |
| | weight of the positive loss | 1 | 1 |
| | weight of the unlabeled loss | 1 | 1 |
| | weight of the entropy loss | 0.1 | 0.1 |
| LBE | warm up iteration | 2000 | 2000 |
| Robust-PU | warm up iteration | 2000 | 2000 |
| | training scheduler | linear | linear |
| | temperature in the logistic loss | 1 | RandomChoice([1,1.3]) |
| | initial threshold | 0.1 | RandomChoice([0.1,0.11]) |
| | final threshold | 2 | RandomChoice([1,2]) |
| | growing step | 10 | RandomChoice([5,10]) |
| Holistic-PU | warm up iteration | 2000 | 2000 |

# D    DETAILS OF EXPERIMENTAL RESULTS

Tables 7 to 18 report detailed experimental results in terms of different metrics on CIFAR-10, ImageNette, Letter, and USPS, and the hyperparameters are determined with PA, PAUC, and OA, respectively.

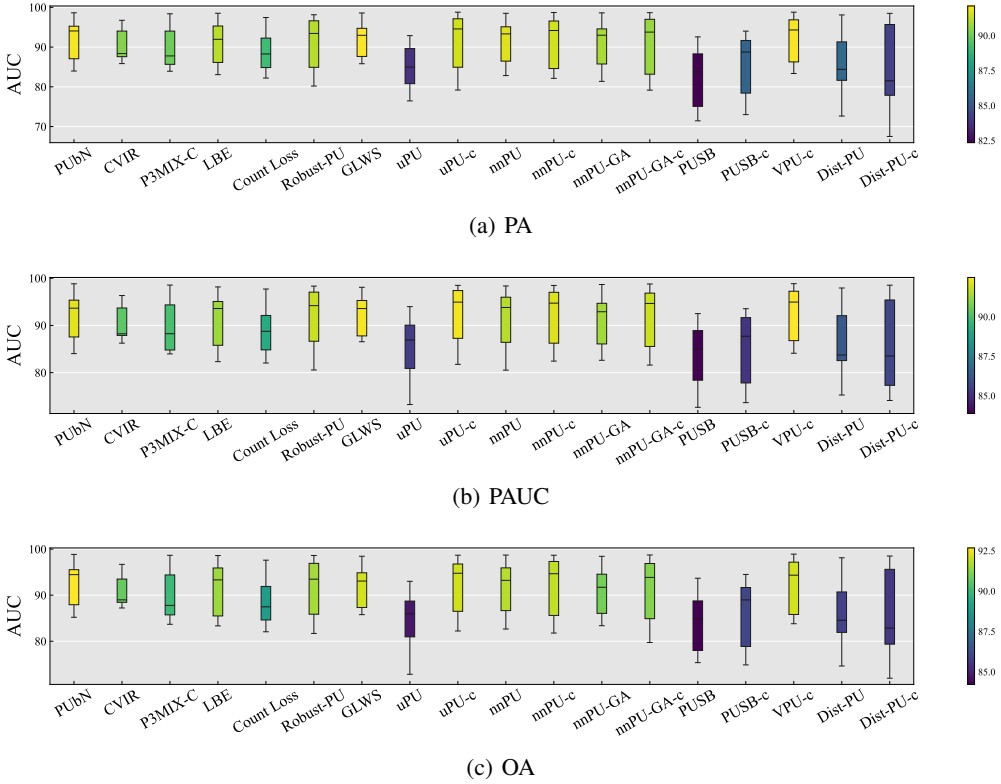

(a) PA

(b) PAUC

(c) OA

Figure 5: Overall performance w.r.t. the AUC score of different algorithms across all datasets. Hyperparameters were tuned using PA, PAUC and OA, respectively; bar colors indicate means.

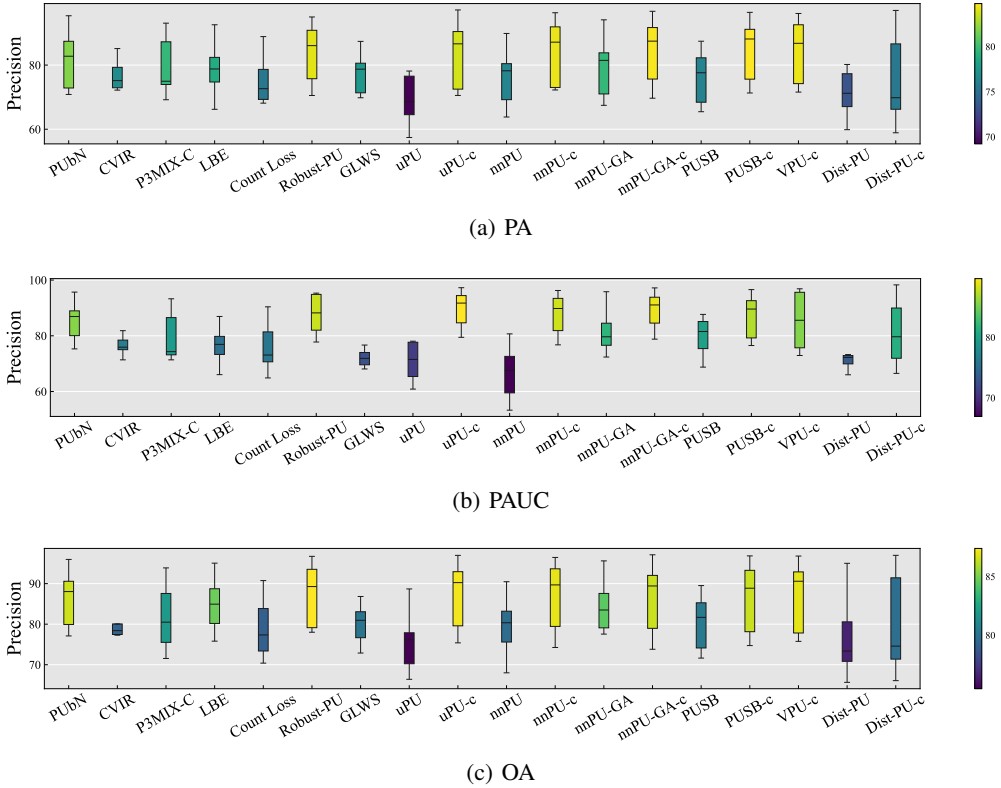

Figure 6: Overall performance w.r.t. precision of different algorithms across all datasets. Hyperparameters were tuned using PA, PAUC and OA, respectively; bar colors indicate means.

Table 7: Test results (mean±std) of precision and recall for each algorithm on CIFAR-10 (Case 1) under different model selection criteria. The best performance w.r.t. each validation metric is shown in bold. Here, "-c" indicates using the proposed calibration technique in Algorithm 1.

| Test metric | Precision | | | Recall | | |
|---|---|---|---|---|---|---|
| Val metric | PA | PAUC | OA | PA | PAUC | OA |
| PUbN | 85.58±0.37 | 86.97±0.18 | 89.46±0.65 | 87.71±1.00 | 85.25±1.87 | 84.64±0.24 |
| PAN | 71.61±0.86 | 73.33±0.14 | 79.51±1.74 | 88.39±1.65 | 86.65±1.61 | 78.41±3.09 |
| CVIR | 82.12±1.38 | 78.27±1.03 | 86.30±0.93 | 90.72±0.42 | 92.42±1.35 | 86.74±0.26 |
| P3MIX-E | 68.93±6.62 | 50.00±0.00 | 82.77±5.96 | 90.99±1.97 | **100.00±0.00** | 67.19±17.74 |
| P3MIX-C | 86.03±0.91 | 84.91±1.00 | 86.27±0.66 | 86.86±0.24 | 86.97±0.45 | 87.19±0.81 |
| LBE | 79.00±1.37 | 66.06±1.22 | 88.64±0.96 | 89.31±0.95 | 97.45±0.33 | 80.41±0.36 |
| Count Loss | 75.81±0.30 | 74.78±1.63 | 79.88±0.66 | 90.73±0.31 | 90.57±1.51 | 86.65±1.07 |
| Robust-PU | 84.19±1.05 | 89.77±1.85 | 88.23±0.69 | 87.73±1.21 | 80.72±2.97 | 82.89±0.26 |
| Holistic-PU | 50.10±0.05 | 50.00±0.00 | 78.03±0.77 | **99.49±0.22** | **100.00±0.00** | 88.60±0.41 |
| PUe | 74.23±1.27 | 80.12±1.97 | 85.70±2.17 | 85.52±0.48 | 76.39±2.72 | 73.49±1.77 |
| GLWS | 79.61±0.65 | 73.12±2.81 | 82.86±0.85 | 92.70±0.34 | 95.53±1.03 | 89.99±0.32 |
| uPU | 78.14±1.90 | 78.06±7.35 | 88.69±0.56 | 84.29±0.41 | 80.13±7.43 | 73.44±0.66 |
| uPU-c | 85.58±0.88 | 89.53±1.39 | 88.50±0.93 | 86.39±0.98 | 77.69±2.63 | 83.91±0.66 |
| nnPU | 77.17±0.27 | 68.25±0.27 | 78.73±1.19 | 91.00±0.32 | 95.62±0.62 | 88.99±1.44 |
| nnPU-c | 83.72±0.48 | 87.75±0.73 | 86.66±0.46 | 88.22±0.98 | 83.76±0.68 | 85.95±0.77 |
| nnPU-GA | 81.92±1.66 | 82.29±0.38 | 86.81±1.62 | 88.18±1.20 | 87.12±0.72 | 82.54±0.65 |
| nnPU-GA-c | 85.33±0.90 | 89.78±1.03 | 89.25±0.78 | 86.55±1.15 | 81.95±0.76 | 82.19±0.41 |
| PUSB | 76.20±1.34 | 78.13±1.39 | 78.64±0.98 | 91.93±0.81 | 90.41±0.19 | **90.44±0.20** |
| PUSB-c | 86.43±0.03 | 89.07±1.32 | 87.87±0.17 | 85.76±0.84 | 79.38±1.23 | 84.66±0.25 |
| VPU | **88.71±0.41** | **97.16±1.53** | **90.61±0.82** | 80.05±0.84 | 33.15±15.89 | 79.93±1.08 |
| VPU-c | 84.97±1.65 | 77.43±2.41 | 89.08±0.15 | 88.67±0.83 | 93.37±0.83 | 85.82±0.61 |
| Dist-PU | 76.34±0.77 | 72.78±0.89 | 84.75±0.15 | 91.79±0.56 | 93.81±0.86 | 81.86±1.26 |
| Dist-PU-c | 84.07±1.13 | 88.22±2.06 | 90.49±0.84 | 91.58±0.99 | 86.67±2.28 | 86.02±0.83 |

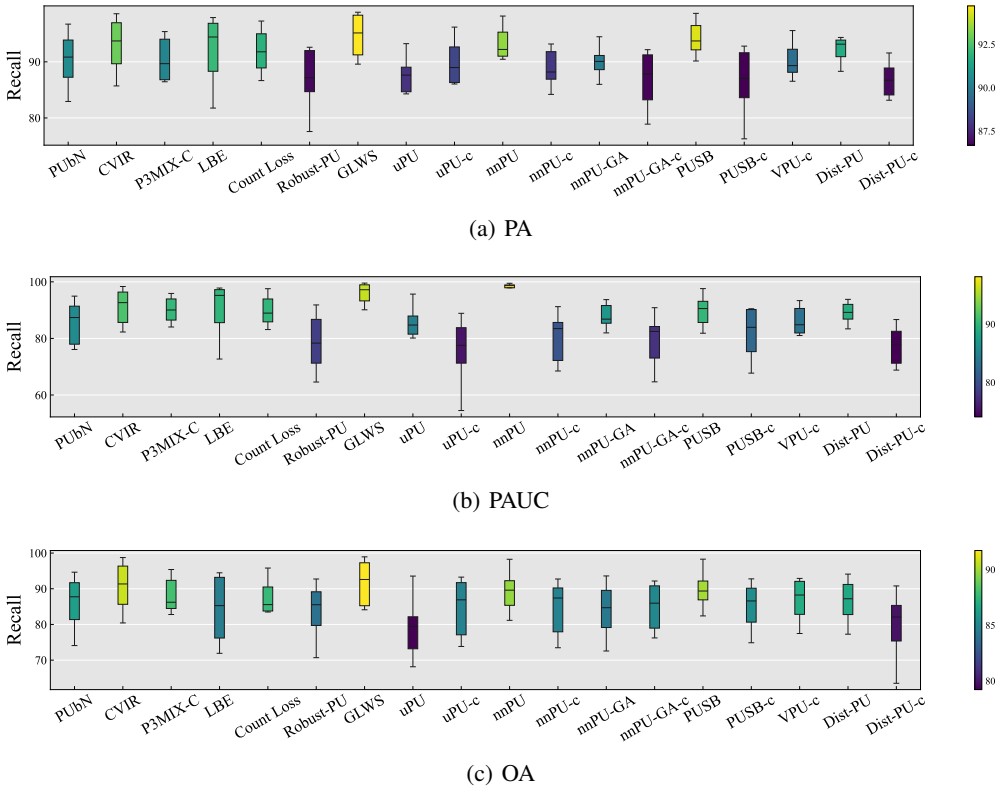

Figure 7: Overall performance w.r.t. recall of different algorithms across all datasets. Hyperparameters were tuned using PA, PAUC and OA, respectively; bar colors indicate means.

Table 8: Test results (mean±std) of precision and recall for each algorithm on CIFAR-10 (Case 2) under different model selection criteria. The best performance w.r.t. each validation metric is shown in bold. Here, "-c" indicates using the proposed calibration technique in Algorithm 1.

| Test metric | Precision | | | Recall | | |
|---|---|---|---|---|---|---|
| Val metric | PA | PAUC | OA | PA | PAUC | OA |
| PUbN | 73.21±1.41 | 80.55±1.72 | 80.14±1.05 | 89.42±0.70 | 78.25±3.04 | 79.75±0.94 |
| PAN | 57.23±1.87 | 56.98±2.87 | 59.50±2.27 | 92.77±0.38 | 94.70±2.27 | 83.05±5.42 |
| CVIR | 73.13±1.91 | 76.32±1.46 | 77.51±0.90 | 90.57±0.80 | 85.61±1.27 | 85.81±0.63 |
| P3MIX-E | 55.91±3.28 | 33.33±13.61 | 56.04±3.39 | 96.32±1.97 | 66.67±27.22 | **96.07±2.17** |
| P3MIX-C | 74.10±1.68 | 74.09±2.13 | 75.63±0.62 | 86.67±1.25 | 85.03±1.78 | 84.71±0.96 |
| LBE | 66.21±1.66 | 58.31±2.11 | 76.30±0.92 | 92.61±1.25 | 97.81±1.05 | 76.30±0.92 |
| Count Loss | 69.39±0.70 | 71.20±0.18 | 73.73±0.09 | 87.54±0.76 | 83.13±1.10 | 83.48±2.12 |
| Robust-PU | 75.32±1.11 | 82.16±1.82 | 79.06±0.85 | 86.25±1.51 | 72.97±1.95 | 80.59±1.55 |
| Holistic-PU | 53.00±0.10 | 59.93±4.67 | 67.62±0.43 | **98.93±0.21** | 54.64±23.85 | 81.48±5.21 |
| PUe | 63.65±0.22 | 62.69±2.20 | 71.16±1.47 | 86.71±0.82 | 88.03±2.08 | 71.53±3.84 |
| GLWS | 71.86±1.10 | 69.66±1.62 | 77.16±0.93 | 91.35±1.11 | 93.40±0.68 | 84.11±0.63 |
| uPU | 62.03±1.56 | 65.14±1.52 | 69.80±0.66 | 84.79±2.37 | 82.83±2.57 | 72.40±3.72 |
| uPU-c | 72.31±0.26 | 82.55±0.25 | 80.15±0.99 | 88.23±0.26 | 74.27±0.66 | 77.23±2.25 |
| nnPU | 68.39±1.63 | 57.41±0.78 | 74.03±0.92 | 91.01±1.51 | **98.65±0.40** | 85.19±0.73 |
| nnPU-c | 73.19±1.09 | 81.98±0.60 | 80.25±0.74 | 87.81±1.17 | 73.10±1.44 | 77.99±1.18 |
| nnPU-GA | 71.42±1.13 | 72.38±0.59 | 78.98±1.66 | 88.75±1.03 | 86.52±2.59 | 77.65±1.40 |
| nnPU-GA-c | 74.94±0.69 | 80.00±1.79 | 79.37±0.90 | 84.28±2.37 | 75.86±1.44 | 78.70±0.96 |
| PUSB | 69.38±0.61 | 76.30±1.64 | 74.79±0.62 | 92.21±0.15 | 84.05±2.62 | 85.62±1.96 |
| PUSB-c | 76.45±0.77 | 79.54±1.37 | 78.93±0.69 | 84.07±0.83 | 75.54±1.16 | 79.71±1.19 |
| VPU | **81.54±0.63** | **92.66±1.97** | **82.10±0.62** | 69.74±1.96 | 29.59±11.86 | 69.91±2.43 |
| VPU-c | 72.30±0.58 | 72.93±1.34 | 77.65±0.43 | 89.87±0.48 | 90.00±0.94 | 83.71±1.10 |
| Dist-PU | 68.12±0.72 | 71.23±1.19 | 71.27±1.24 | 88.31±0.40 | 83.64±1.35 | 83.07±1.45 |
| Dist-PU-c | 69.46±4.49 | 72.97±2.81 | 75.24±2.54 | 83.36±0.70 | 78.05±3.74 | 72.82±5.27 |

Table 9: Test results (mean±std) of precision and recall for each algorithm on ImageNette (Case 1) under different model selection criteria. The best performance w.r.t. each validation metric is shown in bold. Here, "-c" indicates using the proposed calibration technique in Algorithm 1.

| Test metric | Precision | | | Recall | | |
|---|---|---|---|---|---|---|
| Val metric | PA | PAUC | OA | PA | PAUC | OA |
| PUbN | 70.84±0.32 | 78.57±4.30 | 77.10±1.61 | 85.89±0.85 | 76.09±5.82 | 81.90±2.84 |
| PAN | 50.00±0.66 | 53.43±1.40 | 60.22±3.11 | 94.12±2.52 | 38.73±23.64 | 36.64±2.60 |
| CVIR | 74.49±1.81 | 75.11±3.74 | 78.28±1.36 | 86.93±1.69 | 85.67±3.79 | 85.13±1.14 |
| P3MIX-E | 71.37±2.70 | 32.75±13.37 | 71.98±2.64 | 81.92±1.42 | 66.67±27.22 | 81.48±1.38 |
| P3MIX-C | 74.23±1.92 | 74.56±2.46 | 78.59±2.18 | 87.47±0.80 | 87.34±0.62 | 82.78±1.94 |
| LBE | **76.29±0.82** | 81.74±1.60 | **80.73±0.77** | 81.76±0.93 | 73.48±3.85 | 75.86±2.11 |
| Count Loss | 69.02±1.15 | 71.41±2.49 | 74.81±1.18 | 89.37±0.93 | 86.22±1.82 | 83.77±2.66 |
| Robust-PU | 75.89±0.38 | 81.60±2.37 | 79.16±0.86 | 80.00±1.31 | 66.11±7.91 | 77.07±1.53 |
| Holistic-PU | 50.17±0.26 | 54.20±3.82 | 52.95±0.29 | 93.12±1.81 | 84.66±10.32 | 50.48±2.51 |
| PUe | 64.33±2.68 | 70.74±0.59 | 68.85±0.95 | 78.49±2.87 | 71.33±3.07 | 74.38±2.68 |
| GLWS | 69.81±1.18 | 68.10±1.82 | 75.17±0.67 | 90.92±1.13 | 92.88±0.54 | 84.51±0.74 |
| uPU | 65.37±1.04 | 60.87±4.80 | 70.38±0.55 | 87.71±0.58 | 89.47±4.66 | 80.15±1.07 |
| uPU-c | 70.53±2.49 | 85.36±2.00 | 77.92±1.59 | 86.05±3.74 | 54.50±12.20 | 76.69±1.98 |
| nnPU | 69.49±1.81 | 60.22±0.88 | 76.11±2.64 | 90.46±1.00 | **95.92±0.48** | 81.15±3.55 |
| nnPU-c | 72.51±1.01 | 81.53±2.56 | 77.04±0.57 | 84.20±0.62 | 69.52±4.15 | 77.77±2.26 |
| nnPU-GA | 69.74±0.60 | 76.68±1.43 | 79.13±2.26 | 89.37±0.77 | 81.95±3.81 | 79.63±2.88 |
| nnPU-GA-c | 75.88±1.16 | **86.08±2.79** | 77.78±0.87 | 80.10±1.46 | 54.93±8.53 | 79.05±1.77 |
| PUSB | 65.46±0.55 | 72.72±2.12 | 71.63±0.81 | **94.26±0.58** | 86.17±2.78 | **87.31±1.24** |
| PUSB-c | 73.08±0.57 | 78.35±1.28 | 75.79±1.41 | 82.24±0.99 | 74.69±1.16 | 80.96±2.74 |
| VPU | 61.32±5.55 | 33.33±27.22 | 80.12±7.64 | 59.47±17.51 | 0.07±0.06 | 34.73±6.80 |
| VPU-c | 74.86±1.16 | 75.32±0.83 | 77.88±1.56 | 81.67±0.95 | 82.23±0.80 | 77.46±0.23 |
| Dist-PU | 63.85±2.20 | 66.01±2.40 | 69.52±0.61 | 92.79±0.44 | 89.14±1.38 | 86.31±1.05 |
| Dist-PU-c | 67.29±1.38 | 81.49±4.56 | 72.42±1.95 | 84.35±1.74 | 43.02±11.82 | 76.21±2.39 |

Table 10: Test results (mean±std) of precision and recall for each algorithm on ImageNette (Case 2) under different model selection criteria. The best performance w.r.t. each validation metric is shown in bold. Here, "-c" indicates using the proposed calibration technique in Algorithm 1.

| Test metric | Precision | | | Recall | | |
|---|---|---|---|---|---|---|
| Val metric | PA | PAUC | OA | PA | PAUC | OA |
| PUbN | 71.79±1.10 | 75.30±1.17 | **79.28±1.74** | 82.94±1.86 | 77.08±3.45 | 74.08±2.39 |
| PAN | 51.63±0.24 | 70.75±3.83 | 64.81±2.19 | **93.47±1.86** | 53.01±8.18 | 61.43±3.03 |
| CVIR | 72.31±1.00 | 75.51±1.45 | 78.53±1.48 | 85.71±0.51 | 82.27±1.44 | 80.42±1.40 |
| P3MIX-E | 59.73±5.02 | 33.05±13.49 | 60.00±5.03 | 82.05±9.32 | 66.67±27.22 | 82.13±9.07 |
| P3MIX-C | 69.19±0.86 | 71.42±0.87 | 71.54±1.30 | 86.45±0.93 | 84.04±0.35 | 83.78±0.78 |
| LBE | 70.02±1.26 | 75.72±1.63 | 78.52±0.55 | 85.34±1.94 | 72.71±4.32 | 71.91±2.57 |
| Count Loss | 68.13±0.40 | 69.07±0.31 | 70.39±0.88 | 86.66±0.67 | 84.79±0.94 | 83.73±0.75 |
| Robust-PU | 70.52±2.00 | 77.77±2.80 | 78.01±1.20 | 77.58±1.81 | 64.58±4.76 | 70.69±2.37 |
| Holistic-PU | 54.06±1.09 | 53.98±1.86 | 58.96±0.87 | 82.10±6.17 | 71.70±7.18 | 58.89±1.87 |
| PUe | 60.79±0.12 | 63.24±1.65 | 65.25±0.23 | 80.82±2.80 | 75.25±0.71 | 74.24±2.00 |
| GLWS | 69.84±0.87 | 69.39±0.41 | 72.88±0.68 | 89.59±0.58 | 90.12±0.64 | **85.49±1.26** |
| uPU | 57.42±2.33 | 65.47±2.34 | 66.40±2.65 | 84.31±3.94 | 71.10±5.64 | 68.14±6.00 |
| uPU-c | **72.56±2.89** | **79.46±1.02** | 75.39±0.26 | 73.42±3.51 | 62.21±3.91 | 73.83±1.50 |
| nnPU | 63.81±0.91 | 53.31±2.09 | 68.00±0.93 | 91.01±2.53 | **98.56±0.94** | 85.41±1.95 |
| nnPU-c | 72.24±0.48 | 76.75±2.56 | 74.25±0.69 | 78.73±3.00 | 68.48±6.78 | 73.47±3.67 |
| nnPU-GA | 67.45±1.90 | 76.34±0.93 | 77.55±1.48 | 85.99±2.22 | 72.88±4.68 | 72.56±2.15 |
| nnPU-GA-c | 69.67±2.08 | 78.80±0.62 | 73.81±1.46 | 78.90±1.89 | 64.65±0.71 | 76.22±3.06 |
| PUSB | 65.52±2.07 | 68.76±0.60 | 72.14±1.68 | 90.15±1.50 | 81.84±0.98 | 82.39±2.42 |
| PUSB-c | 71.30±1.31 | 76.51±0.32 | 74.72±1.46 | 76.28±0.92 | 67.73±0.61 | 74.87±1.76 |
| VPU | 66.96±2.61 | 55.05±22.59 | 78.42±1.24 | 81.55±2.30 | 22.49±18.22 | 63.98±3.62 |
| VPU-c | 71.58±0.54 | 75.82±0.96 | 75.76±1.51 | 86.54±1.40 | 81.02±0.43 | 80.18±0.64 |
| Dist-PU | 59.86±1.20 | 63.30±0.69 | 65.65±1.35 | 84.36±2.94 | 83.38±1.52 | 77.27±1.64 |
| Dist-PU-c | 58.89±4.09 | 66.51±1.59 | 68.25±0.62 | 83.16±7.58 | 72.05±2.14 | 63.51±5.76 |

Table 11: Test results (mean±std) of accuracy, AUC, and F1 score for each algorithm on Letter (Case 1) under different model selection criteria. The best performance w.r.t. each validation metric is shown in bold. Here, "-c" indicates using the proposed calibration technique in Algorithm 1.

| Test metric | Test ACC | | | AUC | | | Test F1 | | |
|---|---|---|---|---|---|---|---|---|---|
| Val metric | PA | PAUC | OA | PA | PAUC | OA | PA | PAUC | OA |
| PUbN | 88.92±1.90 | 89.05±2.11 | 89.70±1.38 | 94.24±1.48 | 94.48±1.35 | 94.61±1.20 | 89.59±1.58 | 89.43±1.70 | 89.57±1.46 |
| PAN | 49.28±0.27 | 48.20±0.54 | 52.18±1.24 | 47.05±2.18 | 55.92±0.51 | 46.69±2.30 | 65.40±0.26 | 65.04±0.49 | 42.19±17.25 |
| CVIR | 83.35±0.56 | 82.60±0.75 | 84.67±0.58 | 86.40±0.65 | 87.63±0.99 | 87.78±0.90 | 85.16±0.50 | 84.33±0.62 | 85.86±0.35 |
| P3MIX-E | 51.80±1.39 | 49.62±0.87 | 61.42±4.12 | 60.70±5.00 | 81.42±0.26 | 67.00±7.82 | 67.12±0.64 | 43.85±17.57 | 42.69±17.49 |
| P3MIX-C | 80.03±1.13 | 77.58±2.53 | 80.92±1.14 | 85.08±1.23 | 84.43±1.62 | 84.50±0.68 | 82.46±0.83 | 80.56±1.73 | 82.83±0.96 |
| LBE | 85.63±1.13 | 81.37±2.19 | 87.55±0.28 | 91.81±1.52 | 93.96±0.29 | 94.38±0.23 | 87.17±0.85 | 83.32±1.15 | 87.44±0.32 |
| Count Loss | 77.67±0.86 | 73.15±1.96 | 78.27±1.01 | 86.31±1.48 | 87.17±1.55 | 84.67±0.78 | 80.27±0.67 | 77.19±1.62 | 79.98±0.84 |
| Robust-PU | 90.02±0.67 | 89.17±0.33 | 90.63±0.31 | 95.30±0.29 | 95.51±0.32 | 95.91±0.31 | 90.20±0.61 | 89.09±0.66 | 90.58±0.32 |
| Holistic-PU | 85.80±0.99 | 75.22±9.45 | 87.32±1.27 | 94.12±1.36 | 95.72±1.49 | 94.74±1.64 | 87.14±0.83 | 80.97±5.61 | 88.17±1.02 |
| PUe | 79.50±0.24 | 81.83±1.08 | 82.00±0.78 | 89.77±1.07 | 91.42±0.98 | 90.88±0.50 | 81.54±0.21 | 82.32±1.64 | 81.95±1.08 |
| GLWS | 85.87±0.95 | 80.93±1.54 | 86.32±0.58 | 92.91±0.63 | 93.62±0.45 | 92.65±0.83 | 87.03±0.75 | 83.53±1.17 | 87.28±0.54 |
| uPU | 74.98±1.19 | 79.75±0.63 | 77.72±0.79 | 85.87±0.59 | 88.34±0.29 | 86.19±0.71 | 78.05±1.00 | 79.62±0.40 | 77.65±1.10 |
| uPU-c | **92.23±0.26** | 85.97±4.01 | **92.73±0.15** | **96.84±0.15** | **97.26±0.05** | 96.40±0.18 | **92.18±0.14** | 83.09±6.02 | **92.60±0.14** |
| nnPU | 85.13±0.46 | 79.53±1.62 | 85.60±0.31 | 94.16±0.51 | 95.44±0.44 | 94.49±0.66 | 86.19±0.37 | 82.46±1.10 | 85.85±0.40 |
| nnPU-c | 91.87±0.34 | 89.25±1.14 | 91.82±0.14 | 96.15±0.30 | 96.39±0.69 | 96.36±0.38 | 91.85±0.25 | 88.24±1.71 | 91.58±0.21 |
| nnPU-GA | 85.12±0.13 | 82.85±0.68 | 84.27±0.58 | 93.17±0.44 | 93.56±0.61 | 91.18±0.41 | 85.74±0.25 | 82.86±1.69 | 84.46±0.64 |
| nnPU-GA-c | 90.97±0.30 | 88.60±0.57 | 90.97±0.30 | 94.72±0.23 | 96.37±1.16 | 94.72±0.23 | 90.86±0.25 | 87.75±0.25 | 90.86±0.25 |
| PUSB | 85.73±0.70 | 87.43±0.21 | 86.82±0.54 | 86.09±0.63 | 87.42±0.25 | 86.81±0.56 | 86.63±0.67 | 87.66±0.50 | 86.70±0.78 |
| PUSB-c | 91.42±0.86 | **90.68±0.58** | 91.43±0.92 | 91.45±0.87 | 90.66±0.57 | 91.46±0.92 | 91.35±1.02 | **90.30±0.67** | 91.29±1.04 |
| VPU | 89.85±1.07 | 67.88±8.64 | 90.13±0.77 | 95.67±0.40 | 96.03±0.77 | 95.44±0.57 | 89.69±0.98 | 44.13±20.00 | 89.86±0.67 |
| VPU-c | 91.83±0.54 | 90.28±0.98 | 92.15±0.52 | 96.32±0.38 | 97.06±0.26 | **96.96±0.30** | 91.95±0.42 | 89.38±1.17 | 91.93±0.48 |
| Dist-PU | 77.07±0.77 | 77.45±0.78 | 77.55±0.78 | 81.95±1.07 | 82.71±1.23 | 82.07±1.66 | 80.15±0.45 | 79.68±0.14 | 80.07±0.40 |
| Dist-PU-c | 67.65±2.41 | 69.33±2.52 | 70.03±2.28 | 72.96±2.78 | 75.75±2.56 | 74.72±2.66 | 72.61±1.10 | 68.78±2.64 | 72.81±2.05 |

Table 12: Test results (mean±std) of precision and recall for each algorithm on Letter (Case 1) under different model selection criteria. The best performance w.r.t. each validation metric is shown in bold. Here, "-c" indicates using the proposed calibration technique in Algorithm 1.

| Test metric | Precision | | | Recall | | |
|---|---|---|---|---|---|---|
| Val metric | PA | PAUC | OA | PA | PAUC | OA |
| PUbN | 84.12±2.75 | 86.86±4.01 | 88.33±1.64 | 96.00±0.44 | 92.79±2.16 | 90.85±1.27 |
| PAN | 49.11±0.23 | 48.20±0.54 | 34.01±13.90 | 97.89±1.15 | 100.00±0.00 | 56.40±23.56 |
| CVIR | 75.82±0.92 | 74.75±0.73 | 77.28±0.52 | 97.17±0.70 | 96.73±0.42 | 96.58±0.14 |
| P3MIX-E | 50.79±0.93 | 65.70±14.00 | 45.23±18.89 | 99.04±0.74 | 66.80±27.10 | 42.64±18.48 |
| P3MIX-C | 73.49±1.22 | 71.37±2.78 | 75.15±0.90 | 94.00±1.10 | 92.77±0.92 | 92.26±1.02 |
| LBE | 78.58±1.20 | 75.84±4.43 | 85.17±1.79 | 97.89±0.29 | 93.97±3.87 | 90.14±2.36 |
| Count Loss | 69.56±1.08 | 64.88±1.99 | 72.33±0.83 | 94.95±0.69 | 95.42±1.62 | 89.44±0.97 |
| Robust-PU | 87.94±0.82 | 86.68±1.06 | 90.32±0.77 | 92.60±0.67 | 91.84±2.47 | 90.86±0.32 |
| Holistic-PU | 79.36±1.07 | 71.38±8.69 | 82.39±2.08 | 96.62±0.46 | 96.74±1.50 | 94.99±1.21 |
| PUe | 73.33±0.21 | 78.59±0.74 | 80.47±0.23 | 91.82±0.32 | 86.90±4.26 | 83.62±2.52 |
| GLWS | 78.56±1.32 | 72.32±1.87 | 79.44±0.61 | 97.60±0.27 | 98.98±0.30 | **96.84±0.48** |
| uPU | 67.24±1.28 | 77.65±2.09 | 74.98±0.74 | 93.05±0.71 | 81.96±1.61 | 80.56±1.73 |
| uPU-c | 89.10±0.60 | 93.95±2.97 | 92.00±0.92 | 95.49±0.48 | 77.42±10.67 | 93.26±0.91 |
| nnPU | 79.27±0.28 | 70.89±1.82 | 81.97±1.57 | 94.44±0.61 | 98.70±0.39 | 90.42±2.48 |
| nnPU-c | **90.57±0.67** | 93.06±1.83 | **93.26±1.26** | 93.18±0.49 | 84.51±4.36 | 90.08±1.46 |
| nnPU-GA | 81.01±0.13 | 81.22±3.50 | 80.81±1.19 | 91.07±0.63 | 86.51±6.63 | 88.48±0.53 |
| nnPU-GA-c | 89.60±0.51 | 92.37±3.02 | 89.60±0.51 | 92.17±0.32 | 84.05±2.57 | 92.17±0.32 |
| PUSB | 79.01±1.09 | 84.95±0.84 | 84.96±1.33 | 95.94±0.96 | 90.70±2.01 | 88.78±2.76 |
| PUSB-c | 90.00±1.00 | 90.13±0.90 | 89.87±0.76 | 92.79±1.59 | 90.49±0.97 | 92.77±1.40 |
| VPU | 89.11±1.55 | 65.54±26.76 | 91.24±1.24 | 90.42±1.99 | 35.30±17.59 | 88.61±1.31 |
| VPU-c | 88.60±0.48 | **95.21±0.75** | 92.09±0.68 | 95.57±0.65 | 84.40±2.68 | 91.84±1.42 |
| Dist-PU | 70.13±0.47 | 72.05±1.16 | 71.50±1.04 | 93.51±0.55 | 89.28±1.49 | 91.14±1.75 |
| Dist-PU-c | 63.03±3.53 | 68.85±3.51 | 66.07±3.28 | 87.01±3.71 | 68.80±1.90 | 81.54±2.35 |

Table 13: Test results (mean±std) of accuracy, AUC, and F1 score for each algorithm on Letter (Case 2) under different model selection criteria. The best performance w.r.t. each validation metric is shown in bold. Here, "-c" indicates using the proposed calibration technique in Algorithm 1.

| Test metric | Test ACC | | | AUC | | | Test F1 | | |
|---|---|---|---|---|---|---|---|---|---|
| Val metric | PA | PAUC | OA | PA | PAUC | OA | PA | PAUC | OA |
| PUbN | 87.47±0.58 | 88.98±1.45 | 89.63±0.98 | 94.15±1.14 | 93.88±1.45 | 94.59±1.09 | 88.40±0.61 | 89.15±1.44 | 89.74±1.02 |
| PAN | 50.02±0.48 | 49.88±0.85 | 51.73±1.43 | 45.39±4.63 | 57.60±2.16 | 51.85±4.33 | 66.64±0.40 | 44.18±18.05 | 21.43±17.50 |
| CVIR | 84.83±0.73 | 84.22±0.89 | 84.72±0.76 | 88.63±1.49 | 88.18±0.65 | 88.67±1.62 | 86.57±0.61 | 85.38±0.87 | 86.38±0.63 |
| P3MIX-E | 55.70±2.92 | 55.57±2.96 | 65.08±3.09 | 71.43±4.39 | 81.48±2.05 | 71.19±3.66 | 68.60±0.97 | 52.06±13.86 | 64.22±0.99 |
| P3MIX-C | 81.80±2.04 | 80.70±2.16 | 83.32±2.22 | 89.68±2.56 | 90.09±2.58 | 88.23±3.66 | 83.89±1.46 | 83.35±1.46 | 83.46±2.56 |
| LBE | 87.32±0.50 | 80.82±3.97 | 88.18±0.96 | 94.51±0.18 | 94.43±0.11 | 95.34±0.57 | 88.44±0.39 | 82.61±2.51 | 88.65±1.08 |
| Count Loss | 81.35±0.64 | 82.20±1.16 | 82.93±0.85 | 90.22±0.71 | 90.35±0.73 | 90.08±1.03 | 83.36±0.36 | 83.19±0.35 | 83.30±0.37 |
| Robust-PU | 90.88±0.52 | 90.18±0.84 | 91.07±0.39 | 96.31±0.65 | **96.92±0.53** | 96.63±0.57 | 91.00±0.44 | 89.64±1.06 | 90.83±0.42 |
| Holistic-PU | 87.88±1.37 | 86.12±1.82 | 88.65±1.12 | 95.09±0.62 | 95.36±0.69 | 95.40±0.83 | 88.79±0.90 | 87.58±1.22 | 89.49±0.85 |
| PUe | 79.50±0.70 | 78.03±1.17 | 82.53±0.04 | 88.18±1.99 | 91.92±0.19 | 90.65±0.40 | 80.97±0.48 | 80.73±0.63 | 81.94±0.20 |
| GLWS | 86.27±0.43 | 79.88±1.42 | 88.18±0.67 | 93.00±0.61 | 94.46±0.33 | 93.75±0.14 | 87.50±0.43 | 82.97±0.76 | 89.07±0.51 |
| uPU | 75.22±1.07 | 72.03±2.17 | 77.52±0.34 | 84.07±1.04 | 85.49±0.25 | 85.72±0.35 | 77.80±0.45 | 75.49±0.59 | 77.93±0.57 |
| uPU-c | 91.32±0.57 | 89.72±0.59 | 92.13±0.18 | 96.48±0.25 | 96.86±0.22 | 96.30±0.53 | 91.71±0.35 | 89.05±0.91 | 92.14±0.27 |
| nnPU | 84.68±0.34 | 75.22±2.11 | 87.38±0.39 | 94.02±0.74 | 95.30±0.51 | 95.34±0.24 | 85.90±0.44 | 79.94±1.42 | 87.73±0.39 |
| nnPU-c | 91.27±0.43 | 90.50±0.17 | 91.65±0.19 | 96.21±0.40 | **96.92±0.12** | **97.09±0.22** | 91.44±0.44 | 90.42±0.19 | 91.65±0.22 |
| nnPU-GA | 85.63±0.60 | 83.43±1.40 | 86.15±0.13 | 93.84±0.34 | 93.79±0.09 | 93.68±0.02 | 86.37±0.67 | 85.00±1.00 | 86.42±0.41 |
| nnPU-GA-c | 91.55±0.33 | 89.28±1.89 | 91.70±0.39 | **96.79±0.42** | 96.65±0.43 | 96.61±0.56 | 91.58±0.37 | 88.44±2.57 | 91.69±0.41 |
| PUSB | 87.42±0.31 | 87.83±0.13 | 87.63±0.24 | 87.39±0.34 | 87.85±0.13 | 87.61±0.23 | 88.15±0.18 | 88.30±0.24 | 87.98±0.44 |
| PUSB-c | 91.33±0.77 | **91.48±0.40** | 91.53±0.71 | 91.34±0.76 | 91.46±0.41 | 91.47±0.76 | 91.29±0.84 | **91.23±0.51** | 91.22±0.95 |
| VPU | 90.85±0.28 | 74.93±6.54 | 91.18±0.08 | 96.26±0.24 | 95.91±0.10 | 96.23±0.26 | 90.60±0.36 | 64.86±10.97 | 90.98±0.10 |
| VPU-c | **91.95±0.38** | 89.55±0.05 | **92.85±0.29** | 96.63±0.26 | 96.40±0.48 | 96.89±0.12 | **91.94±0.33** | 89.13±0.38 | **92.74±0.27** |
| Dist-PU | 78.92±0.89 | 77.52±0.51 | 79.42±0.71 | 84.82±0.29 | 84.29±0.73 | 85.17±0.34 | 81.73±0.94 | 79.36±0.51 | 81.10±0.82 |
| Dist-PU-c | 75.33±1.22 | 77.58±0.65 | 76.87±0.77 | 82.69±0.74 | 84.55±0.23 | 83.73±0.43 | 78.14±1.03 | 77.51±1.23 | 78.00±1.36 |

Table 14: Test results (mean±std) of precision and recall for each algorithm on Letter (Case 2) under different model selection criteria. The best performance w.r.t. each validation metric is shown in bold. Here, "-c" indicates using the proposed calibration technique in Algorithm 1.

| Test metric | Precision | | | Recall | | |
|---|---|---|---|---|---|---|
| Val metric | PA | PAUC | OA | PA | PAUC | OA |
| PUbN | 81.40±0.45 | 87.45±1.99 | 87.75±1.02 | 96.72±0.84 | 90.94±0.83 | 91.83±1.24 |
| PAN | 50.04±0.51 | 33.05±13.52 | 18.13±14.80 | **99.74±0.21** | 66.67±27.22 | 26.22±21.41 |
| CVIR | 78.35±0.93 | 79.00±0.54 | 78.58±1.09 | 96.73±0.14 | 92.88±1.34 | **95.95±0.56** |
| P3MIX-E | 53.00±1.72 | 69.22±12.66 | 66.29±4.29 | 97.68±1.77 | 67.78±23.64 | 63.34±2.72 |
| P3MIX-C | 75.69±1.88 | 73.75±1.92 | 82.40±1.72 | 94.13±0.75 | 95.91±1.19 | 85.25±5.36 |
| LBE | 81.36±0.74 | 79.16±6.57 | 84.70±1.08 | 96.89±0.35 | 89.61±6.60 | 93.03±1.51 |
| Count Loss | 75.71±1.09 | 80.17±3.40 | 82.45±2.34 | 92.84±1.32 | 87.33±3.35 | 84.47±1.68 |
| Robust-PU | 89.96±1.62 | 94.75±0.38 | **93.22±0.86** | 92.17±0.98 | 85.16±2.21 | 88.58±0.79 |
| Holistic-PU | 84.34±2.63 | 81.09±3.15 | 85.42±2.12 | 94.05±1.39 | 95.65±1.66 | 94.18±1.43 |
| PUe | 74.35±1.10 | 70.71±1.81 | 82.69±0.51 | 89.04±1.83 | 94.32±1.43 | 81.25±0.86 |
| GLWS | 78.89±0.68 | 71.48±1.33 | 83.68±0.94 | 98.23±0.11 | 98.94±0.40 | 95.21±0.38 |
| uPU | 70.01±1.43 | 67.60±4.04 | 77.28±0.22 | 87.70±1.18 | 87.40±4.78 | 78.61±1.09 |
| uPU-c | 87.67±0.95 | 94.08±1.70 | 92.56±0.46 | 96.19±0.92 | 84.81±2.70 | 91.72±0.28 |
| nnPU | 79.59±0.97 | 66.88±1.98 | 85.51±1.48 | 93.41±1.66 | **99.49±0.08** | 90.21±1.41 |
| nnPU-c | 91.19±1.12 | 91.85±0.63 | 92.71±0.80 | 91.80±1.68 | 89.05±0.60 | 90.66±1.05 |
| nnPU-GA | 82.09±1.25 | 78.07±1.98 | 86.18±1.35 | 91.33±2.21 | 93.44±0.65 | 86.84±1.84 |
| nnPU-GA-c | 91.10±0.34 | 93.73±1.91 | 91.70±0.36 | 92.09±0.96 | 84.69±5.82 | 91.69±0.83 |
| PUSB | 83.69±1.06 | 85.27±0.95 | 86.21±0.86 | 93.19±1.01 | 91.66±1.39 | 89.95±1.78 |
| PUSB-c | 89.84±0.67 | 92.04±0.36 | 92.96±0.25 | 92.80±1.20 | 90.44±0.72 | 89.58±1.67 |
| VPU | **92.35±0.79** | **98.33±0.85** | 92.56±0.96 | 89.00±1.39 | 51.86±13.08 | 89.52±0.97 |
| VPU-c | 92.02±0.61 | 93.71±1.97 | 92.76±0.53 | 91.85±0.15 | 85.28±2.42 | 92.75±0.68 |
| Dist-PU | 72.28±1.26 | 72.46±1.03 | 75.23±1.44 | 94.04±0.38 | 87.89±1.96 | 88.08±1.38 |
| Dist-PU-c | 70.19±2.12 | 77.82±0.81 | 73.93±0.49 | 88.32±0.66 | 77.27±1.93 | 82.61±2.50 |

Table 15: Test results (mean±std) of accuracy, AUC, and F1 score for each algorithm on USPS (Case 1) under different model selection criteria. The best performance w.r.t. each validation metric is shown in bold. Here, "-c" indicates using the proposed calibration technique in Algorithm 1.

| Test metric | Test ACC | | | AUC | | | Test F1 | | |
|---|---|---|---|---|---|---|---|---|---|
| Val metric | PA | PAUC | OA | PA | PAUC | OA | PA | PAUC | OA |
| PUbN | **93.76±0.23** | 92.89±0.24 | **93.95±0.13** | **98.28±0.03** | 98.01±0.08 | **98.29±0.04** | **92.60±0.28** | 91.42±0.36 | **92.77±0.17** |
| PAN | 84.52±0.50 | 84.52±0.65 | 84.97±0.48 | 89.98±0.20 | 90.89±0.48 | 90.06±0.46 | 81.23±0.46 | 80.56±1.19 | 80.61±0.58 |
| CVIR | 82.79±1.48 | 82.01±0.96 | 82.98±1.39 | 94.88±0.36 | 93.80±0.16 | 93.42±0.69 | 82.72±1.31 | 81.95±0.79 | 82.76±1.26 |
| P3MIX-E | 88.99±1.40 | 89.49±1.29 | 89.84±1.17 | 96.18±0.44 | 96.33±0.43 | 96.23±0.47 | 87.54±1.30 | 88.02±1.23 | 87.90±1.30 |
| P3MIX-C | 92.69±0.66 | **93.47±0.49** | 93.22±0.31 | 97.98±0.22 | **98.16±0.14** | 98.09±0.11 | 91.41±0.78 | **92.38±0.57** | 92.05±0.36 |
| LBE | 91.45±0.62 | 87.10±1.25 | 92.29±0.33 | 97.67±0.12 | 97.04±0.46 | 97.60±0.18 | 90.52±0.55 | 86.49±1.17 | 91.16±0.18 |
| Count Loss | 91.99±0.34 | 90.08±0.84 | 91.76±0.81 | 97.44±0.27 | 97.27±0.09 | 97.60±0.09 | 90.97±0.31 | 88.91±0.66 | 90.64±0.69 |
| Robust-PU | 91.73±0.27 | 88.19±3.28 | 92.79±0.12 | 97.51±0.20 | 97.48±0.22 | 97.73±0.15 | 89.88±0.20 | 83.74±5.36 | 91.20±0.14 |
| Holistic-PU | 91.94±0.82 | 92.56±0.11 | 93.46±0.36 | 97.22±0.34 | 97.47±0.17 | 97.76±0.16 | 90.88±0.84 | 91.12±0.02 | 92.27±0.40 |
| PUe | 84.82±1.01 | 84.22±0.30 | 86.93±0.27 | 95.41±0.12 | 95.25±0.13 | 94.40±1.03 | 84.23±0.76 | 83.60±0.19 | 85.24±0.59 |
| GLWS | 91.13±0.37 | 86.78±0.60 | 90.52±0.47 | 98.21±0.02 | 97.78±0.17 | 98.18±0.09 | 90.40±0.36 | 86.38±0.55 | 89.81±0.45 |
| uPU | 83.14±0.93 | 83.87±0.11 | 83.86±0.83 | 92.88±0.15 | 93.10±0.18 | 93.01±0.05 | 81.51±0.81 | 81.98±0.19 | 82.04±0.70 |
| uPU-c | 93.44±0.26 | 91.30±1.16 | 93.32±0.10 | 97.95±0.12 | 97.79±0.11 | 97.85±0.09 | 92.05±0.34 | 88.94±1.72 | 91.94±0.16 |
| nnPU | 90.60±0.28 | 87.49±0.81 | 90.22±0.42 | 97.94±0.09 | 97.63±0.06 | 97.70±0.15 | 89.82±0.27 | 87.02±0.73 | 89.44±0.42 |
| nnPU-c | 92.64±0.08 | 90.82±0.94 | 93.24±0.18 | 97.60±0.05 | 97.34±0.17 | 97.99±0.03 | 91.03±0.12 | 88.41±1.37 | 91.76±0.23 |
| nnPU-GA | 91.28±0.16 | 92.46±0.11 | 92.51±0.31 | 96.79±0.10 | 97.41±0.11 | 97.17±0.27 | 89.80±0.36 | 91.09±0.07 | 91.30±0.35 |
| nnPU-GA-c | 92.76±0.38 | 90.60±1.44 | 92.79±0.22 | 97.58±0.05 | 97.46±0.16 | 97.66±0.11 | 91.23±0.55 | 88.05±2.15 | 91.33±0.32 |
| PUSB | 89.90±0.73 | 91.73±0.26 | 91.38±0.83 | 90.98±0.65 | 92.51±0.26 | 92.17±0.70 | 89.17±0.71 | 90.91±0.29 | 90.56±0.80 |
| PUSB-c | 92.91±0.30 | 92.84±0.24 | 92.83±0.18 | 92.30±0.29 | 92.26±0.27 | 92.25±0.29 | 91.34±0.35 | 91.28±0.31 | 91.26±0.28 |
| VPU | 88.14±2.21 | 57.71±0.04 | 89.89±1.71 | 92.98±3.98 | 97.31±0.13 | 97.76±0.19 | 84.36±3.09 | 0.31±0.17 | 86.58±2.62 |
| VPU-c | 92.92±0.07 | 80.17±7.36 | 93.29±0.32 | 97.55±0.13 | 97.82±0.24 | 97.79±0.18 | 91.40±0.08 | 63.80±17.92 | 91.97±0.38 |
| Dist-PU | 87.73±0.55 | 82.15±2.23 | 86.10±0.14 | 92.52±0.85 | 92.58±0.43 | 91.03±0.77 | 86.69±0.55 | 81.64±1.82 | 84.77±0.17 |
| Dist-PU-c | 92.01±0.19 | 90.47±0.77 | 91.50±0.34 | 97.92±0.16 | 97.95±0.21 | 97.74±0.21 | 90.16±0.22 | 87.84±1.11 | 89.44±0.44 |

Table 16: Test results (mean±std) of precision and recall for each algorithm on USPS (Case 1) under different model selection criteria. The best performance w.r.t. each validation metric is shown in bold. Here, "-c" indicates using the proposed calibration technique in Algorithm 1.

| Test metric | Precision | | | Recall | | |
|---|---|---|---|---|---|---|
| Val metric | PA | PAUC | OA | PA | PAUC | OA |
| PUbN | 92.93±0.28 | 93.46±0.72 | 93.93±0.10 | 92.27±0.31 | 89.53±1.26 | 91.65±0.35 |
| PAN | 84.90±5.07 | 87.64±5.36 | 90.59±5.35 | 79.49±4.77 | 76.59±6.00 | 74.27±5.22 |
| CVIR | 72.19±1.80 | 71.37±1.25 | 72.62±1.66 | 96.90±0.42 | 96.27±0.23 | 96.27±0.86 |
| P3MIX-E | 85.21±3.71 | 86.04±3.34 | 89.31±3.14 | 90.47±1.33 | 90.43±1.08 | 87.02±2.80 |
| P3MIX-C | 90.96±0.73 | 91.40±0.59 | 91.48±0.43 | 91.88±0.91 | 93.37±0.55 | 92.63±0.31 |
| LBE | 85.53±1.52 | 78.00±1.86 | 89.02±2.12 | 96.24±0.81 | 97.18±1.13 | 93.69±1.91 |
| Count Loss | 87.21±1.11 | 85.10±2.53 | 88.05±2.45 | 95.14±0.74 | 93.45±1.97 | 93.65±1.43 |
| Robust-PU | 93.49±1.50 | 95.16±0.47 | 94.46±0.25 | 86.63±0.95 | 75.96±8.06 | 88.16±0.16 |
| Holistic-PU | 87.61±1.66 | 92.29±1.53 | 92.36±0.77 | 94.47±0.28 | 90.12±1.45 | 92.20±0.26 |
| PUe | 75.50±1.85 | 74.71±0.58 | 81.82±1.80 | 95.45±1.08 | 94.90±0.45 | 89.41±3.20 |
| GLWS | 83.50±0.64 | 76.68±0.78 | 82.47±0.75 | **98.55±0.12** | **98.90±0.19** | **98.59±0.06** |
| uPU | 76.29±1.56 | 77.81±0.18 | 77.77±1.65 | 87.57±0.17 | 86.63±0.61 | 86.90±0.47 |
| uPU-c | 94.51±0.21 | 95.48±0.55 | 94.10±0.32 | 89.73±0.68 | 83.45±3.19 | 89.88±0.60 |
| nnPU | 83.01±0.49 | 77.78±1.21 | 82.43±0.66 | 97.84±0.22 | 98.78±0.14 | 97.76±0.36 |
| nnPU-c | 94.11±0.36 | 94.52±0.71 | 94.85±0.20 | 88.16±0.50 | 83.18±2.52 | 88.86±0.44 |
| nnPU-GA | 89.02±1.35 | 91.22±0.96 | 89.89±0.46 | 90.78±2.06 | 91.02±0.92 | 92.75±0.37 |
| nnPU-GA-c | 93.49±0.40 | 94.18±0.39 | 93.02±0.36 | 89.14±1.38 | 82.98±3.91 | 89.73±0.94 |
| PUSB | 81.80±1.12 | 85.07±0.28 | 84.72±1.55 | 98.04±0.28 | 97.61±0.28 | 97.33±0.42 |
| PUSB-c | **94.59±0.58** | 94.29±0.46 | 94.23±0.58 | 88.31±0.42 | 88.47±0.62 | 88.51±1.03 |
| VPU | 94.38±2.09 | 66.67±27.22 | **97.19±0.39** | 76.55±4.48 | 0.16±0.08 | 78.43±4.44 |
| VPU-c | 94.22±0.79 | **96.82±0.89** | 93.30±0.50 | 88.78±0.76 | 55.53±18.31 | 90.67±0.28 |
| Dist-PU | 80.19±0.78 | 73.27±3.53 | 79.18±1.24 | 94.35±0.22 | 92.71±1.44 | 91.41±1.93 |
| Dist-PU-c | 94.24±0.58 | 95.22±0.37 | 94.27±0.38 | 86.43±0.43 | 81.61±2.06 | 85.10±0.71 |

Table 17: Test results (mean±std) of accuracy, AUC, and F1 score for each algorithm on USPS (Case 2) under different model selection criteria. The best performance w.r.t. each validation metric is shown in bold. Here, "-c" indicates using the proposed calibration technique in Algorithm 1.

| Test metric | Test ACC | | | AUC | | | Test F1 | | |
|---|---|---|---|---|---|---|---|---|---|
| Val metric | PA | PAUC | OA | PA | PAUC | OA | PA | PAUC | OA |
| PUbN | 94.45±0.26 | **95.45±0.14** | **95.45±0.20** | 98.62±0.18 | 98.83±0.08 | 98.85±0.10 | 94.23±0.29 | **95.31±0.13** | **95.29±0.20** |
| PAN | 80.70±3.24 | 83.54±1.27 | 84.22±1.12 | 88.16±3.61 | 92.49±0.23 | 92.89±0.18 | 78.89±3.82 | 81.47±2.11 | 82.45±1.90 |
| CVIR | 90.93±0.26 | 88.57±0.29 | 90.55±0.20 | 96.74±0.23 | 96.34±0.22 | 96.69±0.21 | 91.37±0.23 | 89.34±0.24 | 91.05±0.16 |
| P3MIX-E | 94.04±0.43 | 93.90±0.43 | 93.90±0.39 | 98.26±0.27 | 98.24±0.26 | 98.14±0.19 | 93.92±0.39 | 93.79±0.38 | 93.81±0.36 |
| P3MIX-C | 94.27±0.52 | 94.54±0.51 | 94.72±0.35 | 98.38±0.24 | 98.56±0.17 | 98.66±0.13 | 94.20±0.48 | 94.47±0.48 | 94.62±0.34 |
| LBE | 94.67±0.20 | 90.82±1.28 | 94.88±0.05 | 98.51±0.16 | 98.17±0.08 | 98.60±0.07 | 94.65±0.16 | 91.18±0.98 | 94.73±0.10 |
| Count Loss | 92.73±0.22 | 93.76±0.45 | 93.17±0.14 | 97.15±0.27 | 97.72±0.20 | 97.33±0.26 | 92.87±0.16 | 93.84±0.40 | 93.18±0.14 |
| Robust-PU | 93.72±0.41 | 93.64±0.31 | 94.93±0.19 | 98.13±0.11 | 98.34±0.15 | 98.62±0.23 | 93.44±0.45 | 93.33±0.31 | 94.69±0.20 |
| Holistic-PU | 95.15±0.28 | 94.83±0.24 | 95.02±0.53 | 98.76±0.11 | 98.73±0.17 | 98.49±0.20 | 94.99±0.29 | 94.65±0.24 | 94.84±0.57 |
| PUe | 85.27±1.11 | 85.00±0.63 | 86.05±0.33 | 93.95±0.48 | 95.26±0.23 | 93.48±0.92 | 86.55±0.96 | 86.38±0.50 | 87.08±0.25 |
| GLWS | 92.48±0.50 | 88.19±0.44 | 92.18±0.26 | 98.58±0.05 | 98.09±0.29 | 98.45±0.06 | 92.76±0.44 | 89.15±0.35 | 92.49±0.23 |
| uPU | 83.36±0.48 | 82.68±0.81 | 84.12±0.06 | 92.33±0.30 | 93.99±0.25 | 92.79±0.79 | 84.51±0.42 | 84.34±0.56 | 85.14±0.27 |
| uPU-c | 94.67±0.10 | 93.36±0.76 | 94.57±0.28 | **98.78±0.10** | 98.50±0.30 | 98.68±0.11 | 94.36±0.11 | 92.85±0.88 | 94.26±0.32 |
| nnPU | 93.64±1.13 | 88.01±1.30 | 94.10±0.37 | 98.50±0.15 | 98.37±0.09 | 98.71±0.08 | 93.79±1.03 | 89.01±1.07 | 94.20±0.33 |
| nnPU-c | 94.32±0.20 | 94.00±0.24 | 94.80±0.12 | 98.69±0.02 | 98.48±0.04 | 98.67±0.04 | 94.03±0.23 | 93.67±0.27 | 94.56±0.10 |
| nnPU-GA | 94.42±0.26 | 94.95±0.13 | 94.78±0.07 | 98.61±0.10 | 98.68±0.10 | 98.44±0.12 | 94.28±0.24 | 94.76±0.14 | 94.59±0.08 |
| nnPU-GA-c | 94.12±0.04 | 94.27±0.17 | 94.07±0.36 | 98.66±0.06 | 98.79±0.07 | 98.73±0.07 | 93.77±0.04 | 93.92±0.17 | 93.69±0.40 |
| PUSB | 92.41±0.67 | 91.96±1.50 | 93.56±0.34 | 92.57±0.65 | 92.10±1.45 | 93.68±0.33 | 92.69±0.59 | 92.25±1.30 | 93.70±0.30 |
| PUSB-c | 94.09±0.18 | 93.64±0.21 | 94.57±0.25 | 94.01±0.18 | 93.55±0.22 | 94.50±0.25 | 93.76±0.19 | 93.24±0.24 | 94.27±0.26 |
| VPU | 89.82±2.61 | 76.63±10.29 | 89.54±1.88 | 97.91±0.35 | 98.58±0.10 | 97.99±0.57 | 88.33±3.43 | 58.65±23.78 | 88.11±2.41 |
| VPU-c | 94.93±0.13 | 94.82±0.12 | 95.05±0.14 | 98.78±0.06 | **98.86±0.09** | **98.91±0.11** | 94.72±0.17 | 94.55±0.16 | 94.81±0.14 |
| Dist-PU | 94.82±0.12 | 94.02±0.18 | 94.72±0.38 | 98.09±0.17 | 97.94±0.28 | 98.12±0.19 | 94.63±0.13 | 93.72±0.25 | 94.55±0.39 |
| Dist-PU-c | 94.10±0.44 | 92.09±0.50 | 94.14±0.37 | 98.49±0.16 | 98.53±0.20 | 98.50±0.03 | 93.73±0.49 | 91.30±0.59 | 93.77±0.42 |

Table 18: Test results (mean±std) of precision and recall for each algorithm on USPS (Case 2) under different model selection criteria. The best performance w.r.t. each validation metric is shown in bold. Here, "-c" indicates using the proposed calibration technique in Algorithm 1.

| Test metric | Precision | | | Recall | | |
|---|---|---|---|---|---|---|
| Val metric | PA | PAUC | OA | PA | PAUC | OA |
| PUbN | 95.38±0.94 | 95.68±0.48 | 95.97±0.59 | 93.18±1.09 | 94.95±0.27 | 94.64±0.42 |
| PAN | 83.76±3.59 | 89.42±1.97 | 89.52±2.59 | 75.13±5.70 | 75.67±5.30 | 77.35±5.13 |
| CVIR | 85.15±0.42 | 81.84±0.41 | 84.48±0.37 | 98.57±0.05 | 98.36±0.22 | 98.74±0.18 |
| P3MIX-E | 93.52±1.27 | 93.30±1.32 | 92.82±0.87 | 94.37±0.51 | 94.34±0.58 | 94.85±0.29 |
| P3MIX-C | 93.07±1.19 | 93.28±0.94 | 93.88±0.75 | 95.39±0.26 | 95.70±0.08 | 95.39±0.10 |
| LBE | 92.57±0.82 | 86.94±3.54 | 95.05±0.89 | 96.86±0.56 | 96.45±2.16 | 94.47±1.09 |
| Count Loss | 88.88±0.75 | 90.40±0.96 | 90.75±0.87 | 97.27±0.57 | 97.58±0.41 | 95.80±1.05 |
| Robust-PU | 95.00±0.46 | 95.31±0.48 | 96.73±0.26 | 91.95±0.83 | 91.44±0.17 | 92.73±0.39 |
| Holistic-PU | 95.52±0.49 | 95.53±0.63 | 95.60±0.47 | 94.47±0.42 | 93.79±0.31 | 94.10±0.92 |
| PUe | 77.96±1.20 | 77.46±0.78 | 79.31±0.72 | 97.30±0.88 | 97.65±0.21 | 96.59±0.91 |
| GLWS | 87.39±0.88 | 80.72±0.66 | 86.83±0.38 | **98.84±0.12** | **99.56±0.20** | **98.94±0.06** |
| uPU | 77.29±0.61 | 75.45±1.20 | 78.20±0.75 | 93.24±0.79 | 95.67±0.58 | 93.55±1.69 |
| uPU-c | **97.19±0.48** | 97.26±0.65 | 96.97±0.16 | 91.71±0.50 | 88.88±1.65 | 91.71±0.70 |
| nnPU | 89.86±1.86 | 80.69±1.82 | 90.47±0.71 | 98.16±0.35 | 99.32±0.07 | 98.26±0.13 |
| nnPU-c | 96.32±0.13 | 96.26±0.18 | 96.46±0.51 | 91.85±0.53 | 91.23±0.63 | 92.73±0.29 |
| nnPU-GA | 94.11±0.67 | 95.82±0.28 | 95.61±0.20 | 94.47±0.26 | 93.72±0.44 | 93.59±0.28 |
| nnPU-GA-c | 96.77±0.20 | 97.19±0.29 | 97.11±0.18 | 90.96±0.16 | 90.86±0.11 | 90.52±0.59 |
| PUSB | 87.44±1.11 | 87.68±2.47 | 89.52±0.75 | 98.64±0.10 | 97.48±0.32 | 98.29±0.27 |
| PUSB-c | 96.44±0.41 | 96.58±0.49 | 96.84±0.63 | 91.23±0.44 | 90.14±0.63 | 91.85±0.27 |
| VPU | 96.67±0.70 | **98.82±0.56** | **97.26±0.20** | 82.02±5.96 | 52.95±21.53 | 80.76±3.82 |
| VPU-c | 96.09±0.48 | 96.87±0.51 | 96.80±0.21 | 93.42±0.78 | 92.36±0.79 | 92.90±0.07 |
| Dist-PU | 95.45±0.21 | 95.77±0.70 | 95.01±0.40 | 93.82±0.18 | 91.81±1.11 | 94.10±0.50 |
| Dist-PU-c | 97.04±0.15 | 98.27±0.26 | 96.98±0.06 | 90.65±0.83 | 85.26±0.88 | 90.79±0.80 |

