# OpenReview forum: "Accessible, Realistic, and Fair Evaluation of Positive-Unlabeled Learning Algorithms"
_ICLR.cc/2026/Conference — ICLR 2026 Poster_

### Official Review · Reviewer_WGLx · 2025-10-29

**Soundness:** 2
**Presentation:** 2
**Contribution:** 1
**Rating:** 2
**Confidence:** 3

**Summary:**

This work investigates the experimental settings in PU learning.
The authors argue that using a validation set containing both positive and negative labeled samples for model selection is unrealistic, and therefore propose a proxy metric as a surrogate for this purpose.
They further discuss two sampling processes in PU learning, namely One-Sample (OS) and Two-Sample (TS), and introduce a calibration technique to mitigate the class-shift problem when transitioning from the TS to the OS setting.

**Strengths:**

1. The proposed proxy metric is useful for model selection in cases where a validation set with both positive and negative samples cannot be obtained.

**Weaknesses:**

1. The authors discuss two data generation processes, namely One-Sample and Two-Sample. However, it remains unclear why they do not consider benchmark settings with varying ratios of positive to unlabeled data, or different proportions of positive samples within the unlabeled set. By controlling the ratios, we can validate PU algorithms across varying settings to demonstrate their robustness.
2. Regarding the proposed PA, although the authors argue that access to a fully labeled validation dataset is unrealistic, their assumption of knowing the class prior probability $\\pi$ is equally impractical.
3. Although the authors theoretically prove that the proposed proxy metric preserves the partial order of the original metric, they provide no experimental evidence that the model selected by this proxy coincides with the one chosen using a validation set containing both positive and negative samples.
4. The introduction of the calibration technique for the OS setting seems redundant, since the OS setting can be straightforwardly transformed into a TS setting by redefining the unlabeled training data to combine the original positive and unlabeled sets. Essentially, the proposed calibration achieves the same effect implicitly, as demonstrated below.

We start from Eq. 8:
$$
R = \\frac{\\pi}{n_p} \\sum_{i=1}^{n_p} ( \\ell( f(x_i), +1 ) + (c-1) \\ell (f(x_i), -1)) + \\frac{1 - c\\pi}{n_u} \\sum_{j=1}^{n_u}\\ell(f(x_i), -1).
$$
This can be rewritten as
$$
R = \\frac{\\pi}{n_p} \\sum_{i=1}^{n_p} \\ell( f(x_i), +1)- \\frac{\\pi}{n_p} \\sum_{i=1}^{n_p} \\ell (f(x_i), -1) + \\frac{\\pi}{n_p} c \\sum_{i=1}^{n_p} \\ell (f(x_i), -1) + \\frac{1 - c\\pi}{n_u} \\sum_{j=1}^{n_u} \\ell(f(x_i), -1).
$$
Substituting $c = n_p / (\\pi (n_p + n_u))$ gives
$$
R = \\frac{\\pi}{n_p} \\sum_{i=1}^{n_p} \\ell( f(x_i), +1) - \\frac{\\pi}{n_p} \\sum_{i=1}^{n_p} \\ell (f(x_i), -1) + \\frac{1}{n_p + n_u} \\sum_{i=1}^{n_p} \\ell (f(x_i), -1) + \\frac{1}{n_p + n_u} \\sum_{j=1}^{n_u} \\ell(f(x_i), -1).
$$
Thus,
$$
R = \\frac{\\pi}{n_p} \\sum_{i=1}^{n_p} \\ell( f(x_i), +1) - \\frac{\\pi}{n_p} \\sum_{i=1}^{n_p} \\ell (f(x_i), -1) + \\frac{1}{n_p + n_u} \\sum_{i=1}^{n_u + n_p} \\ell(f(x_i), -1)
$$
This derivation shows that the proposed calibration is equivalent to simply merging the positive and unlabeled sets into a single new unlabeled set.

**Questions:**

See weaknesses.

---

> ### Author Response · Authors · 2025-11-24
> **Response to Reviewer WGLx (1/4)**
>
> First of all, we would like to thank you for taking the time to review our paper. Below are our answers (A) to the weaknesses (W).
>
> **W1: Experiments with varying ratios of positive to unlabeled data.**
>
> **A1:** We would like to point out that we **did include some experimental results of two-sample (TS) approaches with varying numbers of positive data** in the original version of our paper. Figures 2 and 3 show the experimental results on CIFAR-10 and ImageNet with varying positive data ratios, respectively. Based on these results, we observe that TS methods perform worse even with more positive data. However, the performance of the proposed calibration approaches remains stable, which validates their effectiveness.
>
> To show the effect of varying positive data ratios on different PU learning algorithms comprehensively, we conducted experiments on the USPS and Letter datasets. The full set of experimental results for various test metrics can be found in Appendix D. Below, we list some of the experimental results for test accuracy for illustration purposes.
>
> Experimental results on the Letter dataset:
>
> | Ratio | 10%    | 20%        | 30%        | 40%        | 50%        |
> | ------------------- | --------------------- | --------------------- | --------------------- | --------------------- | --------------------- |
> | PUbN                | 61.77  ± 10.4         | 76.47  ± 10.7         | 89.70  ± 1.38         | 62.75  ± 11.2         | 78.57  ± 11.5         |
> | PAN                 | 48.30  ± 0.91         | 48.30  ± 0.91         | 52.18  ± 1.24         | 48.30  ± 0.91         | 48.30  ± 0.91         |
> | CVIR                | 82.63  ± 0.86         | 83.55  ± 0.39         | 84.67  ± 0.58         | 79.90  ± 0.64         | 74.35  ± 0.27         |
> | P3MIX-E             | 49.43  ± 0.21         | 49.43  ± 0.21         | 61.42  ± 4.12         | 49.43  ± 0.21         | 49.43  ± 0.21         |
> | P3MIX-C             | 75.43  ± 1.40         | 76.02  ± 1.25         | 80.92  ± 1.14         | 80.87  ± 0.33         | 81.68  ± 0.04         |
> | LBE                 | 80.07  ± 0.47         | 81.82  ± 1.01         | 87.55  ± 0.28         | 84.18  ± 0.19         | 86.42  ± 0.72         |
> | Count Loss          | 73.07  ± 4.15         | 70.78  ± 5.47         | 78.27  ± 1.01         | 56.83  ± 2.84         | 55.90  ± 2.53         |
> | Robust-PU           | 84.50  ± 0.66         | 89.08  ± 1.14         | 90.63  ± 0.31         | 92.77  ± 0.15         | 93.98  ± 0.42         |
> | Holistic-PU         | 80.80  ± 0.43         | 82.90  ± 0.66         | 87.32  ± 1.27         | 85.37  ± 0.41         | 86.88  ± 0.30         |
> | PUe                 | 82.13  ± 0.59         | 81.18  ± 1.02         | 82.00  ± 0.78         | 76.98  ± 0.44         | 74.72  ± 0.46         |
> | GLWS                | 85.53  ± 0.46         | 86.60  ± 0.25         | 86.32  ± 0.58         | 82.15  ± 0.22         | 77.80  ± 0.32         |
> | uPU                 | 81.22  ± 0.76         | 80.12  ± 0.48         | 77.72  ± 0.79         | 75.30  ± 0.74         | 72.52  ± 0.55         |
> | uPU-c               | 86.17  ± 0.53         | 89.55  ± 0.45         | **92.73**  ± **0.15** | 93.33  ± 0.41         | 94.53  ± 0.41         |
> | nnPU                | **86.57**  ± **0.28** | 88.55  ± 0.09         | 85.60  ± 0.31         | 80.40  ± 0.43         | 76.93  ± 0.57         |
> | nnPU-c              | 86.08  ± 0.36         | **90.38**  ± **0.25** | 91.82  ± 0.14         | **93.60**  ± **0.22** | **94.72**  ± **0.15** |
> | nnPU-GA             | 82.12  ± 0.50         | 85.78  ± 0.34         | 84.27  ± 0.58         | 84.90  ± 1.10         | 84.38  ± 0.33         |
> | nnPU-GA-c           | 85.12  ± 0.24         | 89.65  ± 0.27         | 90.97  ± 0.30         | 93.32  ± 0.28         | 94.38  ± 0.16         |
> | PUSB                | 85.40  ± 0.72         | 87.68  ± 0.22         | 86.82  ± 0.54         | 80.00  ± 0.95         | 74.08  ± 0.62         |
> | PUSB-c              | 85.23  ± 0.45         | 89.60  ± 0.55         | 91.43  ± 0.92         | 93.40  ± 0.08         | 94.35  ± 0.06         |
> | VPU                 | 79.87  ± 0.55         | 86.22  ± 0.43         | 90.13  ± 0.77         | 88.12  ± 0.19         | 68.90  ± 1.88         |
> | VPU-c               | 83.78  ± 0.74         | 89.67  ± 0.59         | 92.15  ± 0.52         | 93.13  ± 0.38         | 94.52  ± 0.42         |
> | Dist-PU             | 47.97  ± 0.63         | 47.97  ± 0.63         | 77.55  ± 0.78         | 47.97  ± 0.63         | 47.97  ± 0.63         |
> | Dist-PU-c           | 47.97  ± 0.63         | 47.97  ± 0.63         | 70.03  ± 2.28         | 47.97  ± 0.63         | 47.97  ± 0.63         |

---

> ### Author Response · Authors · 2025-11-24
> **Response to Reviewer WGLx (2/4)**
>
> Experimental results on the USPS dataset:
>
> | Ratio | 10%    | 20%        | 30%        | 40%        | 50%        |
> | ------------------- | -------------------- | -------------------- | -------------------- | -------------------- | -------------------- |
> | PUbN                | 90.68 ± 0.40         | **92.92** ± **0.15** | **93.95** ± **0.13** | 94.20 ± 0.25         | 94.88 ± 0.11         |
> | PAN                 | 85.60 ± 0.26         | 85.57 ± 0.70         | 84.97 ± 0.48         | 79.59 ± 0.84         | 56.34 ± 0.42         |
> | CVIR                | **92.31** ± **0.28** | 84.60 ± 3.01         | 82.98 ± 1.39         | 79.36 ± 0.17         | 73.64 ± 0.09         |
> | P3MIX-E             | 88.36 ± 0.41         | 88.77 ± 0.14         | 89.84 ± 1.17         | 89.54 ± 0.05         | 90.43 ± 0.08         |
> | P3MIX-C             | 91.20 ± 0.08         | 91.03 ± 0.02         | 93.22 ± 0.31         | 91.46 ± 0.23         | 92.01 ± 0.10         |
> | LBE                 | 89.97 ± 0.54         | 91.03 ± 0.11         | 92.29 ± 0.33         | 92.92 ± 0.14         | 94.30 ± 0.15         |
> | Count Loss          | 91.18 ± 0.00         | 92.36 ± 0.26         | 91.76 ± 0.81         | 90.23 ± 0.22         | 86.21 ± 0.76         |
> | Robust-PU           | 89.19 ± 0.27         | 91.41 ± 0.36         | 92.79 ± 0.12         | **94.49** ± **0.05** | **95.42** ± **0.22** |
> | Holistic-PU         | 88.61 ± 0.24         | 92.13 ± 0.27         | 93.46 ± 0.36         | 93.81 ± 0.07         | 93.34 ± 0.14         |
> | PUe                 | 87.91 ± 0.54         | 87.11 ± 0.33         | 86.93 ± 0.27         | 79.94 ± 1.16         | 78.39 ± 0.79         |
> | GLWS                | 91.65 ± 0.44         | 91.60 ± 0.50         | 90.52 ± 0.47         | 83.04 ± 0.84         | 81.02 ± 0.08         |
> | uPU                 | 87.10 ± 0.67         | 86.55 ± 1.07         | 83.86 ± 0.83         | 80.25 ± 0.92         | 77.01 ± 0.50         |
> | uPU-c               | 89.01 ± 0.32         | 90.95 ± 0.33         | 93.32 ± 0.10         | 94.15 ± 0.13         | 95.13 ± 0.09         |
> | nnPU                | 91.38 ± 0.27         | 90.96 ± 0.64         | 90.22 ± 0.42         | 71.98 ± 1.90         | 49.51 ± 0.97         |
> | nnPU-c              | 89.24 ± 0.20         | 91.50 ± 0.21         | 93.24 ± 0.18         | 94.17 ± 0.19         | 95.20 ± 0.11         |
> | nnPU-GA             | 89.94 ± 0.31         | 91.43 ± 0.38         | 92.51 ± 0.31         | 92.87 ± 0.24         | 93.59 ± 0.25         |
> | nnPU-GA-c           | 88.89 ± 0.36         | 90.25 ± 0.25         | 92.79 ± 0.22         | 93.90 ± 0.18         | 95.35 ± 0.03         |
> | PUSB                | 89.92 ± 0.09         | 90.80 ± 0.26         | 91.38 ± 0.83         | 72.46 ± 0.55         | 53.84 ± 1.48         |
> | PUSB-c              | 88.76 ± 0.47         | 92.11 ± 0.32         | 92.83 ± 0.18         | 93.89 ± 0.24         | 94.92 ± 0.18         |
> | VPU                 | 82.46 ± 1.27         | 82.36 ± 0.40         | 89.89 ± 1.71         | 82.00 ± 1.77         | 87.06 ± 1.14         |
> | VPU-c               | 85.24 ± 1.38         | 91.10 ± 0.90         | 93.29 ± 0.32         | 94.47 ± 0.14         | 95.20 ± 0.15         |
> | Dist-PU             | 87.36 ± 1.58         | 88.74 ± 1.65         | 86.10 ± 0.14         | 84.16 ± 0.67         | 85.07 ± 0.04         |
> | Dist-PU-c           | 89.14 ± 0.15         | 90.27 ± 0.37         | 91.50 ± 0.34         | 93.44 ± 0.17         | 94.04 ± 0.20         |
>
> We observe that the performance of many two-sample methods deteriorates significantly, even with more positive data. The proposed calibration technique effectively improves their robustness, and their performance increases with more positive data.
>
> **W2: Assuming knowledge of $\pi$ for PA is equally impractical as having no negative data.**
>
> **A2:** Thank you for pointing out a limitation of PA. However, **reliance on $\pi$ is an intrinsic and common issue in the original PU learning problem** and is not the focus of our paper. This paper focuses on evaluating PU learning and does not specifically address the class prior estimation problem in PU learning. Therefore, this is not a problem that we introduced, but rather a common issue in PU learning literature that is beyond the scope of our paper.
>
> Besides, we respectfully believe that **knowing $\pi$ is more practical than having negative data**. It is because obtaining negative data is impossible in the standard PU learning setting, but class prior estimation is still possible. First, theoretically, estimating $\pi$ is possible, and many algorithms have been proposed to improve the accuracy of class prior estimation [1-6]. Under mild assumptions, it is often proven that the estimated $\pi$ can converge to the true class prior for these algorithms. Additionally, many state-of-the-art PU learning algorithms rely on knowledge of $\pi$, and we can use their estimated $\pi$ for PA calculations directly.

---

> ### Author Response · Authors · 2025-11-24
> **Response to Reviewer WGLx (3/4)**
>
> Second, we can usually obtain $\pi$ from other sources free of charge. In some real-world applications, proportions of different classes can be collected. For example, governments often publish economic and social science statistics [7], and hospitals frequently publish disease incidence rates on their websites. This means that we can obtain such statistics for specific real-world applications (see Section 12.1 in [7]). Third, even if we do not know $\pi$, we can ask domain experts to estimate the class prior, which is **much cheaper than annotating labels for an unlabeled dataset** [7].
>
> Furthermore, we demonstrate through additional experiments that **PA can still work effectively with inaccurate class prior estimations**. The experimental results in terms of test accuracy on the Letter dataset are shown here, and the full results can be found in Appendix E.
>
> | Validation Metric |PA   | PAUC  | OA   |
> | ----------- | ---------------- | ---------------- | ---------------- |
> | PUbN        | 76.78 ± 10.83    | 77.23 ± 11.02    | 77.65 ± 11.19    |
> | PAN         | 48.30 ± 0.91     | 48.30 ± 0.91     | 48.30 ± 0.91     |
> | CVIR        | 81.82 ± 0.22     | 81.30 ± 0.43     | 82.23 ± 0.55     |
> | P3MIX-E     | 49.43 ± 0.21     | 49.43 ± 0.21     | 49.43 ± 0.21     |
> | P3MIX-C     | 76.80 ± 2.47     | 77.25 ± 2.39     | 78.05 ± 1.51     |
> | LBE         | 81.35 ± 0.44     | 76.83 ± 2.00     | 83.98 ± 0.25     |
> | Count Loss  | 63.07 ± 4.67     | 63.07 ± 4.67     | 62.50 ± 4.23     |
> | Robust-PU   | 91.17 ± 0.54     | 89.82 ± 0.04     | 90.97 ± 0.47     |
> | Holistic-PU | 83.87 ± 0.82     | 78.68 ± 3.95     | 84.35 ± 0.46     |
> | PUe         | 74.93 ± 1.18     | 76.20 ± 1.72     | 78.45 ± 0.67     |
> | GLWS        | 85.05 ± 0.62     | 81.90 ± 0.92     | 85.50 ± 0.26     |
> | uPU         | 76.60 ± 0.99     | 76.82 ± 1.98     | 78.07 ± 0.64     |
> | uPU-c       | 91.98 ± 0.31 | 90.90 ± 0.57     | 91.78 ± 0.46     |
> | nnPU        | 86.12 ± 0.31     | 76.33 ± 3.01     | 87.28 ± 0.50     |
> | nnPU-c      | 91.97 ± 0.27     | 90.43 ± 0.85     | 92.05 ± 0.19     |
> | nnPU-GA     | 84.67 ± 0.92     | 83.87 ± 0.47     | 85.98 ± 0.48     |
> | nnPU-GA-c   | 90.98 ± 0.29     | 87.10 ± 0.93     | 90.98 ± 0.29     |
> | PUSB        | 86.08 ± 0.51     | 86.08 ± 0.51     | 85.73 ± 0.77     |
> | PUSB-c      | 91.73 ± 0.22     | 91.08 ± 0.60 | 92.17 ± 0.28 |
> | VPU         | 87.07 ± 0.60     | 66.03 ± 2.83     | 88.85 ± 0.52     |
> | VPU-c       | 91.38 ± 0.33     | 87.83 ± 2.25     | 91.93 ± 0.44     |
> | Dist-PU     | 47.97 ± 0.63     | 47.97 ± 0.63     | 47.97 ± 0.63     |
> | Dist-PU-c   | 47.97 ± 0.63     | 47.97 ± 0.63     | 47.97 ± 0.63     |
>
> Here are the experimental results of test accuracy on the USPS dataset:
>
> | Validation Metric |PA   | PAUC  | OA   |
> | ----------- | ---------------- | ---------------- | ---------------- |
> | PUbN        | 93.72 ± 0.25 | 93.66 ± 0.35 | 93.90 ± 0.18 |
> | PAN         | 85.70 ± 0.15     | 83.89 ± 0.28     | 85.68 ± 0.14     |
> | CVIR        | 81.12 ± 0.22     | 81.03 ± 0.24     | 81.17 ± 0.18     |
> | P3MIX-E     | 88.77 ± 0.28     | 89.27 ± 0.15     | 88.94 ± 0.16     |
> | P3MIX-C     | 91.26 ± 0.07     | 91.35 ± 0.12     | 91.41 ± 0.11     |
> | LBE         | 90.77 ± 0.20     | 91.93 ± 0.45     | 92.01 ± 0.46     |
> | Count Loss  | 91.76 ± 0.66     | 91.58 ± 0.40     | 92.14 ± 0.34     |
> | Robust-PU   | 92.99 ± 0.21     | 92.81 ± 0.29     | 93.07 ± 0.19     |
> | Holistic-PU | 93.29 ± 0.24     | 93.16 ± 0.10     | 93.24 ± 0.23     |
> | PUe         | 84.55 ± 0.31     | 84.35 ± 0.71     | 84.84 ± 0.39     |
> | GLWS        | 88.82 ± 0.39     | 87.78 ± 0.44     | 88.39 ± 0.24     |
> | uPU         | 82.74 ± 0.60     | 83.99 ± 0.49     | 83.56 ± 0.84     |
> | uPU-c       | 92.64 ± 0.46     | 92.23 ± 0.04     | 93.16 ± 0.32     |
> | nnPU        | 85.50 ± 0.38     | 80.15 ± 0.58     | 84.80 ± 0.63     |
> | nnPU-c      | 92.73 ± 0.05     | 91.93 ± 0.14     | 93.14 ± 0.27     |
> | nnPU-GA     | 92.56 ± 0.61     | 91.50 ± 0.44     | 92.79 ± 0.12     |
> | nnPU-GA-c   | 92.18 ± 0.35     | 92.14 ± 0.63     | 92.63 ± 0.21     |
> | PUSB        | 84.97 ± 1.26     | 84.97 ± 1.26     | 85.19 ± 1.09     |
> | PUSB-c      | 92.79 ± 0.40     | 92.58 ± 0.18     | 92.87 ± 0.12     |
> | VPU         | 83.74 ± 1.41     | 60.62 ± 2.37     | 83.74 ± 1.41     |
> | VPU-c       | 93.56 ± 0.29     | 82.03 ± 8.04     | 93.36 ± 0.29     |
> | Dist-PU     | 86.31 ± 0.07     | 83.62 ± 0.93     | 86.26 ± 0.21     |
> | Dist-PU-c   | 92.01 ± 0.42     | 90.90 ± 0.54     | 91.94 ± 0.46     |
>
> As we can see, the model selected using PA still performs quite strongly compared with the other two metrics. This validates the effectiveness of PA, even when class priors are estimated incorrectly. For these reasons, we emphasize that knowledge of class prior is more practical than negative data, which is impossible to obtain in the standard PU learning setting. We have added the discussion to the revised version of the paper.

---

> ### Author Response · Authors · 2025-11-24
> **Response to Reviewer WGLx (4/4)**
>
> **W3: There are no experiments about the effectiveness of proxy metrics.**
>
> **A3:** The effectiveness of the PA and PAUC metrics can be compared by checking the tables horizontally. This can be done for each row in Tables 1, 2, and 5 through 18. We observe that models selected using PA or PAUC often achieve comparable performance to those selected using OA. This empirically validates the effectiveness of the proxy metrics. While the model selected by PA and PAUC can sometimes perform worse than the model selected by OA, it's important to note that **OA is not calculable in real-world PU learning scenarios and should be considered an unrealizable upper bound for model selection**. Since the performance of the model selected using our proposed proxy metrics is often similar to the model selected using OA, we believe that the proxy metrics are effective and practical for real-world applications.
>
> **W4: The proposed calibration technique is equivalent to merging positive and unlabeled data.**
>
> **A4:** Your derivations are correct and show exactly what we propose in this paper (see Appendix A.6). Although the idea is concise, its novelty and significance lie in the following: First, our main goal is to **take the first step toward revealing that one-sample and two-sample settings and algorithms are completely different**. This issue has been overlooked in PU learning literature and was discovered by us **for the first time**. Second, as far as we know, the merging strategy has not been proposed in PU learning literature before. Although the technique is simple, it is useful for adapting two-sample algorithms to one-sample settings. **This adaptation is also original in the PU learning literature**. Third, while the merging technique is simple, it has been unknown why it works. This paper provides a **solid theoretical analysis** of its soundness, which is validated by extensive experimental results. Therefore, we believe that the proposed calibration technique is a valuable addition to PU learning literature, particularly with regard to ensuring the fairness of algorithmic evaluation.
>
> References:
>
> [1] Classification with Asymmetric Label Noise: Consistency and Maximal Denoising, COLT 2013.
>
> [2] Classification with Noisy Labels by Importance Reweighting, TPAMI 2016.
>
> [3] Mixture Proportion Estimation via Kernel Embedding of Distributions, ICML 2016.
>
> [4] Mixture Proportion Estimation and PU Learning: A Modern Approach, NeurIPS 2021.
>
> [5] Rethinking Class-Prior Estimation for Positive-Unlabeled Learning, ICLR 2022.
>
> [6] Mixture Proportion Estimation Beyond Irreducibility, ICML 2023.
>
> [7] Machine Learning from Weak Supervision: An Empirical Risk Minimization Approach, The MIT Press, 2022.

---

> ### Author Response · Authors · 2025-11-28
> **Summary of the Response to Reviewer WGLx**
>
> Dear Reviewer WGLx,
>
> Thank you very much for taking the time to review our paper. We have posted a detailed rebuttal that addresses your comments and includes additional clarifications and new experiments.
>
> In our rebuttal, we attempted to address your **misunderstandings** and concerns about our paper. These may have included the experiments of varying numbers of positive data that we **did include in the initial version**, the **practicality** of knowing class priors over having negative data in the standard PU learning setting, the scope of the paper, and some details of methodologies and experiments. We believe that our rebuttal adequately addresses your concerns. If you have any other concerns, please do not hesitate to contact us.
>
> Thanks again for your help in reviewing our submission.
>
> Best,
>
> Author of ICLR 2026 Conference Submission 9413

---

### Official Review · Reviewer_BgWp · 2025-10-31

**Soundness:** 3
**Presentation:** 3
**Contribution:** 3
**Rating:** 8
**Confidence:** 3

**Summary:**

This paper establish the first comprehensive benchmark for evaluating PU learning algorithms. Through extensive experiment with many state-of-the-art algorithms, the authors identify the internal label shift problem between OS and TS PU learning settings. The proposed model selection criteria doesn't rely on negative validation data, which contradicts PU learning's core premise. In addition, the authors proposed a calibration methods with theoretical guarantees to address the label shift issue.

**Strengths:**

The authors conducted a thorough evaluation using four datasets with systematic data generation and consistent evaluation protocols. Three criteria are compared for model selection purposes. The experimental framework is particularly well-designed, addressing the unrealistic assumption that many existing evaluations make about having access to negative validation data.

**Weaknesses:**

1. The authors use relatively small benchmark dataset, which may not reflect large-scale performance.
2. Deeper exploration on how estimation errors in class prior affect performance is not presented.

**Questions:**

See weaknesses section.

---

> ### Author Response · Authors · 2025-11-25
> **Response to Reviewer BgWp (1/2)**
>
> First, we would like to express our sincere gratitude for your time and efforts in reviewing our submission. We are very grateful that you agree with the contributions of the paper. Below are our answers (A) to the weaknesses (W).
>
> **W1: Lack of large-scale datasets.**
>
> **A1:** In this paper, we follow existing PU learning evaluation protocols and primarily use synthetic benchmark datasets to compare the performance of different algorithms. Since open-source large-scale PU datasets are lacking in the literature, we consider collecting large-scale real-world PU datasets to be very promising future work.
>
> To test the performance of algorithms on off-the-shelf real-world datasets, we conducted additional experiments on a real-world benchmark dataset for the Kaggle Credit Card Fraud Detection task [1], which is also used in PU learning literature[2,3]. We followed the previous experimental setup and used the MLP as the model architecture. The other settings are the same as those of the tabular datasets in the paper. Below are parts of the accuracy results. The full results can be found in Appendix F of the revised version of the paper.
>
> | Val Metric | PA   | PAUC  | OA   |
> | ----------- | ---------------- | ---------------- | ---------------- |
> | PUbN        | 96.31 ± 2.01     | 90.02 ± 5.66     | 97.13 ± 0.89     |
> | PAN         | 94.54 ± 2.90     | 19.44 ± 7.80     | 94.09 ± 1.58     |
> | CVIR        | 98.69 ± 0.95     | 99.61 ± 0.20     | 99.88 ± 0.04     |
> | P3MIX-E     | 98.38 ± 0.45     | 96.38 ± 1.55     | 98.21 ± 1.09     |
> | P3MIX-C     | 99.07 ± 0.58     | 98.71 ± 0.48     | 97.26 ± 1.13     |
> | LBE         | 90.96 ± 3.02     | 83.66 ± 8.10     | 95.02 ± 1.73     |
> | Count Loss  | 90.46 ± 2.26     | 94.78 ± 2.16     | 94.82 ± 0.95     |
> | Robust-PU   | 92.51 ± 2.86     | 80.53 ± 7.39     | 94.88 ± 1.73     |
> | Holistic-PU | 90.11 ± 0.30     | 85.14 ± 2.38     | 90.96 ± 1.49     |
> | PUe         | 97.21 ± 1.51     | 74.09 ± 20.61    | 98.49 ± 0.46     |
> | GLWS        | 99.20 ± 0.53     | 99.81 ± 0.05     | 99.23 ± 0.55     |
> | uPU         | 96.97 ± 1.72     | 98.12 ± 0.88     | 99.12 ± 0.20     |
> | uPU-c       | 95.46 ± 0.73     | 88.54 ± 4.56     | 93.80 ± 0.71     |
> | nnPU        | 98.98 ± 0.61     | **99.92 ± 0.01** | 99.89 ± 0.02     |
> | nnPU-c      | 92.96 ± 1.83     | 92.44 ± 3.60     | 94.99 ± 0.09     |
> | nnPU-GA     | 88.02 ± 6.19     | 78.64 ± 3.11     | 95.33 ± 2.10     |
> | nnPU-GA-c   | 90.35 ± 4.04     | 83.02 ± 6.83     | 92.74 ± 0.14     |
> | PUSB        | 99.00 ± 0.56     | 99.00 ± 0.56     | 99.03 ± 0.74     |
> | PUSB-c      | 94.41 ± 0.81     | 90.41 ± 2.41     | 96.02 ± 0.50     |
> | Dist-PU     | **99.94 ± 0.01** | **99.92 ± 0.00** | **99.93 ± 0.01** |
> | Dist-PU-c   | 99.59 ± 0.29     | 99.16 ± 0.62     | 99.58 ± 0.28     |
>
> In the future, we will consider collecting more large-scale, real-world datasets to facilitate PU learning benchmarking.

---

> > ### Author Response · Authors · 2025-11-25
> > **Response to Reviewer BgWp (2/2)**
> >
> > **W2: Deeper explorations of the effect of class priors are lacking.**
> >
> > **A2:** This paper focuses on evaluating PU learning algorithms and does not specifically address the problem of estimating class priors in PU learning. We will consider evaluating the performance of class prior estimation methods in future work.
> >
> > Since the proposed PA metric uses the class prior, we conducted additional experiments on its effect on model selection. The results for the Letter dataset are shown here; the full results can be found in Appendix E.
> >
> > | Validation Metric |PA   | PAUC  | OA   |
> > | ----------- | ---------------- | ---------------- | ---------------- |
> > | PUbN        | 76.78 ± 10.83    | 77.23 ± 11.02    | 77.65 ± 11.19    |
> > | PAN         | 48.30 ± 0.91     | 48.30 ± 0.91     | 48.30 ± 0.91     |
> > | CVIR        | 81.82 ± 0.22     | 81.30 ± 0.43     | 82.23 ± 0.55     |
> > | P3MIX-E     | 49.43 ± 0.21     | 49.43 ± 0.21     | 49.43 ± 0.21     |
> > | P3MIX-C     | 76.80 ± 2.47     | 77.25 ± 2.39     | 78.05 ± 1.51     |
> > | LBE         | 81.35 ± 0.44     | 76.83 ± 2.00     | 83.98 ± 0.25     |
> > | Count Loss  | 63.07 ± 4.67     | 63.07 ± 4.67     | 62.50 ± 4.23     |
> > | Robust-PU   | 91.17 ± 0.54     | 89.82 ± 0.04     | 90.97 ± 0.47     |
> > | Holistic-PU | 83.87 ± 0.82     | 78.68 ± 3.95     | 84.35 ± 0.46     |
> > | PUe         | 74.93 ± 1.18     | 76.20 ± 1.72     | 78.45 ± 0.67     |
> > | GLWS        | 85.05 ± 0.62     | 81.90 ± 0.92     | 85.50 ± 0.26     |
> > | uPU         | 76.60 ± 0.99     | 76.82 ± 1.98     | 78.07 ± 0.64     |
> > | uPU-c       | 91.98 ± 0.31 | 90.90 ± 0.57     | 91.78 ± 0.46     |
> > | nnPU        | 86.12 ± 0.31     | 76.33 ± 3.01     | 87.28 ± 0.50     |
> > | nnPU-c      | 91.97 ± 0.27     | 90.43 ± 0.85     | 92.05 ± 0.19     |
> > | nnPU-GA     | 84.67 ± 0.92     | 83.87 ± 0.47     | 85.98 ± 0.48     |
> > | nnPU-GA-c   | 90.98 ± 0.29     | 87.10 ± 0.93     | 90.98 ± 0.29     |
> > | PUSB        | 86.08 ± 0.51     | 86.08 ± 0.51     | 85.73 ± 0.77     |
> > | PUSB-c      | 91.73 ± 0.22     | 91.08 ± 0.60 | 92.17 ± 0.28 |
> > | VPU         | 87.07 ± 0.60     | 66.03 ± 2.83     | 88.85 ± 0.52     |
> > | VPU-c       | 91.38 ± 0.33     | 87.83 ± 2.25     | 91.93 ± 0.44     |
> > | Dist-PU     | 47.97 ± 0.63     | 47.97 ± 0.63     | 47.97 ± 0.63     |
> > | Dist-PU-c   | 47.97 ± 0.63     | 47.97 ± 0.63     | 47.97 ± 0.63     |
> >
> >
> > Here are the experimental results on the USPS dataset:
> >
> > | Validation Metric |PA   | PAUC  | OA   |
> > | ----------- | ---------------- | ---------------- | ---------------- |
> > | PUbN        | 93.72 ± 0.25 | 93.66 ± 0.35 | 93.90 ± 0.18 |
> > | PAN         | 85.70 ± 0.15     | 83.89 ± 0.28     | 85.68 ± 0.14     |
> > | CVIR        | 81.12 ± 0.22     | 81.03 ± 0.24     | 81.17 ± 0.18     |
> > | P3MIX-E     | 88.77 ± 0.28     | 89.27 ± 0.15     | 88.94 ± 0.16     |
> > | P3MIX-C     | 91.26 ± 0.07     | 91.35 ± 0.12     | 91.41 ± 0.11     |
> > | LBE         | 90.77 ± 0.20     | 91.93 ± 0.45     | 92.01 ± 0.46     |
> > | Count Loss  | 91.76 ± 0.66     | 91.58 ± 0.40     | 92.14 ± 0.34     |
> > | Robust-PU   | 92.99 ± 0.21     | 92.81 ± 0.29     | 93.07 ± 0.19     |
> > | Holistic-PU | 93.29 ± 0.24     | 93.16 ± 0.10     | 93.24 ± 0.23     |
> > | PUe         | 84.55 ± 0.31     | 84.35 ± 0.71     | 84.84 ± 0.39     |
> > | GLWS        | 88.82 ± 0.39     | 87.78 ± 0.44     | 88.39 ± 0.24     |
> > | uPU         | 82.74 ± 0.60     | 83.99 ± 0.49     | 83.56 ± 0.84     |
> > | uPU-c       | 92.64 ± 0.46     | 92.23 ± 0.04     | 93.16 ± 0.32     |
> > | nnPU        | 85.50 ± 0.38     | 80.15 ± 0.58     | 84.80 ± 0.63     |
> > | nnPU-c      | 92.73 ± 0.05     | 91.93 ± 0.14     | 93.14 ± 0.27     |
> > | nnPU-GA     | 92.56 ± 0.61     | 91.50 ± 0.44     | 92.79 ± 0.12     |
> > | nnPU-GA-c   | 92.18 ± 0.35     | 92.14 ± 0.63     | 92.63 ± 0.21     |
> > | PUSB        | 84.97 ± 1.26     | 84.97 ± 1.26     | 85.19 ± 1.09     |
> > | PUSB-c      | 92.79 ± 0.40     | 92.58 ± 0.18     | 92.87 ± 0.12     |
> > | VPU         | 83.74 ± 1.41     | 60.62 ± 2.37     | 83.74 ± 1.41     |
> > | VPU-c       | 93.56 ± 0.29     | 82.03 ± 8.04     | 93.36 ± 0.29     |
> > | Dist-PU     | 86.31 ± 0.07     | 83.62 ± 0.93     | 86.26 ± 0.21     |
> > | Dist-PU-c   | 92.01 ± 0.42     | 90.90 ± 0.54     | 91.94 ± 0.46     |
> >
> > As we can see, the model selected using PA still performs quite strongly compared with the other two metrics. This validates the effectiveness of PA, even when class priors are estimated incorrectly.
> >
> > References:
> >
> > [1] https://www.kaggle.com/datasets/mlg-ulb/creditcardfraud
> >
> > [2] Who Is Your Right Mixup Partner in Positive and Unlabeled Learning? ICLR 2022.
> >
> > [3] Beyond Myopia: Learning from Positive and Unlabeled Data through Holistic Predictive Trends, NeurIPS 2023.

---

### Official Review · Reviewer_c2Ap · 2025-11-01

**Soundness:** 4
**Presentation:** 3
**Contribution:** 3
**Rating:** 8
**Confidence:** 5

**Summary:**

This paper focuses on empirical studies on PU learning methods. It systematically investigate model selection criteria to reduce the dependence on validation set, and designs a simple approach to ensure fair comparisons within and across one-sample and two-sample settings.

**Strengths:**

1. This paper provides a realistic and fair evaluation framework of PU Learning, including model selection criteria and evaluation protocols. The framework is important to identify the effectiveness of components within existing methods and will promote the development of PU Learning.
2. The paper is well-structured and written in a fluent manner, making it easy to follow and understand.
3. The analysis is thorough and the experiments are comprehensive.

**Weaknesses:**

1. Some evaluations over real-world datasets (e.g., from medical imaging, fraud detection, or recommender systems) are expected to sufficiently demonstrate the benchmark's effectiveness on larger-scale, more complex, or truly real-world PU datasets.
2. The proposed calibration approach for the Internal Label Shift (ILS) problem is presented as "simple yet effective." However, the paper lacks a discussion of its potential limitations.

**Questions:**

Please refer to Weaknesses.

---

> ### Author Response · Authors · 2025-11-25
> **Response to Reviewer c2Ap**
>
> First of all, we greatly appreciate your time and efforts in reviewing our paper. We am encouraged and thankful that you agreed with the contributions of the paper. Below are our answers (A) to the weaknesses (W).
>
> **W1: Benchmark results on real-world datasets are lacking.**
>
> **A1:** We conducted additional experiments on a real-world benchmark dataset for the Kaggle Credit Card Fraud Detection task [1], which is also used in PU learning literature[2,3]. We followed the previous experimental setup and used the MLP as the model architecture. The other settings are the same as those of the tabular datasets in the paper. Below are parts of the accuracy results. The full results can be found in Appendix F of the revised version of the paper.
>
> | Val Metric | PA   | PAUC  | OA   |
> | ----------- | ---------------- | ---------------- | ---------------- |
> | PUbN        | 96.31 ± 2.01     | 90.02 ± 5.66     | 97.13 ± 0.89     |
> | PAN         | 94.54 ± 2.90     | 19.44 ± 7.80     | 94.09 ± 1.58     |
> | CVIR        | 98.69 ± 0.95     | 99.61 ± 0.20     | 99.88 ± 0.04     |
> | P3MIX-E     | 98.38 ± 0.45     | 96.38 ± 1.55     | 98.21 ± 1.09     |
> | P3MIX-C     | 99.07 ± 0.58     | 98.71 ± 0.48     | 97.26 ± 1.13     |
> | LBE         | 90.96 ± 3.02     | 83.66 ± 8.10     | 95.02 ± 1.73     |
> | Count Loss  | 90.46 ± 2.26     | 94.78 ± 2.16     | 94.82 ± 0.95     |
> | Robust-PU   | 92.51 ± 2.86     | 80.53 ± 7.39     | 94.88 ± 1.73     |
> | Holistic-PU | 90.11 ± 0.30     | 85.14 ± 2.38     | 90.96 ± 1.49     |
> | PUe         | 97.21 ± 1.51     | 74.09 ± 20.61    | 98.49 ± 0.46     |
> | GLWS        | 99.20 ± 0.53     | 99.81 ± 0.05     | 99.23 ± 0.55     |
> | uPU         | 96.97 ± 1.72     | 98.12 ± 0.88     | 99.12 ± 0.20     |
> | uPU-c       | 95.46 ± 0.73     | 88.54 ± 4.56     | 93.80 ± 0.71     |
> | nnPU        | 98.98 ± 0.61     | **99.92 ± 0.01** | 99.89 ± 0.02     |
> | nnPU-c      | 92.96 ± 1.83     | 92.44 ± 3.60     | 94.99 ± 0.09     |
> | nnPU-GA     | 88.02 ± 6.19     | 78.64 ± 3.11     | 95.33 ± 2.10     |
> | nnPU-GA-c   | 90.35 ± 4.04     | 83.02 ± 6.83     | 92.74 ± 0.14     |
> | PUSB        | 99.00 ± 0.56     | 99.00 ± 0.56     | 99.03 ± 0.74     |
> | PUSB-c      | 94.41 ± 0.81     | 90.41 ± 2.41     | 96.02 ± 0.50     |
> | Dist-PU     | **99.94 ± 0.01** | **99.92 ± 0.00** | **99.93 ± 0.01** |
> | Dist-PU-c   | 99.59 ± 0.29     | 99.16 ± 0.62     | 99.58 ± 0.28     |
>
> In the future, we will consider collecting more large-scale, real-world datasets to facilitate PU learning benchmarking.
>
> **W2: The limitation of the calibration approach for the ILS problem is not discussed.**
>
> **A2:** In this paper, we introduce the ILS problem in PU learning for the first time and present a straightforward calibration approach backed by theoretical guarantees. One limitation is that the proposed method is data-perspective based, merging positive and negative data together. This strategy may not be optimal from a performance perspective. Notably, there has been extensive research on algorithmic design for label shift adaptation. Therefore, developing more algorithms to improve ILS performance is promising.
>
>
> References:
>
> [1] https://www.kaggle.com/datasets/mlg-ulb/creditcardfraud
>
> [2] Who Is Your Right Mixup Partner in Positive and Unlabeled Learning? ICLR 2022.
>
> [3] Beyond Myopia: Learning from Positive and Unlabeled Data through Holistic Predictive Trends, NeurIPS 2023.

---

### Official Review · Reviewer_DKd8 · 2025-11-01

**Soundness:** 3
**Presentation:** 3
**Contribution:** 3
**Rating:** 8
**Confidence:** 3

**Summary:**

This paper performs an extensive benchmarking of many PU learning algorithms in a unified setting that is designed to be realistic and fair, and the results provide novel insights on the performance of those algorithms. In addition, through theoretical analysis, it identifies the problem of directly apply a two-sample PU learning algorithm to the one-sample setting, and provides a simple solution to this problem.

**Strengths:**

As far as I know, this is the first rigorous and extensive benchmarking for PU learning algorithms. I believe the benchmarking provides interesting insights that can be useful for further research in the field, and the methodology can be adopted in future work for more rigorous comparison.

I think the theoretical analysis is interesting and it seems to be sound.

The paper is well written and easy to follow.

**Weaknesses:**

Currently the benchmarking is performed on four benchmarks while some other datasets are also commonly used, and it would be helpful to have benchmarking results on some of those datasets for comparison with results in the literature. Nevertheless, current benchmarking already requires substantial effort though.

The model details seem to be missing. Table 4 mention ResNet and MLP but there are no details.

Minor comment: "accessible" doesn't seem needed as this is subsumed by "realistic".

**Questions:**

* How are the minibatch sizes for the P and U data set in Algorithm 1?
* Why is SGD used while Adam is often better?
* Training is done for 20,000 iterations. Is this generally large enough to ensure that the models are sufficiently well-trained, despite dataset size differences?

---

> ### Author Response · Authors · 2025-11-25
> **Response to Reviewer DKd8**
>
> First, we would like to thank you for taking the time to review our paper. We are very grateful for your helpful, high-quality comments and suggestions. Below are our answers (A) to the weaknesses (W) and questions (Q).
>
> **W1: More benchmark results.**
>
> **A1:** We conducted additional experiments on a real-world benchmark dataset for the Kaggle Credit Card Fraud Detection task [1], which is also used in PU learning literature[2,3]. We followed the previous experimental setup and used the MLP as the model architecture. The other settings are the same as those of the tabular datasets in the paper. Below are parts of the accuracy results. The full results can be found in Appendix F of the revised version of the paper.
>
> | Val Metric | PA   | PAUC  | OA   |
> | ----------- | ---------------- | ---------------- | ---------------- |
> | PUbN        | 96.31 ± 2.01     | 90.02 ± 5.66     | 97.13 ± 0.89     |
> | PAN         | 94.54 ± 2.90     | 19.44 ± 7.80     | 94.09 ± 1.58     |
> | CVIR        | 98.69 ± 0.95     | 99.61 ± 0.20     | 99.88 ± 0.04     |
> | P3MIX-E     | 98.38 ± 0.45     | 96.38 ± 1.55     | 98.21 ± 1.09     |
> | P3MIX-C     | 99.07 ± 0.58     | 98.71 ± 0.48     | 97.26 ± 1.13     |
> | LBE         | 90.96 ± 3.02     | 83.66 ± 8.10     | 95.02 ± 1.73     |
> | Count Loss  | 90.46 ± 2.26     | 94.78 ± 2.16     | 94.82 ± 0.95     |
> | Robust-PU   | 92.51 ± 2.86     | 80.53 ± 7.39     | 94.88 ± 1.73     |
> | Holistic-PU | 90.11 ± 0.30     | 85.14 ± 2.38     | 90.96 ± 1.49     |
> | PUe         | 97.21 ± 1.51     | 74.09 ± 20.61    | 98.49 ± 0.46     |
> | GLWS        | 99.20 ± 0.53     | 99.81 ± 0.05     | 99.23 ± 0.55     |
> | uPU         | 96.97 ± 1.72     | 98.12 ± 0.88     | 99.12 ± 0.20     |
> | uPU-c       | 95.46 ± 0.73     | 88.54 ± 4.56     | 93.80 ± 0.71     |
> | nnPU        | 98.98 ± 0.61     | **99.92 ± 0.01** | 99.89 ± 0.02     |
> | nnPU-c      | 92.96 ± 1.83     | 92.44 ± 3.60     | 94.99 ± 0.09     |
> | nnPU-GA     | 88.02 ± 6.19     | 78.64 ± 3.11     | 95.33 ± 2.10     |
> | nnPU-GA-c   | 90.35 ± 4.04     | 83.02 ± 6.83     | 92.74 ± 0.14     |
> | PUSB        | 99.00 ± 0.56     | 99.00 ± 0.56     | 99.03 ± 0.74     |
> | PUSB-c      | 94.41 ± 0.81     | 90.41 ± 2.41     | 96.02 ± 0.50     |
> | Dist-PU     | **99.94 ± 0.01** | **99.92 ± 0.00** | **99.93 ± 0.01** |
> | Dist-PU-c   | 99.59 ± 0.29     | 99.16 ± 0.62     | 99.58 ± 0.28     |
>
> In the future, we will consider collecting more large-scale, real-world datasets to facilitate PU learning benchmarking.
>
> **W2: Details of network architectures are missing.**
>
> **A2:** We used ResNet-34 and for image datasets and a multilayer perceptron (MLP) with a hidden layer width of 500 equipped with the ReLU activation function for tabular datasets. We have added these descriptions to the revised version of the paper.
>
> **W3: "Accessible" does not seem necessary since it is included in "realistic."**
>
> **A3:** Here, "accessible" evaluation corresponds to our paper's first contribution: proposing the first PU learning benchmark. Previous experimental evaluations of PU learning are not unified or convenient, but our proposal facilitates the evaluation of PU learning algorithms and makes it easier for users to implement newly proposed algorithms. Therefore, we consider our evaluation framework to be accessible.
>
> **Q1: What are the batch sizes?**
>
> **A4:** The hyperparameter details can be found in Table 4, where the batch size was selected from the set \{32, 64, 128, 256\}.
>
> **Q2: Why is SGD used?**
>
> **A5:** We used SGD as the optimizer in our experiments to ensure fairness. However, it is up to the user to decide which optimizer to use, as our framework is flexible.
>
> **Q3: Is the iteration number sufficient?**
>
> **A6:** Since the dataset used in the paper is not large-scale, we found that the iteration number was sufficient for the model to converge, according to our experimental results.
>
> References:
>
> [1] https://www.kaggle.com/datasets/mlg-ulb/creditcardfraud
>
> [2] Who Is Your Right Mixup Partner in Positive and Unlabeled Learning? ICLR 2022.
>
> [3] Beyond Myopia: Learning from Positive and Unlabeled Data through Holistic Predictive Trends, NeurIPS 2023.

---

### Author Response · Authors · 2025-11-28
**Summary of Our Rebuttal**

Dear SACs, ACs, and Reviewers,

We thank you for your time and efforts in reviewing our submission. Your feedback greatly enhanced our work.

We would like to respectfully point out that we tried our best to **address Reviewer WGLx's misunderstanding and concerns** about our paper, but we did not receive any feedback. Our summarized responses to the "Weaknesses" mentioned by Reviewer WGLx are as follows:
1. Regarding Weakness 1, **we did include experimental results with different positive data ratios in the initial version of the paper. However, reviewer WGLx overlooked them and claimed that we did not conduct such experiments**. In addition to the existing figures, we have included more experiments in the revised version.
2. Regarding Weakness 2, we respectfully pointed out that **it is more practical to know the class prior than to have negative data, instead of "equally impractical" commented by Reviewer WGLx.** This is because negative data is impossible to obtain in the standard PU learning setting. However, class prior estimation can be obtained in many ways, as discussed in the rebuttal. Additionally, we demonstrate the robustness of PA with inaccurately estimated class priors.
3. Regarding Weakness 3, the effectiveness of PA and PAUC can be examined by comparing them horizontally. They perform comparably against OA, which cannot be calculated in real-world PU learning scenarios and should be considered an unrealizable upper bound.
4. Regarding Weakness 4, the merging strategy is simple yet effective. Observing significant differences between one-sample and two-sample settings is novel and important for the PU learning literature. Furthermore, we provide solid theoretical guarantees that validate its effectiveness. We believe this strategy is a significant contribution to PU learning and ensures fair algorithmic comparisons.

We are pleased that several of our strengths and contributions were recognized, including:
1. **The first rigorous and comprehensive PU learning evaluation framework (DKd8, c2Ap, BgWp):** The proposed evaluation framework can greatly promote the standardized, fair, and user-friendly evaluation of PU learning algorithms.
2. **Novel and useful model selection and evaluation protocols (DKd8, c2Ap, BgWp, WGLx):** The model selection criteria greatly facilitate realistic evaluation in real-world PU applications without access to negative data.
3. **Solid theoretical guarantees (DKd8, c2Ap):** We provide extensive and novel theoretical analyses of the proposed calibration technique and the model selection criteria.
4. **Good paper writing (DKd8, c2Ap, BgWp):** The paper is well-structured and clearly written.

In our rebuttal, we addressed each of the reviewers' concerns point by point and revised the manuscript accordingly. The revised text is shown in blue. The major contents include:
1. **Experiments on real-world datasets (DKd8, c2Ap, BgWp):** We included experiments on a credit card fraud detection task.
2. **Influence of class priors (BgWp, WGLx)**: We also respectfully point out that **knowing the class prior is more practical than having negative data in the standard PU learning setting**. We also emphasize that this paper focuses on algorithm evaluation rather than class prior estimation, which is an intrinsic issue in PU learning and beyond the scope of this paper. We conducted additional experiments with inaccurately estimated class priors to show the robustness of PA.
3. **Experiments with varying ratios of positive data (WGLx)**: We respectfully point out that we did include some experimental results in the initial version of the paper. We conducted additional experiments with varying ratios of positive data.

We respectfully point out that, although simple, the proposed calibration technique is **novel and theoretically sound**. It can greatly **ensure fairness in the evaluation of PU learning algorithms**. Furthermore, we believe that our framework and evaluation criteria are significant contributions that will greatly **advance the future evaluation of PU learning algorithms**.

**We believe that our rebuttal has adequately addressed your concerns**. If anything remains unclear, please feel free to contact us. Thank you again for your time and efforts in reviewing our paper.

Best,

Author of ICLR 2026 Conference Submission 9413

---

### Meta-Review · Area_Chair_jjC4 · 2026-01-06

**Summary:**

The authors propose a unified benchmark for Positive-Unlabeled (PU) learning to address the lack of consistent experimental protocols in the field. They identify two critical issues in existing evaluations: the unrealistic reliance on negative data for validation and the "Internal Label Shift" (ILS) problem caused by differences between one-sample (OS) and two-sample (TS) data generation settings. To resolve these, the paper introduces a systematic investigation of model selection criteria that do not require negative data, and proposes a simple yet arguably effective calibration approach. The authors also carry out various experiments to demonstrate that uncalibrated methods fail and that the proposed framework ensures fair and realistic comparisons.
As the reviewers noted, a major strength is that the authors establish a rigorous, comprehensive benchmark that identifies and fixes the overlooked Internal Label Shift (ILS) problem. Three reviewers (DKd8, c2Ap, BgWp) strongly endorsed the work for its "excellent" soundness, clear presentation, and the practical value of removing the dependency on negative validation data. However, a dissenting reviewer (WGLx) criticized the contribution as "poor," arguing that the proposed calibration technique is mathematically redundant (equivalent to simply merging datasets) and that the assumption of a known class prior is just as impractical as having negative data. I think the authors response adequately addressed this critique and thus the paper is a good fit for ICLR.

**Reviewer Concerns:**

In my opinion the authors addressed all concerns.

**Reviewer Scores:**

Reviewer DKd8 (Current: 8): Likely to remain 8.
Reviewer c2Ap (Current: 8): Likely to remain 8.
Reviewer BgWp (Current: 8): Likely to remain 8.
Reviewer WGLx (Current: 2): Likely to remain 2 or increase slightly to 3. This reviewer appears fundamentally opposed to the novelty of the solution (viewing it as redundant) and the practicality of the problem setup.

---

### Decision · Program_Chairs · 2026-01-26

Accept (Poster)